# Functional regimes define soil microbiome response to environmental change

Kiseok Keith Lee[1,2,3,9], Siqi Liu[4,9], Kyle Crocker[1,2,3], Jocelyn Wang[1], David R. Huggins[5], Mikhail Tikhonov[6✉], Madhav Mani[3,7,8✉] & Seppe Kuehn[1,2,3,8✉]

The metabolic activity of soil microbiomes has a central role in global nutrient cycles[1]. Understanding how soil metabolic activity responds to climate-driven environmental perturbations is a key challenge[2,3]. However, the ecological, spatial and chemical complexity of soils[4–6] impedes understanding how these communities respond to perturbations. Here we address this complexity by combining dynamic measurements of respiratory nitrate metabolism[7] with modelling to reveal functional regimes that define soil responses to environmental change. Measurements across more than 1,500 soil microcosms subjected to pH perturbations[8,9] reveal regimes in which distinct mechanisms govern metabolite dynamics. A minimal model with two parameters, biomass activity and growth-limiting nutrient availability, predicts nitrate utilization dynamics across soils and pH perturbations. Parameter shifts under perturbation reveal three functional regimes, each linked to distinct mechanisms: (1) an acidic regime marked by cell death and suppressed metabolism; (2) a nutrient-limited regime in which dominant taxa exploit matrix-released nutrients; and (3) a resurgent growth regime driven by exponential growth of rare taxa in nutrient-rich conditions. We validated these model-derived mechanisms with nutrient measurements, amendment experiments, sequencing and isolate studies. Additional experiments and meta-analyses suggest that functional regimes are widespread in pH-perturbed soils.

The metabolic activity of soil, marine and freshwater microbiomes drives carbon and nitrogen transformations that sustain biogeochemical cycles and life in the biosphere[2]. These microbiomes are also subjected to environmental perturbations including changes in temperature, pH, moisture, oxygen and nutrients stemming from natural and anthropogenic events[10,11]. To predict the effect of climate change on global nutrient cycles, it is necessary to understand how microbiome metabolism responds to environmental change in nature.

Determining how environmental change affects community metabolism has proved difficult owing to the complexity of natural microbiomes. This complexity is perhaps most apparent in soils, which possess immense taxonomic diversity[12], spatial heterogeneity[5] and chemically diverse environments[13]. As a result, environmental perturbations can modify collective metabolic activity in many ways, from direct changes in microbial composition, physiology[14] and ecological interactions[8] to indirect modification of nutrient availability[15,16] and spatial organization[17]. Thus, a key question is which mechanisms determine the metabolic response of complex microbiomes to environmental change.

Large-scale surveys approach this question by quantifying correlations between environmental variation, community composition and metabolic processes in the wild[4,18–21]. Although surveys have revealed robust correlations, they face two challenges in uncovering the mechanisms determining community response to environmental change. First and most importantly, surveys cannot control for confounding factors such as correlated environmental variables, rendering any causal inference infeasible. Second, it is difficult to quantify metabolic dynamics in situ on a large scale in the wild. As a result, surveys have limited power for determining the mechanisms that govern the metabolic response to environmental change in natural communities.

To control for confounding factors and gain mechanistic insights, we use soil microcosms—this removes correlated environmental fluctuations and permits controlled perturbations in the laboratory. To further control for confounding factors, these soils are sourced from a single site (Cook Agronomy Farm, WA, USA)[22] that exhibits large natural pH variation but minimal variability in other environmental factors. Crucially, soil microcosms enable high-throughput quantification of metabolite dynamics in response to environmental perturbations. Leveraging insights from global surveys, we focus on pH, the environmental variable that is most strongly correlated with soil microbiome composition and metabolism[4,8,23].

Our metabolic measurements target anaerobic nitrate respiration, a central process in nitrogen cycling that is widely carried out by soil bacteria across pH gradients[24,25]. Nitrate ($NO_3^-$), which has critical

[1]Department of Ecology and Evolution, The University of Chicago, Chicago, IL, USA. [2]Center for the Physics of Evolving Systems, The University of Chicago, Chicago, IL, USA. [3]Center for Living Systems, The University of Chicago, Chicago, IL, USA. [4]Department of Engineering Sciences and Applied Mathematics, Northwestern University, Evanston, IL, USA. [5]USDA-ARS, Northwest Sustainable Agroecosystems Research Unit, Pullman, WA, USA. [6]Department of Physics, Washington University in St Louis, St Louis, MO, USA. [7]NSF-Simons Center for Quantitative Biology, Northwestern University, Evanston, IL, USA. [8]National Institute for Theory and Mathematics in Biology, Northwestern University and The University of Chicago, Chicago, IL, USA. [9]These authors contributed equally: Kiseok Keith Lee, Siqi Liu. ✉e-mail: tikhonov@wustl.edu; madhav.mani@gmail.com; seppe.kuehn@gmail.com

implications for agriculture and climate, is reduced in soils when bacteria use it as an electron acceptor during anaerobic respiration in the absence of oxygen. Both denitrification ($NO_3^- \rightarrow NO_2^- \rightarrow \ldots \rightarrow N_2$) and dissimilatory nitrate reduction to ammonia (DNRA; $NO_3^- \rightarrow NO_2^- \rightarrow NH_4^+$) reduce nitrate to nitrite ($NO_2^-$) while consuming organic carbon.

We measure nitrate utilization dynamics in more than 1,500 microcosms across a wide range of natural and laboratory-induced pH changes. A judicious dynamic model describes nitrate utilization across microcosms in terms of three variables: the quantity of a single functional biomass, nitrate, and a limiting nutrient. Changes in nitrate dynamics in response to perturbations arise from differences in two model parameters that vary with pH: the initial quantity of biomass activity utilizing nitrate and the initial quantity of limiting nutrients. These two parameters emerge naturally from our mathematical model using only the community-level nitrate uptake data. The model predicts that changes in pH influence nitrate utilization dynamics through mechanisms that affect biomass activity and nutrient availability, and these predictions are validated experimentally. We demonstrate the generality of these findings through experiments on soils from other sampling sites and a quantitative meta-analysis of past studies.

Despite the ecological, chemical and spatial complexity of soils, we find that the functional response of the soil microbiome to changes in pH can be categorized into three mechanistically distinct regimes demarcated by the levels of these two parameters. Each functional regime is defined by which of the two parameters exerts greater control over nitrate utilization rates. During moderate pH perturbations, metabolic rates are set by the pH-mediated release of nutrients from soil particles that limit the growth of biomass (nutrient-limiting regime, Regime II). When soils are subjected to large basic perturbations, massive nutrient release relieves the nutrient limitation, but the dominant taxa are no longer metabolically active, and metabolism is set by the rapid growth of rare taxa (resurgent growth regime, Regime III). During large acidic perturbations, functional responses are limited by the pervasive death of the functional biomass in the community (acidic death regime, Regime I). The transition between functional regimes can be abrupt (Regime II to III) or smooth (Regime I to II) as pH is varied. Although the presence of functional regimes is consistent across soils, the pH at which regime transitions occur varies according to the long-term pH of the soil. Our study presents a generalizable approach in which high-throughput soil microcosm experiments and mathematical models reveal the microscopic processes driving microbiome responses to environmental change.

## Soil metabolite dynamics change with pH

We measured nitrate utilization dynamics in soil microcosms across a range of native and perturbed pH levels. We sampled 20 topsoils with native pH from 4.7 to 8.3 (Fig. 1a and Supplementary Table 1) at the Long-term Agricultural Research Cook Agronomy Farm (CAF) (Pullman, WA, USA). At this site, long-term variation in soil pH arises from agricultural practices and differential erosion. Although 20 CAF sites had similar soil texture (silty clay loam) (Supplementary Table 1), their variation in soil pH correlated well with cation exchange capacity, sulfur and phosphorous levels (Extended Data Fig. 8q, Supplementary Information and Supplementary Table 3).

For each soil sample, we created mixtures of soil and water (slurries) with 2 mM nitrate and varying levels of strong acid or base to perturb the pH of each soil to 13 values between 3 and 9 (Fig. 1a). By applying pH perturbations to soils sampled across a gradient of native pH values, our experiment probed short-term and long-term responses to perturbations. We used slurries to make amendments easier, limit the effects of differential water content and mimic rain events, when most anaerobic respiratory nitrate utilization occurs[26]. Soil slurries retained much, but not all, of the complexity of the natural context, including

the diversity of the communities, the soil nutrient composition and the spatial structure due to intact soil grains.

To separate the activity of pre-existing nitrate utilizers from growth in each condition[27], we included controls treated with chloramphenicol, which inhibits protein synthesis (Fig. 1a,b). In each microcosm, we focused on the dynamics of nitrate during the four-day incubation in anaerobic conditions (Fig. 1a). Focusing on a non-gaseous metabolite enabled us to perform measurements of metabolite dynamics with high temporal resolution across the approximately 1,500 microcosms. In addition, we selected ten soils spread evenly across the native pH gradient and performed 16S ribosomal RNA (rRNA) amplicon sequencing before and after incubation.

Nitrate dynamics for a subset of soils and pH perturbations are shown in Fig. 1b. All chloramphenicol-treated conditions, regardless of soil or pH, exhibited linear nitrate dynamics (Fig. 1b and Extended Data Fig. 1). Chloramphenicol is bacteriostatic and inhibits protein synthesis, arresting growth while leaving existing enzymes intact; as a result, nitrate is reduced at a constant rate (linear dynamics). Thus, in chloramphenicol-treated conditions, the slope of nitrate decline quantifies the activity of the pre-existing nitrate-reducing biomass at each pH[26]. By contrast, non-chloramphenicol conditions reflect metabolite dynamics influenced by both pre-existing activity and growth.

Across pH perturbations and soils, we observed three distinct nitrate utilization regimes (Fig. 1b). First, under strong acidic perturbations, nitrate reduction was minimal in both chloramphenicol-treated and untreated conditions (Fig. 1b, left columns), indicating little pre-existing nitrate-reducing biomass and no growth. Second, at pH levels near the native pH, nitrate declined linearly even in non-chloramphenicol samples (Fig. 1b and Extended Data Fig. 1), at faster rates than chloramphenicol-treated controls. This suggests that some growth occurred in non-chloramphenicol conditions but was inhibited, potentially by a lack of nutrients other than nitrate (Fig. 1c, schematic). Third, under strongly basic conditions (pH > 8), non-chloramphenicol samples showed accelerating nitrate reduction, whereas chloramphenicol-treated samples showed little activity (Fig. 1b, right columns). The lack of nitrate reduction in chloramphenicol-treated suggests that the indigenous nitrate-utilizing population is small, but this rare population expands rapidly, exhausting nitrate in non-chloramphenicol conditions.

## Model captures metabolite dynamics

To describe the nitrate dynamics, we used the consumer-resource model presented in Fig. 2. The model subsumes the ecological complexity of the soil microbiome into a single effective biomass rather than explicitly considering the multitude of taxa and their interactions. The model includes three variables: functional nitrate-utilizing biomass ($x$), nitrate concentration ($A$) and a growth-limiting nutrient ($C$), along with five parameters: consumption rates ($r_A$ and $r_C$), affinities ($K_A$ and $K_C$) and a biomass growth rate ($\gamma$).

The model has two key properties. First, the nitrate utilization rate ($\dot{A}$) is proportional to the amount of functional biomass ($x$), even in the absence of growth. Thus, in chloramphenicol-treated conditions where biomass does not grow ($\dot{x} = 0$), the consumption rate of $A$ is determined by $r_A x$ and is constant over time, as observed experimentally (Fig. 1b). Second, the uptake of $A$ and $C$ by functional biomass follows a co-limiting Monod form, such that depletion of either resource halts growth. Together, these properties imply that when $C$ is exhausted, growth ceases and nitrate ($A$) is consumed at a constant rate proportional to $x$.

To build intuition for how the model captures the observed nitrate dynamics, the right two columns of Fig. 2 illustrate two regimes. When the initial concentration of the limiting nutrient $C(0)$ is low (middle column), $C$ is depleted early (at time $t^*$), halting growth and leading to constant nitrate ($A$) consumption thereafter (grey dashed line).

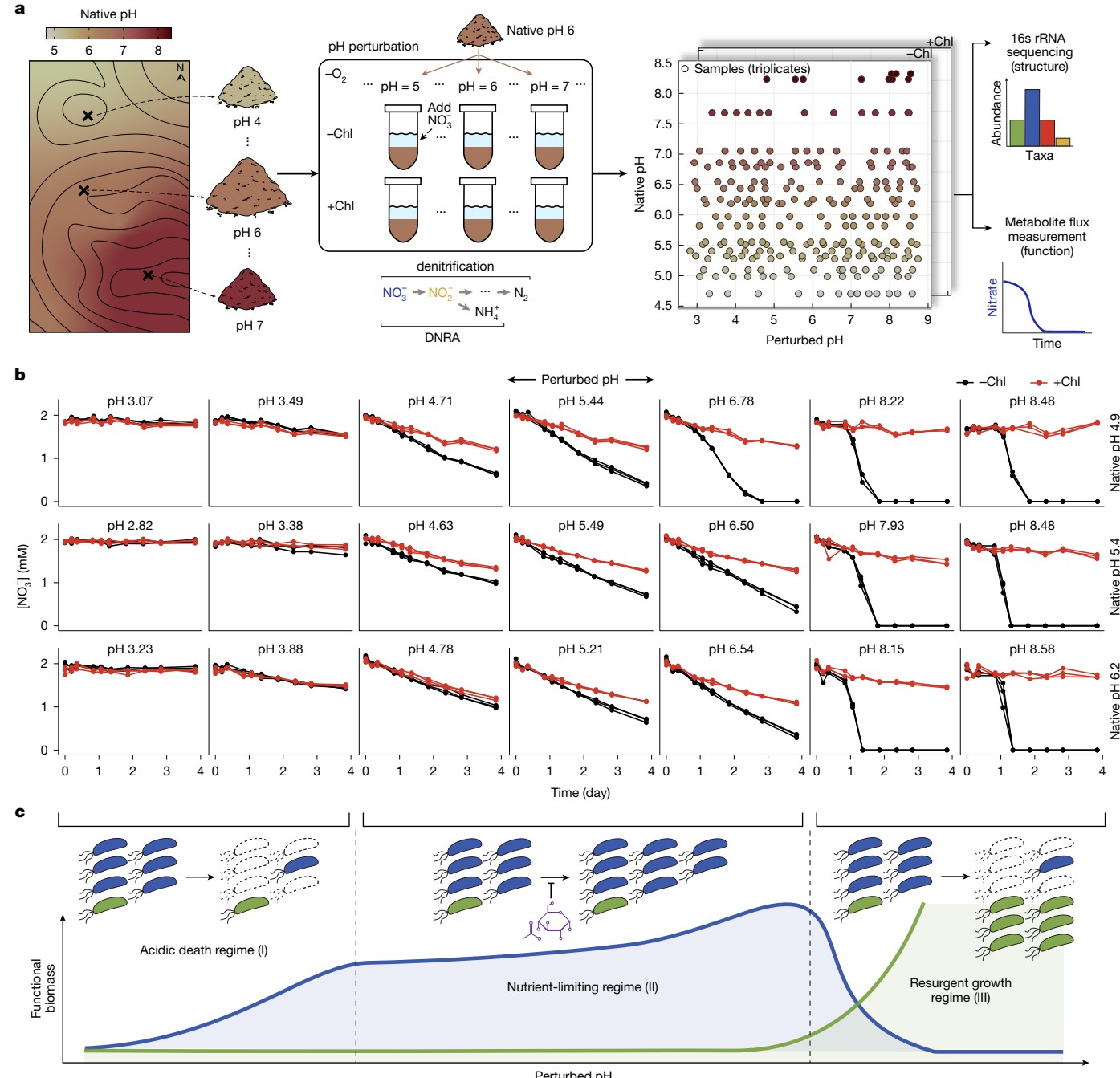

**Fig. 1 | Soil microbiome metabolite dynamics under short-term and long-term pH variation. a**, Schematic of field sampling across a long-term pH gradient (*n* = 20 soils, pH 4.7 to 8.3) subjected to short-term pH perturbations (*n* = 13) in laboratory conditions (Methods). Soil microcosms were created by making slurries (1:2, soil:water) amended with 2 mM nitrate, adjusted to 13 different pH levels, and treated with (no growth) or without chloramphenicol (growth) in triplicate (*n* = 1, 704 microcosms including no-nitrate (*n* = 120) and cycloheximide controls (*n* = 120)). Microcosms were incubated anaerobically for four days, and nitrate was quantified colorimetrically via sampling. Communities were quantified by 16S rRNA amplicon sequencing before and after incubation. Chl, chloramphenicol. **b**, Nitrate concentration over time for 3 of 20 soils with different native pH, perturbed to either acidic or basic pH.

Results are shown with or without chloramphenicol treatment (*n* = 126 microcosms). Endpoint pH is indicated in each graph (Methods). **c**, Schematic of functional regimes. For moderate pH perturbations (middle), the growth of dominant taxa (blue) is limited by available nutrients (purple) and nitrate dynamics are linear (nutrient-limiting, Regime II). During strong basic perturbations (right), growth-limiting nutrients released from the soil matrix are in excess and rare taxa (green) dominate growth (resurgent growth, Regime III). Acidic perturbations show minimal activity, partly owing to cell death (acidic death, Regime I). Lines depict dominant (blue) and rare (green) biomass. See Extended Data Fig. 1 for nitrate dynamics across all slurry experiments.

This recapitulates the late-time linear dynamics in non-chloramphenicol conditions for moderate pH perturbations (Fig. 1b). By contrast, when the initial nutrient concentration $C(0)$ is large (Fig. 2, right column), it is nitrate ($A$) that runs out first. Therefore, a small $x(0)$ and a large $C(0)$

recapitulate the initially slow but accelerating rate of nitrate utilization observed for basic perturbations (Fig. 2, right column).

We fit this model to nitrate utilization dynamics from all 20 soils and 13 pH perturbations, with and without chloramphenicol treatment.

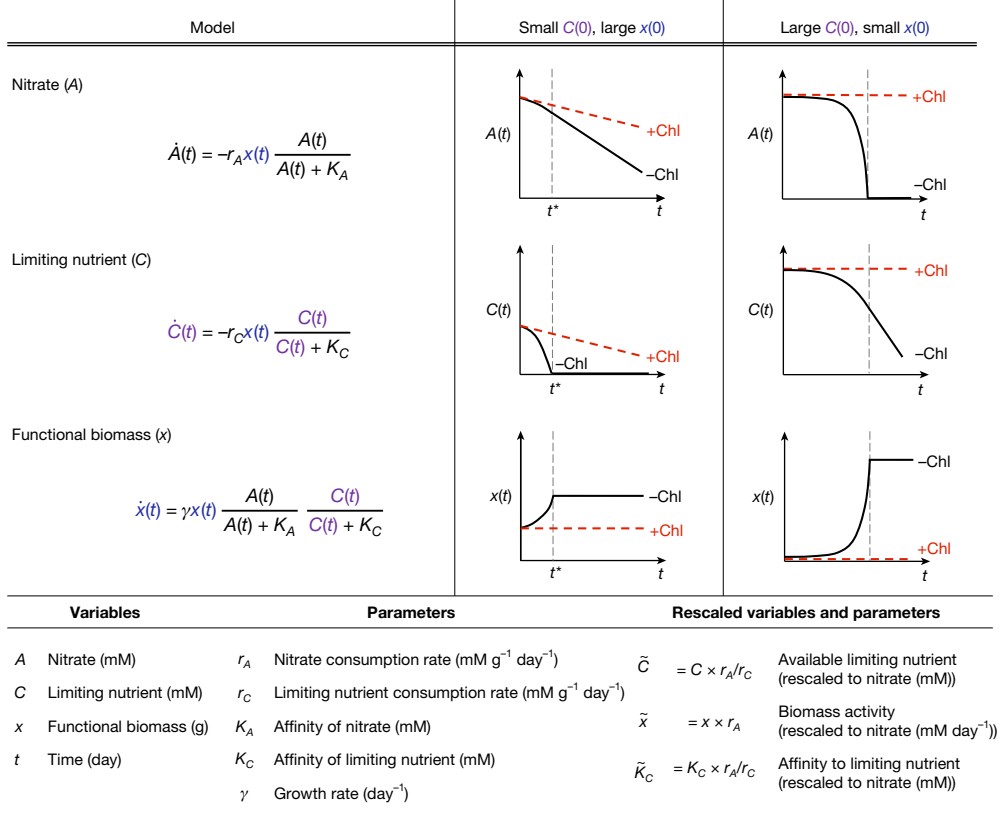

**Fig. 2 | Consumer-resource model describes metabolite dynamics.**
Left column, a consumer-resource model describes the community-level metabolism via a single functional biomass ($x$), nitrate concentration ($A$ (mM)) and a growth-limiting nutrient concentration ($C$ (mM)). Nitrate consumption rate ($\dot{A}(t)$) takes a Monod form with a reduction rate parameter ($r_A$ (mM per g biomass per day)) and an affinity parameter ($K_A$ (mM)). A non-substitutable nutrient ($C$) is consumed at a rate $r_C$ (mM per g biomass per day) with affinity $K_C$ (mM). Growth of functional biomass ($\dot{x}(t)$) is determined by the product of Monod uptake terms for non-substitutable nutrients ($A$ and $C$) and the growth rate ($\gamma$ (day$^{-1}$)). If either nutrient is exhausted, growth halts. The middle and right columns show dynamics of $x(t)$, $A(t)$ and $C(t)$ for two regimes. Chloramphenicol-treated conditions always show constant $A$ and $C$ utilization rates. Middle column, small $C(0)$, large $x(0)$. With low initial $C(0)$ and large $x(0)$, $C$ is rapidly exhausted (at $t^*$), arresting biomass growth (bottom) and resulting in constant $A$ utilization (top). Right column, large $C(0)$, small $x(0)$. Excess $C(0)$ and small $x(0)$ lead to exponential growth in functional biomass, driving exponential depletion of $A$ (black line, top). Growth arrests when $A$ is exhausted. We assume $K_A \ll A(0)$ and $K_C \ll C(0)$ (Methods). Tables define variables and parameters. See Extended Data Fig. 3a–c.

We fixed the growth rate $\gamma$ and the affinity parameters ($K_A$ and $K_C$), and varied two rescaled parameters: $\tilde{x}(0) = x(0)r_A$ and $\gamma\tilde{C}(0) = \gamma C(0)r_A/r_C$ (Extended Data Fig. 3c and Methods). These parameters retain the same interpretation as $x$ and $C$: $\tilde{x}(0)$ reflects the indigenous metabolic activity of all taxa that can perform nitrate reduction in a given condition, and $\gamma\tilde{C}(0)$ reflects the available limiting nutrient. The rescaling corresponds to measuring these quantities in terms of nitrate utilization rates, and their values determine the metabolite dynamics as shown in Fig. 2. The model provided a good fit in all soils (less than 10% error per data point, Extended Data Fig. 3a,b).

## Model reveals functional regimes

We plotted indigenous biomass activity ($\tilde{x}(0)$) against available limiting nutrient ($\gamma\tilde{C}(0)$) (Fig. 3a) and identified three regimes of nitrate utilization dynamics (Extended Data Fig. 3d–g and Methods). Regime I—the acidic death regime—in which both $\tilde{x}(0)$ and $\gamma\tilde{C}(0)$ are low, is observed for pH ≤ 4 and shows little to no nitrate reduction (Fig. 3b(i),(iv)). Regime II—the nutrient-limiting regime—in which $\tilde{x}(0)$ is large and $\gamma\tilde{C}(0)$ is small, is observed for 4 ≤ pH ≤ 8 and exhibits constant nitrate reduction rates with and without chloramphenicol treatment, with higher rates in the latter (Fig. 3b(ii),(v)). Regime III—the resurgent growth regime—in which $\tilde{x}(0)$ is small and $\gamma\tilde{C}(0)$ is large, is observed for pH ≥ 8 and displays a near-zero initial utilization rate,

followed by an exponential speed-up that continues until nitrate is depleted (Fig. 3b(iii),(vi)). We observe the three functional regimes in all soils, but the exact pH at which a given soil transitions between regimes depends on the native pH. This is illustrated in Fig. 3c,d, in which the indigenous biomass activity and the quantity of the limiting nutrient are shown as heatmaps for soils at different perturbed and native pH; the same data are presented in Fig. 3e,f as lines with the $x$ axis corresponding to perturbed pH.

The pH-dependent changes in $\tilde{x}(0)$ (Fig. 3c,e) reflect shifts in indigenous biomass activity. These changes may result from variations in the abundance of nitrate-reducing taxa, differential expression of relevant enzymes, or changes in enzymatic activity, all influenced by pH. To begin to determine the dominant mechanisms that govern changes in metabolism across regimes, we quantified compositional changes through sequencing, which enabled us to test model predictions.

## Taxonomic patterns across regimes

We measured absolute abundances through 16S amplicon sequencing after the four-day incubation in both chloramphenicol-treated and untreated conditions in half of the soils shown in Fig. 3. We computed phylum-level growth folds as the ratio of endpoint absolute abundance between untreated and chloramphenicol-treated conditions (Extended Data Fig. 5b), where phylum-level abundance refers to the aggregate

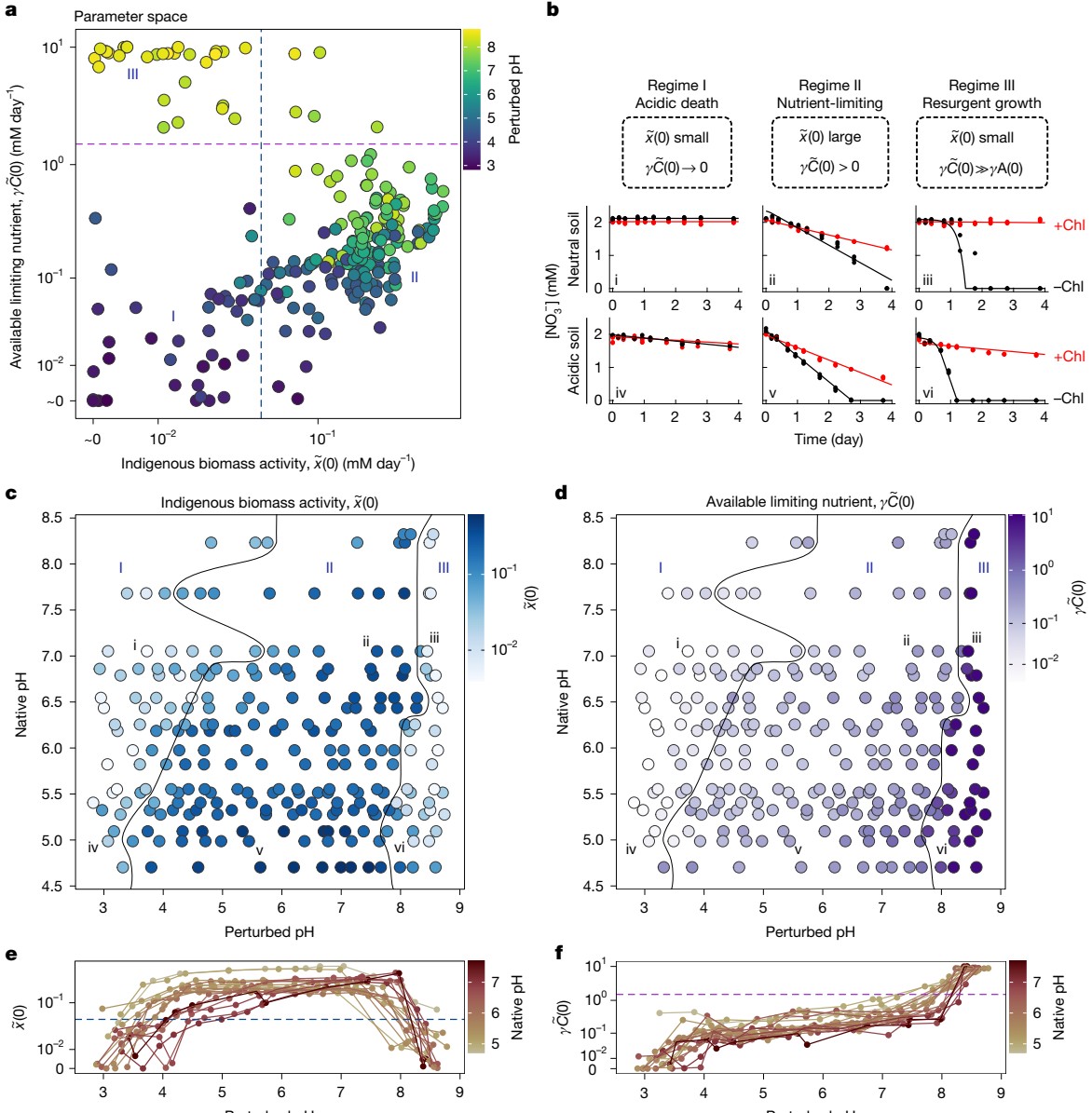

**Fig. 3 | Functional regimes capture soil response to pH perturbations.**
**a**, Scatter plot of two model parameters—indigenous biomass activity ($\tilde{x}(0)$) and limiting nutrient concentration ($\gamma\tilde{C}(0)$)—inferred from nitrate dynamics across all samples ($n = 244$ conditions, representing median fitted parameter values from 3 biological replicates, $n = 732$ dynamics). Data are $\log_{10}(x + 0.01)$-transformed for visualization. Point colour indicates perturbed pH. Dashed lines separate three functional regimes: acidic death (Regime I), nutrient-limiting (Regime II) and resurgent growth (Regime III). Regime boundaries are set by thresholds in $\tilde{x}(0)$ and $\gamma\tilde{C}(0)$ (Extended Data Fig. 3d,e and Methods).

**b**, Example nitrate dynamics in each regime for neutral and acidic soils. Panel labels correspond to points in **c**,**d**. **c**,**d**, Indigenous biomass (**c**) and limiting nutrient (**d**) for different native pH and perturbed pH values ($n = 244$ conditions). Colour indicates fitted values. Black lines indicate regime boundaries. **e**, $\tilde{x}(0)$ trends across perturbed pH values for soils with different native pH, showing regime transitions and a plateau of high activity in Regime II. **f**, Same as **e**, but for $\gamma\tilde{C}(0)$, showing increased limiting nutrients under basic pH. Points in **e**,**f**, show the median of three replicates ($n = 244$). Horizontal dashed lines mark thresholds from **a** (Extended Data Fig. 3d,e).

of all amplicon sequence variants (ASVs) within that phylum. A growth fold greater than 1 indicates increased abundance in the absence of the drug. We used non-negative matrix factorization (NMF) to decompose the variation in growth at the phylum level across all soils and pH perturbations (Methods). The analysis showed that most of the growth could be captured with just two axes of variation (Extended Data Fig. 5c). One axis comprised Pseudomonadota combined with Bacteroidota, and the other comprised Bacillota alone.

Figure 4a,b shows growth folds for the two groups of phyla identified by NMF that dominate growth across all soils and pH conditions. In Regime II, Pseudomonadota and Bacteroidota showed increased growth with increasing pH, followed by a decline near the onset of

Regime III. This matches the behaviour of the indigenous biomass activity ($\tilde{x}(0)$) revealed by the model in Regime II (Fig. 3c). Bacillota did not grow until a pH threshold between 7 and 8.5, which matches the onset of exponential nitrate utilization dynamics in Regime III (Fig. 3f). Notably, the boundary between Regime II and Regime III derived from the functional dynamics data (Fig. 3c,d) aligns with the increased growth of Bacillota (Fig. 4b) and declining growth of Pseudomonadota and Bacteroidota (Fig. 4a). These patterns suggest that changes in the identity of the phyla responsible for nitrate reduction reflect functional regimes.

An analysis of the likely metabolic traits of these taxa[28] suggests that the transition from Regime II to Regime III is accompanied by a

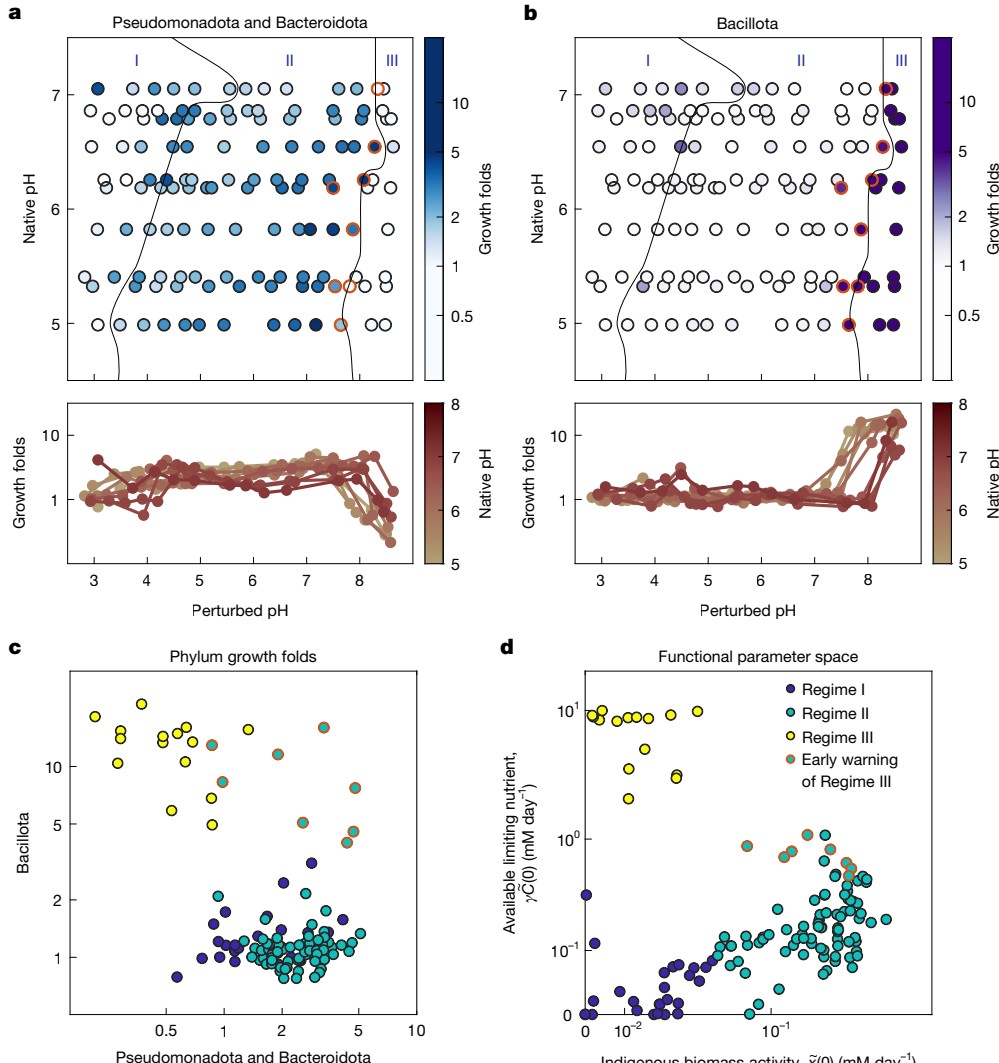

**Fig. 4 | Taxonomic signatures of functional regimes.** 16S rRNA amplicon sequencing at the end of each incubation was used to identify ASVs in chloramphenicol-treated and untreated conditions (Methods). ASVs were aggregated at the phylum level. For each phylum, a growth fold was computed as the ratio of abundance with growth to that without growth. A statistical decomposition across all conditions identified two groups of phyla that dominated abundance changes due to growth: with Pseudomonadota and Bacteroidota in one group, and Bacillota in another (Extended Data Fig. 5a–c). **a**, Top, growth folds for the aggregated abundances of Pseudomonadota and Bacteroidota (combined abundance) are indicated by colour for each native and perturbed pH condition ($n = 130$ conditions). Regime boundary lines are those determined in Fig. 3. Bottom, growth folds are plotted on a log scale,

with colour indicating native pH given in the colour bar. **b**, Identical to **a**, but showing growth folds of Bacillota increasing during the transition from Regime II to Regime III after strong basic pH perturbations ($n = 130$ conditions). **c**, Scatter plot of growth folds of Pseudomonadota and Bacteroidota versus Bacillota ($n = 130$ conditions). Points outlined in red are associated with Regime II and exhibit high growth for all three phyla. **d**, As in Fig. 3a, plot of $\tilde{x}(0)$ versus $\gamma\tilde{C}(0)$ with points marked by regime, with red outlines indicating points in Regime II conditions near the boundary with Regime III; these points are in Regime II, but they exhibit high Bacillota abundance (**c**). Data in **a**,**b**,**d** are from a subset of samples from those in Fig. 3 because only half of the soils were sequenced ($n = 10$ soils). See also Extended Data Fig. 6.

metabolic shift from denitrification to DNRA, which agrees with the fact that excess carbon favours DNRA[29] (Extended Data Fig. 6g and Supplementary Information). Consistent with the metabolic shift to DNRA, we observed an accumulation of ammonia in Regime III (Extended Data Fig. 5g)

### Increase in Bacilliota signals regime change

We observed that Bacillota began to increase in abundance at pH levels just below the transition from Regime II to Regime III, thereby acting as 'early warning indicators' for the impending transition (Fig. 4a–d, red circles). When we plot the growth folds of the Bacillota versus Pseudomonadota and Bacteroidota, we find that Bacillota abundances begin to increase before the system enters Regime III as defined by nitrate

utilization dynamics (Fig. 4c,d). This finding indicates that compositional data can be used to anticipate impending regime transitions during environmental perturbations.

Having quantified pH-induced shifts in community composition, we next explored whether these data, combined with changes in model parameters, could reveal mechanisms underlying community metabolism across regimes.

### Rare taxa grow rapidly in Regime III

Under strong basic pH perturbations, all soils enter Regime III, which is marked by a sharp shift from linear to exponential nitrate consumption dynamics in chloramphenicol-free conditions. Our model suggests that these metabolite dynamics arise from a small indigenous biomass

activity ($\tilde{x}(0)$), indicative of rare taxa undergoing exponential growth in response to excess nutrient availability ($\gamma\tilde{C}(0)$) (Fig. 3c–f). Sequencing measurements show that Bacilliota dominate growth in Regime III (Fig. 4b,d). Together, these observations suggest that initially rare members of the Bacilliota phylum are responsible for growth in Regime III (Fig. 4b). Consistent with this prediction, we found that the taxa that drive growth in Regime III start from very low abundance and are enriched several hundred fold (Extended Data Fig. 6d).

## Carbon release fuels Regime II growth

In the nutrient-limiting Regime II, nitrate reduction rates remain constant with or without chloramphenicol, but are higher in the absence of the drug. The model attributes the increased rate in the absence of the drug to the rapid initial utilization of nutrient ($C(0)$) (Fig. 2, middle column). Further, as the perturbed pH increases, the nitrate reduction rate in chloramphenicol-free conditions increases (Fig. 1b). Our model proposes that the increasing availability of the growth-limiting nutrient with pH ($\gamma\tilde{C}(0)$) (Fig. 3a,d,f) drives growth that increases nitrate reduction rate. Here, we test this hypothesis.

Increasing pH can enhance the availability of organic carbon in soils[30,31] through substitution reactions at ion exchange sites on clay particles[32,33] (Fig. 5b). Since nutrients are released via exchange reactions, we hypothesized that the increase in biomass, and therefore nitrate reduction rate, should be proportional to the amount of acid or base added to the system. In Fig. 5a, we observed precisely this trend across all soils, as evidenced by a consistent increase in nitrate reduction rate with NaOH (light blue region). The trend is specific to Regime II (Extended Data Fig. 4c), and if the data are plotted against pH, the correlation is weaker (Extended Data Fig. 4a). If the increasing rate of nitrate utilization with NaOH is due to increased nutrient availability ($\gamma\tilde{C}(0)$) driving the growth of nitrate reducers, we expect their abundances to rise with increasing NaOH. As expected, we observe a linear relationship between absolute abundances, measured via 16S rRNA amplicon sequencing, and the amount of NaOH added to the system (Extended Data Fig. 4c–h and Methods). This increase is reflected at coarse (total biomass; Extended Data Fig. 4d,e) and fine taxonomic levels (putative nitrate-reducing ASVs; Extended Data Fig. 4f–h and Supplementary Information).

The charge of the limiting nutrient is suggested by the asymmetric response to NaOH and HCl (Fig. 5a), in which NaOH releases anions and HCl releases cations (Extended Data Fig. 4j and Supplementary Information). Since NaOH increased nitrate reduction rates, we infer that the limiting nutrient is anionic. Water-soluble organic carbon (WSOC) also increased linearly with NaOH (Fig. 5c), suggesting that it, or co-released anionic N, S or P, could be the limiting nutrient.

To identify the limiting nutrient, we performed an amendment experiment on a representative soil (Soil 6, pH 5.4; Methods). We amended a soil slurry with glucose (neutral), succinate (anion when pH > p$K_a$, p$K_a$ = 4.2), acetate (anion when pH > p$K_a$, p$K_a$ = 4.75), phosphate (anion), ammonium (cation) and sulfate (anion) added in varying concentrations without perturbing pH (Extended Data Fig. 2h and Methods). We found that the amendment of carbon, but not other resources (N, S or P), increased the nitrate reduction rate, changing the linear dynamics to exponential (Fig. 5d,e), indicating that carbon was the limiting nutrient. With a single free parameter ($r_C/r_A$), our model fit the nitrate utilization dynamics in soil amended with carbon (Fig. 5d,e). The ratio $r_C/r_A$ can be interpreted as the stoichiometry of carbon to nitrate utilization (Fig. 2). This ratio is highest for glucose (2.5) and lowest for acetate (1), suggesting that glucose amendments support faster carbon utilization. This may reflect the fact that glucose is fermentable, whereas acetate is not.

The amendment experiment confirms the model prediction that a nutrient other than nitrate limits reduction dynamics and provides strong evidence against other mechanisms limiting growth, such as predation by phage or eukaryotes (Extended Data Fig. 2a,b). Critically, this insight emerged from our mathematical description of the nitrate utilization dynamics across pH perturbations.

## Biomass is reduced by death in Regime I

In response to acidic pH perturbations, the model indicates a reduction in indigenous biomass activity ($\tilde{x}(0)$) and a decrease in the availability of limiting nutrients (Fig. 3). To test whether the decline in indigenous biomass activity ($\tilde{x}(0)$) with pH is associated with bacterial abundance, we computed the fold change in endpoint absolute abundance for each phylum in chloramphenicol-treated conditions relative to abundance at the initial time point $T_0$ ('survival fold'; Extended Data Fig. 7a). Survival folds reflect the change in abundance in the absence of growth—thus, we regard this as a proxy for death. For all phyla except Bacillota, we observed a consistent decrease in survival folds during acidic perturbations (Extended Data Fig. 7a). Survival folds decline approximately linearly with the $\tilde{x}(0)$, suggesting that a reduction in biomass activity might be associated with cell death.

However, changes in survival folds might also arise from pH-dependent degradation of relic DNA[34]. To test the hypothesis that cell death reduces biomass activity in Regime I, we used isolates from our soils or strain collections representing the three phyla identified in Fig. 4. Using these isolates, we measured DNA degradation and cell death rates across a range of pH values from 3 to 7. Combining these rates with a model enabled us to conclude that the change in abundances measured via sequencing must arise at least in part from cell death (Extended Data Fig. 7c–j and Supplementary Information). Bacillota isolates exhibited lower death rates at low pH, consistent with their high survival fold in acidic conditions (Extended Data Fig. 7a). We conclude that acidic perturbations must induce some cell death, but that other physiological or ecological mechanisms are also likely to contribute (Extended Data Fig. 7a).

## Functional regimes generalize across soils

Having characterized functional regimes in one set of soil samples (CAF; Fig. 3), we investigated whether the finding of regimes could be generalized to other soils. First, we performed pH perturbation experiments and sequencing for four additional soils from LaBagh (IL, USA), Pinhook (IN, USA), CLG13 and ELG13 (Sedgwick Reserve, CA, USA) (Methods and Supplementary Table 1). For each soil, we found that our model accurately described nitrate dynamics, and the same qualitative functional regimes observed in CAF soils (Fig. 3) were also present (Fig. 6b). Namely, indigenous biomass activity $\tilde{x}(0)$ decreased, and the available limiting nutrient $\gamma\tilde{C}(0)$ increased during basic perturbations (Regime III), large $\tilde{x}(0)$ and modest $\gamma\tilde{C}(0)$ were observed in Regime II, and both $\tilde{x}(0)$ and $\gamma\tilde{C}(0)$ diminished during acidic perturbations (Regime I; Fig. 6b). Sequencing analysis identical to that presented in Fig. 4 revealed qualitatively similar taxonomic patterns in these additional soils. First, an NMF analysis at the phylum level again showed two axes of variation with Bacilliota on one axis and Pseudomonodota and Bacteriodota on the other (compare Extended Data Fig. 5a–c with Extended Data Fig. 5d–f). Second, we observed a qualitatively consistent increase in Bacillota abundance in basic conditions (Fig. 6d). Corroborating this result, we re-analysed a pH perturbation experiment from another study and again found increasing Bacteroidota abundances for modest pH perturbations and substantial Bacillota growth under basic perturbations[30] (Fig. 6e).

Next, we conducted a meta-analysis of nine denitrification studies covering soils from diverse locations and soil types (Fig. 6a and Supplementary Table 2). These studies confirmed the fundamental findings of regimes. We were able to infer $\tilde{x}(0)$ and $\gamma\tilde{C}(0)$ from denitrification rates measured for three soils from the Czech Republic[35], confirming the existence of functional regimes during pH perturbations (Fig. 6c).

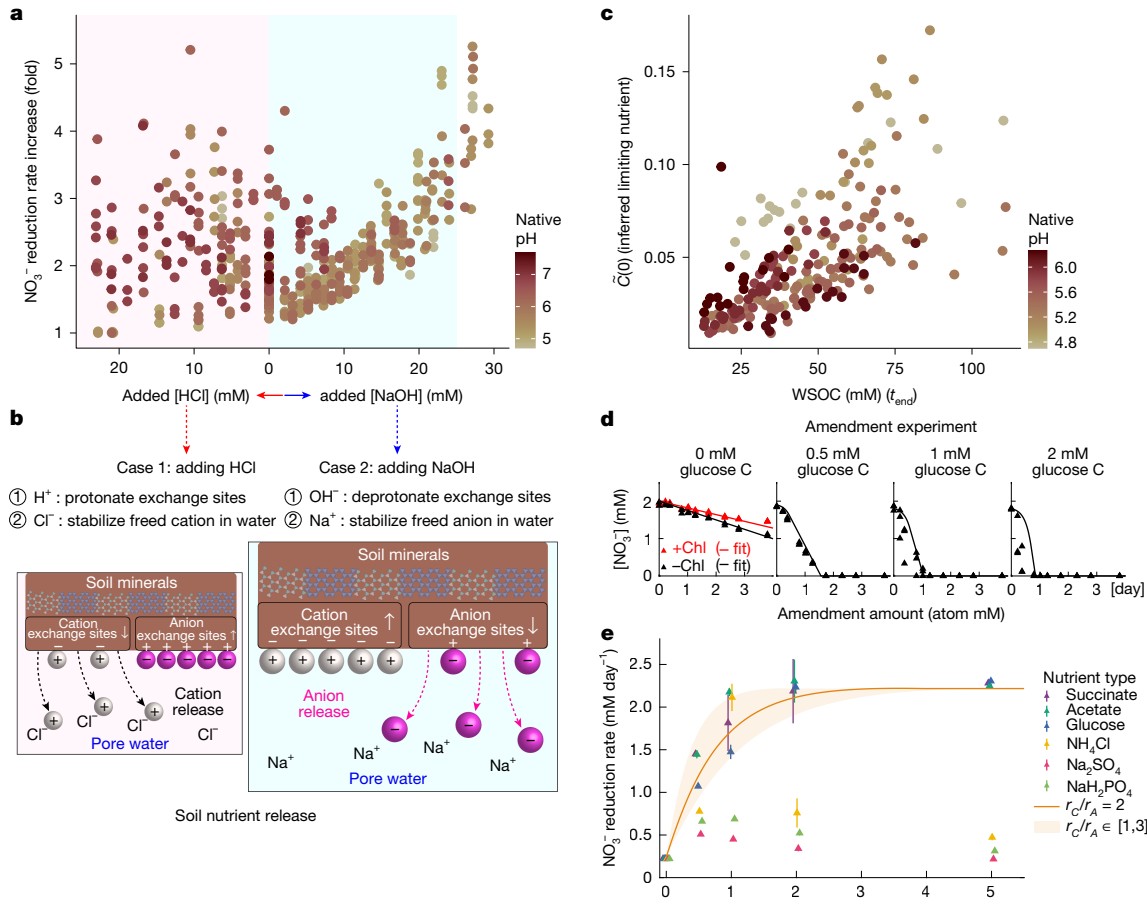

**Fig. 5 | Carbon limits growth in Regime II and is released from the soil matrix. a**, Fold change in nitrate reduction rate (without chloramphenicol/chloramphenicol-treated) as a function of the amount of NaOH added to the soil ($n = 470$ samples of <30 mM $OH^-$, <25 mM $H^+$; 28 data points with fold change >5.5 were excluded for clarity of visualization). A linear increase in fold change from 0 mM to 25 mM NaOH corresponds to the nutrient-limiting regime (Regime II, light blue region), and suggests that $\widetilde{C}(0)$ increases with increasing NaOH. **b**, Cartoon illustrating the hypothesized mechanism of nutrient release. Adding NaOH results in the release of anionic nutrients (magenta spheres, Case 2) from soil particles (brown region), whereas the addition of HCl would release cationic nutrients (grey spheres, Case 1). Microorganisms access nutrients in pore water but not those adsorbed to soil particles. Added $OH^-$ ions decrease the number of anion exchange sites, releasing anionic nutrients. $Na^+$ ions stabilize the released anions (Extended Data Fig. 4j and Supplementary

Information). Data in **a** and model in **b** suggest that growth-limiting nutrients are anionic (negatively charged). **c**, Scatter plot of model-inferred available limiting nutrient ($\widetilde{C}(0)$) with measured WSOC (endpoint WSOC, Methods) ($n = 222$ samples of 0–25 mM $OH^-$ in soil samples 1–12). **d,e**, Amendment experiments for soil in the nutrient-limiting regime at an unperturbed pH (5.4). **d**, Nitrate dynamics with different glucose amendments. Dots represent data and lines show model predictions. **e**, Nitrate reduction rates after amending soils with different concentrations of nutrients. Dots represent mean rates, estimated by linear regression on triplicates, and error bars indicate s.d. Carbon amendments (succinate, acetate and glucose) increased nitrate reduction rates, but ammonium, sulfate and phosphate did not. Model predictions are shown for $1 < r_A/r_C < 3$ (shaded region) with a line for $r_A/r_C = 2$ (best fit). See also Extended Data Figs. 2g,h and 4.

Second, we observed the same decline in indigenous biomass activity, $\widetilde{x}(0)$ (Fig. 3c,e), under extreme acidic or basic pH perturbations inferred from short-timescale measurements of denitrification rates in ten other soils across four studies[27,35–37] (Fig. 6f). Third, consistent with our Regime II, other soils showed linear utilization of electron acceptors near the native pH[38,39] (Fig. 6g). Finally, an increase in reduction rates was observed when the soils were amended with carbon (Fig. 6g), confirming our nutrient-limiting mechanism in Regime II. Together, these results demonstrate the generality of our functional regimes and mechanisms across diverse soil types and environmental conditions (Supplementary Information).

## Long-term pH sets regime boundaries

Finally, we observed that the native (long-term) pH of a soil determined the pH threshold for transitioning between functional regimes, with more acidic soils shifting from Regime II to Regime I after smaller acidic

perturbations than neutral soils (Fig. 3c,d and Extended Data Fig. 8e). This observation suggests that soil communities are adapted to their long-term pH conditions[27,40].

We suspected that this behaviour was shaped by the community's past exposure to pH fluctuations. Although we lack direct records of historical pH fluctuations, we can characterize soil response to pH perturbations using titration curves (Extended Data Fig. 8n). Across soils, these curves show a consistent shape: plateaus at high (pH 9) and low (pH 3) pH values connected by a steep, nonlinear transition. The native pH of a soil determines its position along the curve in the absence of perturbations. Acidic soils lie near the lower plateau and are therefore strongly buffered against pH changes (Extended Data Fig. 8n). By contrast, more neutral soils fall along the steep region, where small acid or base additions lead to large pH shifts. We speculate that this difference in buffering makes communities from acidic soils less tolerant to pH fluctuations, potentially explaining their transition from Regime II to Regime I after smaller perturbations (Extended Data

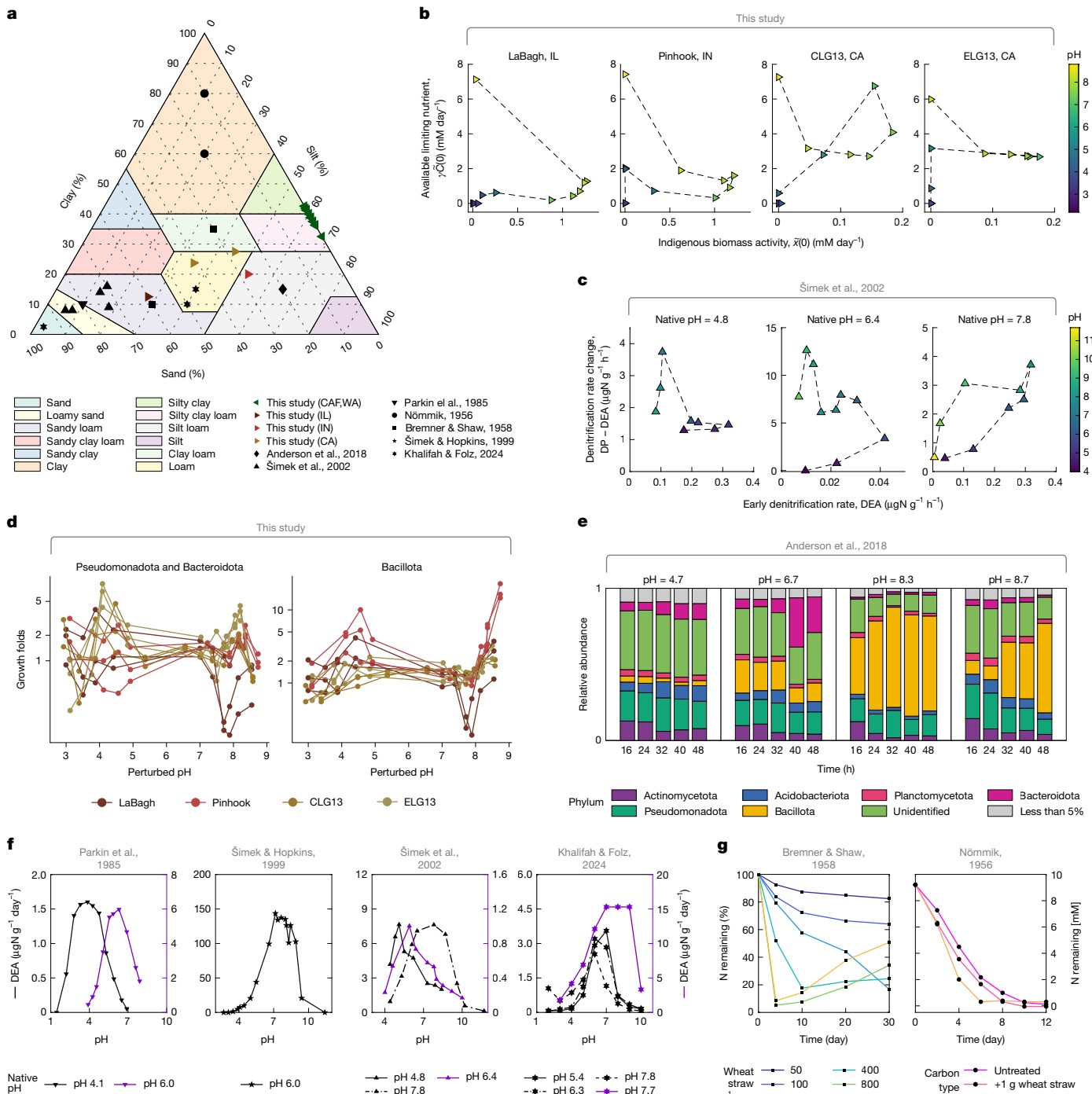

**Fig. 6 | Generality of functional regimes. a**, Ternary plot of soil texture (fractions of sand, clay and silt). Soil types are indicated by the colour of the shaded regions. Points (*n* = 43) represent soils from this study (coloured markers) or our meta-analysis (black markers; Supplementary Tables 1 and 2). Studies in the meta-analysis: Anderson et al.[30], Šimek et al.[35], Parkin et al.[27], Nömmik[38], Bremner & Shaw[39], Šimek & Hopkins[36] and Khalifah & Folz[37]. **b**, Scatter plots of fitted model parameters ($\tilde{x}(0)$ and $\gamma\tilde{C}(0)$) for soil samples from LaBagh (IL, USA), Pinhook (IN, USA) and Sedgwick CLG13 and ELG13 (CA, USA), each with *n* = 11 perturbed conditions (median values from triplicates). Colours indicate perturbed soil pH, and dashed lines connect slurries of adjacent perturbed pH levels. **c**, For three soils from the Czech Republic (Šimek et al.[35]), scatter plots of early-time denitrification rate (denitrification enzyme activity (DEA), same as $\tilde{x}(0)$)) and the difference between DEA and long-term denitrification rate (denitrification potential (DP), same as $\gamma\tilde{C}(0)$); see Supplementary Information). Colours indicate the perturbed soil pH and dashed lines connect soil samples of adjacent perturbed pH. Compare **b**,**c** with Fig. 3a. **d**, Growth

folds of Bacillota and the combined abundances of Pseudomonadota and Bacteroidota for soil samples (*n* = 132 samples) from LaBagh (IL), Pinhook (IN) and Sedgwick CLG13 and ELG13 (CA) across different pH levels (Extended Data Fig. 5d–f and Methods). The *y* axis was log(*x* + 0.01)-transformed, as in Fig. 4a,b. **e**, Relative abundance of phyla over time after pH perturbations, from Anderson et al.[30]. Bacillota abundance increases under basic perturbations (right two panels) and Bacteroidota abundance increases under moderate pH perturbations. **f**, Soils from four studies (Parkin et al.[27], Šimek & Hopkins[36], Šimek et al.[35] and Khalifah & Folz[37]) show a decline in indigenous biomass activity ($\tilde{x}(0)$) under acidic or basic pH perturbations (*n* = 10 soils). The *y* axis is equivalent to $\tilde{x}(0)$. **g**, Constant rates of nitrogen utilization near the native pH are observed in soils from previous studies (Bremner & Shaw[39] and Nömmik[38]), as in our Regime II. Amending carbon (wheat straw or glucose (g per g soil)) increased the rate of denitrification (Fig. 5). The legend indicates the type and amount of carbon amended[38,39].

Fig. 8e). Quantitative analysis of the transition point supports this interpretation (Extended Data Fig. 8a–e and Supplementary Information).

In addition, our sequencing data support the idea that microbiome composition reflects specific adaptation to native pH, helping to explain variation in regime boundaries across soils. In more acidic soils, Pseudomonadota and Bacteroidota exhibit greater survival at lower pH (Extended Data Fig. 8f). Conversely, the pH at which Bacillota begin to grow in Regime III increases with native soil pH (Fig. 4b and Extended Data Fig. 8f,g). Moreover, the identity of strains that grow in Regime III predicts the native pH of the soil (Extended Data Fig. 8h–l). These findings suggest that prolonged exposure to specific pH conditions selects for particular taxa, potentially shaping the pH thresholds at which communities shift between functional regimes (Methods).

## Discussion

Our study has important implications for theoretical microbial ecology and empirical studies of microbiome metabolism. First, the success of our model that abstracts the entire community into a single effective biomass is in stark contrast to complex models of ecosystems that capture many interacting species[41,42]. This contrast suggests that natural communities might be better understood through coarse-grained descriptions[43] that capture the handful of metabolically relevant groups or guilds in a consortium[44,45]. This low-dimensional picture of ecosystems respects underlying mechanisms and is tightly connected to observable functional properties of the community. Extending this success to more complex metabolic processes and connecting effective biomass variables such as $\tilde{x}$ to underlying abundance dynamics are key directions for future work.

Second, understanding how microbial community metabolism responds to environmental perturbations remains a central problem in applied microbiome science. This challenge arises from the complexity of communities comprising many species with diverse metabolic traits and ecological interactions taking place in chemically complex environments. The complexity of communities has motivated increasingly sophisticated measurements, from metagenomics and transcriptomics[46,47] to single-cell metabolomics[48] and quantitative stable isotope probes[49,50]. Yet, connecting these data to community metabolism and its response to environmental perturbations has remained difficult. Our findings suggest that instead of focusing on microscopic processes individually, we first make quantitative, system-level observations such as nitrate fluxes and describe these fluxes using simple models. The model then proposes mechanisms, such as nutrient limitation or shifts in biomass activity, that organize community metabolism. In turn, the physiological, chemical or ecological origins of these proposed mechanisms can be investigated.

For example, in Regime II, our model and amendment experiments demonstrated that carbon limitation gave rise to constant rates of nitrate utilization (Figs. 3 and 4). The constant rate of electron acceptor utilization is likely to reflect bacterial physiology under starvation, where intracellular components are degraded to support maintenance respiration[51,52]. We performed experiments on denitrifying isolates that corroborated this conclusion (Extended Data Fig. 2e,f). This supports previous results linking carbon limitation to nitrate dynamics[15,39,53,54] and highlights how our approach can begin to connect physiological and ecosystem-level processes[26,53,55].

Our analysis revealed conserved phylum-level associations with functional regimes (Fig. 4). Rare Bacillota expanded under strong basic perturbations, whereas dominant Pseudomonadota and Bacteroidota thrived near native pH. These patterns suggest that rare taxa adapted to transient stress and dominant taxa adapted to stable conditions, highlighting the role of fluctuations in maintaining diversity[41]. Understanding these taxonomic patterns requires linking dynamics to physiology. Although recent work has developed quantitative insights into the physiology of balanced growth[56], our understanding of the role of fluctuations in determining microbial traits remains limited[57].

Even in the absence of physiological insights, our approach enables the prediction of nitrate dynamics from sequencing and nutrient measurements. Using abundance of nitrate reductase genes inferred from 16S data[28] and WSOC measurements, we trained regressions to predict the model parameters $\tilde{x}(0)$ and $\gamma\tilde{C}(0)$ (Extended Data Fig. 10a–g). This enabled us to predict nitrate fluxes from pH, WSOC and amplicon data alone, demonstrating that functional regimes can link structure and function even in complex communities[58].

Our study has several limitations. First, soil slurries lack the physical structure and natural environmental fluctuations of intact soils; in particular, in situ nitrate utilization occurs under dynamically changing conditions[24]. Second, natural pH variation is typically small (less than 1 unit)[59,60], meaning that most soils are likely to remain in Regime II, except under large exogenous perturbations such as fertilizer or urine inputs[30]. Third, although our soils span global taxonomic diversity reasonably well, they under-represent highly basic (pH > 8) or strongly acidic (pH < 4) soils (Extended Data Fig. 8m). Highly buffered basic soils may resist pH-induced regime transitions altogether.

This work provides a framework for linking community composition and physiology to ecosystem metabolism, offering a route to understanding how complex microbiomes respond to environmental change.

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

## Methods

### Sample collection, site description and soil characterization

Twenty topsoils were sampled across a range of pH values (4.7–8.32) from the Cook Agronomy Farm (Supplementary Table 1). The CAF (46.78° N, 117.09° W, 800 m above sea level) is a long-term agricultural research site located in Pullman, WA, USA. CAF was established in 1998 as part of the Long-Term Agroecosystem Research (LTAR) network supported by the US Department of Agriculture. Before being converted to an agricultural field, the site was zonal xeric grassland or steppe. CAF operates on a continuous dryland-crop rotation system comprising winter wheat and spring crops. CAF is located in the high rainfall zone of the Pacific Northwest region, and the soil type is classified as Mollisol (Naff, Thatuna and Palouse Series)[61]. Sampling occurred from 8 to 12 September 2022, post-harvest of spring crops, to reduce the impact of the plant on soil microbial communities. This period was during the dry season preceding the concentrated autumn rainfall.

Topsoils were collected from the eastern region of the CAF at a depth of 10–20 cm, other than Soil 1 and Soil 2 (depth of 0–10 cm). Eastern CAF practices 'no tillage', which eliminates soil inversion and mixing of the soil surface to 20 cm. The N fertilizer in this field has been primarily deep banded to depths of approximately 7–10 cm during the time of application, which creates a spike of nitrate resource in the soil depth we sampled. Each soil sample was obtained by cutting down through the hardened dry soil with a spade in a circular motion to create a cylindrical cake of soil of radius 10–20 cm until the desired depth. Each soil sample was not merged from sampling multiple replicates due to differences in pH in different locations. Samples were collected within a diameter of 500 m within the CAF to minimize the variation of edaphic factors other than pH. The large variation in soil pH comes from the long-term use of ammoniacal fertilizers and associated N transformations, which may undergo nitrification, resulting in the release of H ions. In combination, spatial pH variation increases with field-scale hydrologic processes that occur under continuous no tillage superimposed over a landscape that has experienced long-term soil erosion.

To maximize the coverage of sampled native pH, we used a portable pH meter (HI99121, Hanna Instruments) to directly measure and estimate the soil pH without having to make slurry on site to determine whether to sample the soil before sampling. For accurate pH values, pH was measured in the laboratory using a glass electrode in a 1:5 (soil to water w/w) suspension of soil in water (protocol of International organization for standardization (ISO) 10390:2021), where 7 g of soil was vortexed with 35 ml of Milli-Q filtered water, spun down, and filtered with 0.22 μm pore size. With these pH values, we selected 20 topsoil samples that are spread across a range of pH from 4.7–8.32 with intervals of 0.1–0.6. Twenty soil samples were sieved (<2 mm), removed of apparent roots and stones, and gravimetric water content was determined (105 °C, 24 h). The sieved samples were stored in the fridge for 0–3 months until the incubation experiment. For sequencing the initial community, subsamples were stored in −80 °C until the DNA extraction. The twenty soils were sent to the Research Analytical Laboratory (University of Minnesota, USA) to measure soil texture (soil particle analysis; sand, silt and clay composition), total carbon and nitrogen and cation exchange capacity. The soils were also sent to Brookside Laboratories for a standard soil analysis package (Standard Soil with Bray I phosphorus). Twenty soils had relatively similar edaphic properties: 5–9% gravimetric water content (g per g dry soil), soil texture of silty clay loam with 0% sand and 32–43% clay, and C:N ratio of 12–16 with 10–20 total carbon (mgC per g soil) (Supplementary Table 1).

To demonstrate the generality of our results from 20 CAF soils in other soils, we collected topsoil samples from 8 additional sites spanning natural preserves in IL (LaBagh Woods, Orland Grasslands, Moraine), IN (Pinhook Bog, Ambler Flatwoods) and the Sedgwick Reserve in CA (CL, EL and SCL grassland samples, managed by the University of California, Santa Barbara) in August 2024. Owing to the strong pH buffering capacity of the Orland, Moraine and SCL samples, we proceeded with five soils for which pH perturbations were feasible. Using the same methods as in the experiments conducted on the CAF soils (see below), we performed pH perturbation experiments, dynamic measurements of metabolites, and 16S rRNA sequencing. Of these, we analysed the metabolite dynamics of four soils (LaBagh, Pinhook, CLG13 and ELG13), as the Ambler soil had issues with nitrate retrieval under basic conditions during the Griess assay (for example, we could not reliably recover added nitrate using our extraction methods). Details of these soils and their characteristics are provided in Supplementary Table 1.

### Soil rewetting, constructing soil pH titration curves and pH perturbation experiments

To mimic the autumn rainfall in the Pacific Northwest region and minimize the effect of spiking microbial activity by rewetting dry soils[62], we rewetted the sieved soil for 2 weeks before the perturbation experiments at room temperature with sterile Milli-Q water at 40% of each soil's water holding capacity. After resetting, a soil slurry was made by adding 2 mM sodium nitrate solution to the soil (2:1 w/w ratio of water to soil). The slurry was then transferred to 48-deep-well plates (2.35 ml of slurry per well) for incubation under anaerobic conditions (950 rpm, 30 °C) for 4 days. Anaerobic incubation was performed in a vinyl glove box (Coy Laboratory Products 7601-110/220) purged of oxygen with a 99%:1% $N_2$:$CO_2$ gas mixture, where the gaseous oxygen concentration was maintained below 50 ppm to prevent aerobic respiration[58].

To perturb the soil pH to desired levels, we constructed each soil's pH titration curve for the 20 soils with varying native pH to know exactly how much acid or base to add to each soil sample. To do so, separate from the main pH perturbation experiment, we added 23 different levels of HCl (acid) or NaOH (base) in the slurry, final concentrations ranging from 0–100 mM HCl or NaOH. We colorimetrically measured the pH (see section below) immediately after and 4 days after adding each well's designated amount of acid/base. Owing to natural soil's buffering capacity, it takes 1–2 days to stabilize its pH level. Thus, we used the endpoint (after 4 days) pH measurements for all pH perturbations. We did a spline interpolation on the titration data points, plotting endpoint pH (y axis) against acid or base input (x axis), to compute how much HCl and NaOH needs to be added to the soil to obtain 13 different levels of pH with -0.4 intervals ranging from pH 3 to 9, including the pH level without the addition of any acid or base. For Soil 19 and Soil 20, we had only 7 and 3 perturbed pH levels respectively, because the strong buffering capacity of these soils (native pH over 8) limited the range of pH perturbation. We additionally tested whether the anion of acid ($Cl^-$) or the cation of base ($Na^+$) had a distinctive effect on the nitrate reduction dynamics, which was not the case (for results, see Extended Data Fig. 2g and Supplementary Information, 'Effect of base cation on nutrient release in soils').

For the main pH perturbation experiment, the computed levels of concentrated HCl or NaOH were added to the slurry in the 48-deep-well plate with and without chloramphenicol treatment for each perturbed pH level in triplicate. The plates were immediately transferred to the shaking incubator (950 rpm in Fisherbrand Incubating Microplate Shakers 02-217-759, 3 mm orbital radius, 30 °C) inside the anaerobic glove box and incubated for 4 days. For chloramphenicol-treated samples, we added concentrated chloramphenicol solution to the slurry to obtain a final concentration of 1 g $l^{-1}$, after testing different doses to rule out the possibilities of unwanted growth due to chloramphenicol resistance or degradation (Extended Data Fig. 2c and see Supplementary Information, 'Effectiveness of chloramphenicol in preventing microbial growth in the soil'). To gauge the effect of the 2 mM nitrate, we had no-nitrate controls (0 mM nitrate) for both chloramphenicol-treated and untreated treatments in the unperturbed pH conditions. With antifungal cycloheximide controls (200 ppm) for all 20 soils, we confirmed that fungal activity minimally affects nitrate

utilization dynamics (Extended Data Fig. 2a,b). We also confirmed that abiotic nitrate or nitrite reduction does not occur by measuring metabolic dynamics of autoclaved soil (120 °C, 99 min, autoclaved 5 times every 2 days; Extended Data Fig. 2d). To offset the effect of increasing metabolite concentration due to evaporation throughout the 4-day incubation period, we used the wells with just 2 mM nitrate, nitrite and ammonium solutions to correct for evaporation in the slurry samples for every time point. The values of the gravimetric water content of each soil were taken into account to correct for the dilution of 2 mM nitrate due to moisture in the soil. After the incubation, the plates were stored in −80 °C for sequencing endpoint communities.

### Time-series slurry sampling, extraction, and colorimetric assays to measure nitrate, nitrite, ammonium, WSOC and pH

To obtain the metabolic dynamics, we subsampled 60 µl of the slurry into 96-well plates 10 times throughout 4 days (0, 4, 8, 19, 25, 31, 43, 55, 67 and 91 h), where the initial time point ($T_0$) is the time of pH perturbation and the start of anaerobic incubation. To measure nitrate and nitrite dynamics, extracts were prepared from the sampled slurries by adding and vortexing for 2 min with 90 µl of 3.33 M KCl solution (final concentration of 2 M KCl) and centrifuging at 4,000 rpm for 5 min. The supernatant was filtered at 0.22 µm with a vacuum manifold to remove soil particles that could interfere with colorimetric assays. Concentrations of nitrate and nitrite in the extracts were determined colorimetrically using the Griess assay[63] and vanadium (III) chloride reduction method, following the protocol outlined previously[58]. We confirmed that 95%–99% of the nitrate in the soil can be accurately retrieved and detected using this method, as verified by nitrate spike-in and extraction experiments in the soil. For all 20 CAF soils, the ammonium dynamics were measured colorimetrically using the salicylate–hypochlorite assay from the soil extracts[64]. Chloramphenicol treatments in the samples led to consistent detection of 0.5 mM $NH_4^+$ due to its N-H moiety. The salicylate–hypochlorite assay is also affected by the amount of base (NaOH) in the samples, resulting in slightly lower detection of chloramphenicol in the chloramphenicol-treated samples (0.45 mM $NH_4^+$ in 100 mM NaOH perturbations). Taking advantage of these control measurements, we used the constant $NH_4^+$ levels in the controls without 2 mM $NO_3^-$ (no-nitrate controls) in the chloramphenicol-treated samples for each soil to offset the NaOH effect in the non-chloramphenicol samples and subtracted $NH_4^+$ levels caused by chloramphenicol in chloramphenicol-treated samples.

For WSOC measurements, we subsampled 60 µl of the slurry into 96-well plates at $T_0$ and endpoint (4 days). Then, soil extracts were prepared by adding, vortexing with 90 µl Milli-Q water, centrifuging at 4,000 rpm for 5 min, and 0.22 µm filtering the supernatant. Concentrations of the organic carbon in the supernatant were measured colorimetrically by the Walkley–Black assay, which uses dichromate in concentrated sulfuring acid for oxidative digestion[65]. We subtracted 0.4 mgC ml$^{-1}$ from the chloramphenicol-treated samples because chloramphenicol gave rise to a measured value of 0.4 mgWSOC ml$^{-1}$ without additional carbon. For pH measurements, we subsampled 100 µl of the slurry into 96-well plates at $T_0$ and the endpoint. Then, soil extracts were prepared by adding, vortexing with 150 µl KCl solution (final concentration of 1 M KCl), centrifuging at 4,000 rpm for 5 min, and 0.22 µm filtering the supernatant. The pH of the 120 µl supernatant was determined colorimetrically by adding 4 ul of the multiple indicator dye mixture via the protocol described previously[66]. The reason we used 1 M KCl method for pH measurement (ISO 10390:2021) was that, contrary to the KCl method, the H$_2$O method (using water instead of 1M KCl) resulted in a highly yellow colouration of the supernatants in strong basic perturbed samples, which interfered with the wavelength of the colorimetric pH assay. For samples of pH outside the range of the assay (below pH 3 and above pH 9), we used a pH micro-electrode (Orion 8220BNWP, Thermo Scientific). We calculated the endpoint perturbed pH as the average pH of the three biological replicate endpoint samples.

### DNA extraction with internal standards, library preparation and 16S rRNA amplicon sequencing

We performed 16S amplicon sequencing on half of all samples: 10 (soils 3, 5, 6, 9, 11, 12, 14, 15, 16, 17; Supplementary Table 1) out of 20 soils were sequenced before perturbation and at the endpoint in both chloramphenicol-treated and untreated conditions, totalling 1,085 amplicon sequencing measurements. Genomic DNA was extracted from 500 µl aliquots in a combined chemical and mechanical procedure using the DNeasy 96 PowerSoil Pro Kit (Qiagen). Extraction was performed following the manufacturer's protocol, and extracted DNA was stored at −20 °C. To estimate the absolute abundance of bacterial 16S rRNA amplicons, we added known quantities of genomic DNA (gDNA) extracted from *Escherichia coli* K-12 and *Parabacteroides* sp. TM425 (samples sourced from the Duchossois Family Institute Commensal Isolate Library, Chicago, IL, USA) to the slurry subsamples before DNA extraction. Equal concentrations of gDNA from these two strains were added. Both strains have identical rRNA copy numbers of 7 and comparable genome sizes of approximately 5,000 kb. DNA library preparation was performed using the 16S Metagenomic Sequencing Library Preparation protocol with a 2-stage PCR workflow (Illumina). The V3–V4 region was amplified using forward primer 341-b-S-17 (CCTACGGGNGGCWGCAG) and reverse primer 785-a-A-21 (GACTACHVGGGTATCTAATCC)[67]. We confirmed using gel electrophoresis that the negative samples containing all reagents did not show visible bands after PCR amplification. Sequences were obtained on the Illumina MiSeq platform in a 2× 300-bp paired-end run using the MiSeq Reagent Kit v3 (Illumina) with 25% PhiX spike-ins. A standardized 10-strain gDNA mixture (MSA-1000, ATCC) was sequenced as well to serve as a positive control, which was confirmed to have relatively uniform read counts after assigning taxa.

### Model and fitting

**Consumer-resource model.** Consider a consumer-resource model with one consumer variable (functional biomass $x(t)$, biomass) and two resource variables (nitrate $A(t)$ and carbon-nutrient $C(t)$, mM), which evolves in time ($t$, day). The ordinary differential equations of the consumer-resource model can be expressed as:

$$\dot{A}(t) = -r_A x(t)\frac{A(t)}{A(t)+K_A},$$
$$\dot{C}(t) = -r_C x(t)\frac{C(t)}{C(t)+K_C}, \quad\quad (1)$$
$$\dot{x}(t) = \gamma x(t)\frac{A(t)}{A(t)+K_A}\frac{C(t)}{C(t)+K_C}.$$

The first two equations represent the resource consumption rates, which are determined by the functional biomass ($x$ (g biomass)), the maximum consuming rates per unit biomass ($r_A$ and $r_C$ (mM per g biomass per day)), and the Monod functions ($A/(A+K_A)$ and $C/(C+K_C)$ (dimensionless)). Here we assume the affinities ($K_A$ and $K_C$ (mM)) to be fixed and small. So the Monod functions are 1 when $A \gg K_A$ or $C \gg K_C$, and 0 when $A \to 0$ or $C \to 0$. The third equation represents the growth of functional biomass, which is determined by the maximum growth rate per biomass ($\gamma$ (day$^{-1}$)) and the multiplication of two Monod terms indicating the fact that nitrate and carbon are non-substitutable (electron acceptor and donor, respectively). The growth is exponential ($x(t) = x(0)e^{\gamma t}$) when both $A \gg K_A$ or $C \gg K_C$, but growth stops when either $A \to 0$ or $C \to 0$. Therefore, in this model, the growth of biomass is limited by both resources, but the consumption of one resource can continue when the other resource runs out and the biomass growth stops. For example, we believe this happens when $C \to 0$ in Regime II and the consumption of $A$ continues (Fig. 2). For a discussion of models that consider multiple biomasses and carbon sources, see Supplementary Information 'Justifying the effective 1-biomass model despite the diversity of denitrifying taxa'.

**Solution for nitrate dynamics.** To find the solution for nitrate dynamics, we rescale the equations by combining parameters: $\widetilde{x} = r_A x$, $\widetilde{C} = C r_A / r_C$, $\widetilde{K}_C = K_C r_A / r_C$. Therefore, the equations become:

$$\dot{A}(t) = -\widetilde{x}(t) \frac{A(t)}{A(t) + K_A}$$

$$\dot{\widetilde{C}}(t) = -\widetilde{x}(t) \frac{\widetilde{C}(t)}{\widetilde{C}(t) + \widetilde{K}_C} \qquad (2)$$

$$\dot{\widetilde{x}}(t) = \gamma \widetilde{x}(t) \frac{A(t)}{A(t) + K_A} \frac{\widetilde{C}(t)}{\widetilde{C}(t) + \widetilde{K}_C}$$

In the rescaled equations (2), the units of parameters and variables are: $[\widetilde{x}]$ (mM day$^{-1}$) and $[\widetilde{C}] = [\widetilde{K}_C]$ (mM). Therefore, the solution of nitrate dynamics only depends on three parameters ($\gamma$, $K_A$ and $\widetilde{K}_C$) and three initial conditions ($A_0$, $\widetilde{C}(0)$ and $\widetilde{x}(0)$). Since the affinities are very small ($K_A \approx 0.01$ mM, $\widetilde{K}_C \approx 0.01$ mM), the solution of biomass activity approximately equals $\widetilde{x} = \widetilde{x}(0) e^{\gamma t}$ before the time at which growth stops $t^*$ (Fig. 2). So the resource dynamics before $t^*$ are approximately $A = A_0 - \widetilde{x}(0)(e^{\gamma t} - 1)/\gamma$ and $\widetilde{C} = \widetilde{C}(0) - \widetilde{x}(0)(e^{\gamma t} - 1)/\gamma$. Accordingly, the time at which growth stops is given by $t^* = \log(\min(A_0, \widetilde{C}(0))\gamma/\widetilde{x}(0) + 1)/\gamma$. If $\widetilde{C}(0) < A_0$, the nitrate dynamics after $t^*$ and before running out are given by $A = A(t^*) - (\gamma\widetilde{C}(0) + \widetilde{x}(0))(t - t^*)$. As a result, the nitrate consumption rate after $t^*$ is $\gamma\widetilde{C}(0)$ larger than the initial rate $\widetilde{x}(0)$. Therefore, the two key parameters of the model are $\widetilde{x}(0)$ and $\gamma\widetilde{C}(0)$ which are both rates (in mM day$^{-1}$).

**Least-squares fitting scheme.** To infer the model parameters from the metabolite measurement, we use the least-squares fitting scheme to find the closest dynamic curves to the time-series data. Our metabolite measurement including the time points ($\underline{t}^- = [t_1^-, t_2^-, \ldots, t_N^-]$) and nitrate amount ($\underline{a}^- = [a_1^-, a_2^-, \ldots, a_N^-]$) for each CHL- sample, and the measurements of $\underline{t}^+$ and $\underline{a}^+$ for a corresponding chloramphenicol-treated sample. We set up the loss function as the mean-squared error:

$$L = \frac{1}{2N}\left( \sum_{k=1}^{N} (A(t_k^-) - a_k^-)^2 + \sum_{k=1}^{N} (A^c(t_k^+) - a_k^+)^2 \right). \qquad (3)$$

Here, the functions $A(t)$ and $A^c(t)$ are theoretical solutions of the consumer-resource model (2) for non-chloramphenicol and chloramphenicol conditions, respectively. Because the nitrate dynamics $A(t)$ and $A^c(t)$ are determined by the parameter set $\Theta = \{\widetilde{x}(0), \widetilde{C}(0), A_0, A_0^c, \gamma, K_A, \widetilde{K}_C\}$, we minimize the loss function $L(\Theta)$ to get the optimal model parameters $\Theta^*$. We note to the readers that three parameters are fixed ($\gamma = 4.8$ day$^{-1}$, $K_A = \widetilde{K}_C = 0.01$ mM) as justified by the sensitivity analysis in the following paragraph. Note that these parameters were globally fixed across all the data for CAF soils. For other soils (IL, IN, CA), $\gamma$ is fixed within each site but may vary for soils from different sites, the value of which is chosen in the regime where fit quality is insensitive to $\gamma$ (see next section). The optimization algorithm is the interior-point method, which is built in the MATLAB fmincon function. The codes and data are available at the Open Science Framework (https://doi.org/10.17605/OSF.IO/CTF8K). The fitting errors over all samples are shown in Extended Data Fig. 3a,b, in which the root-mean-squared error (RMSE, $\sqrt{L(\Theta^*)}$) and the error per data point ($|A(t_k^-) - a_k^-|$ or $|A^c(t_k^+) - a_k^+|$) are normalized by the input value of nitrate (2 mM).

**Sensitivity analysis on model parameters.** Here we justify the decision to globally fix $\gamma$, $K_A$ and $\widetilde{K}_C$. We analysed the sensitivity of $\gamma$, $K_A$ and $\widetilde{K}_C$ on simulated dynamic data. To reflect the three typical dynamics (regimes) observed from the measurement, we simulated three nitrate curves by setting up the initial conditions to be $\widetilde{x}(0) = 0.01, 0.1$ and 0.001 mM day$^{-1}$ and $\widetilde{C}(0) = 0.005, 0.05$ and 2 mM, respectively. Other parameters are given by $A_0 = A_0^c = 2$ mM, $K_A = \widetilde{K}_C = 0.01$ mM,

$\gamma = 4$ day$^{-1}$. We then used different fixed parameter values to fit the three examples. In the first row of Extended Data Fig. 3c, we used different fixed $\gamma$ values−from $\gamma = 2$ day$^{-1}$ to $\gamma = 6$ day$^{-1}$−to fit three simulations. We demonstrate very small mismatches (RMSE <5%) from these variations of parameter values, which are almost invisible in Regime I and Regime II fittings. In the second and the third row of Extended Data Fig. 3c, we use different fixed $K_A$ and $\widetilde{K}_C$ values−from $10^{-4}$ mM to 1 mM−to fit three simulations. When $K_A < 0.1$ mM or $\widetilde{K}_C < 0.1$ mM, the mismatches were again very small (RMSE <1%) and invisible. These results indicate that the fixed values of $\gamma$, $K_A$ and $\widetilde{K}_C$ are insensitive in large ranges.

**Determination of regime boundary with model parameters.** To define the regime boundaries, we examined the distributions of each parameter's value. $\widetilde{x}(0)$ had a bimodal distribution (Extended Data Fig. 3d). This bi-modality becomes more evident when we separately observe its distribution from the left half (perturbed pH < 4) and right half (perturbed pH > 6) of the parameter space displayed in the perturbed pH versus native pH grid in Fig. 3c (Extended Data Fig. 3e). Therefore, we set the threshold for the $\widetilde{x}(0)$ boundary where these two modes are separated ($\widetilde{x}(0) = 0.05$). The distribution of $\gamma\widetilde{C}(0)$ exhibited a significant mode around 0, prompting us to set the threshold ($\gamma\widetilde{C}(0) = 1.5$) at the tail region, where the $\gamma\widetilde{C}(0)$ threshold also separated the Regime III samples in the top left quadrant of the $\widetilde{x}(0)$ versus $\gamma\widetilde{C}(0)$ scatter plot (Fig. 3a). The separation of Regime I and Regime II data points may not be clear cut in the $\widetilde{x}(0)$ versus $\gamma\widetilde{C}(0)$ scatter plot (Fig. 3a).

### Sequence data analysis

**Sequencing data preprocessing and assigning taxonomy to ASVs with DADA2.** Raw Illumina sequencing reads were stripped of primers, truncated of Phred quality score below 2, trimmed to length 263 for forward reads and 189 for reverse reads (ensuring a 25-nucleotide overlap for most reads), and filtered to a maximum expected error of 4 based on Phred scores; this preprocessing was performed with USEARCH v.11.0[68]. The filtered reads were then processed with DADA2 v.1.18 following the developers' recommended pipeline[69]. In brief, forward and reverse reads were denoised separately, then merged and filtered for chimeras. For greater sensitivity, ASV inference was performed using the DADA2 pseudo-pooling mode, pooling samples by soil. After processing, the sequencing depth of denoised samples was $10^4$–$10^6$ reads per sample. Low-abundance ASVs were dropped (≤10 total reads across all 1,085 samples), retaining 34,696 ASVs for further analysis. Taxonomy was assigned by DADA2 using the SILVA database v.138.1, typically at the genus level, but with species-level attribution recorded in cases of a 100% sequence match. R scripts used for DADA2 sequencing data preprocessing were deposited at the Open Science Framework (https://doi.org/10.17605/OSF.IO/CTF8K).

**Computing absolute abundance with internal standards of each ASV per sample.** As an internal control, we verified that the ASVs corresponding to the two internal standard genera *Escherichia*−*Shigella* and *Parabacteroides* were highly correlated with each other as expected (Pearson correlation ($\rho$) = 0.94). These ASVs were removed from the table and combined into a single reference vector of 'spike-in counts'. The spike-in counts constituted 8.9 ± 8.8% of the total reads in each sample. For downstream analysis, the raw ASV counts in a sample were divided by the spike-in counts of the internal standard per sample to obtain the absolute abundance of the ASV in a sample. Total biomass per sample was obtained by dividing the total raw read counts with the spike-in counts of the sample.

**Differential abundance analysis to identify enriched ASVs.** We conducted differential abundance analysis to statistically determine which ASVs were significantly enriched in the nutrient-limiting regime

(Regime II) or the resurgent growth regime (Regime III), respectively. To do so, we identified enriched ASVs for each perturbed pH condition in each native soil comparing endpoint chloramphenicol-untreated samples with endpoint chloramphenicol-treated samples. For each native soil, we then compiled a list of enriched ASVs by aggregating a union set of enriched ASVs across perturbed conditions that belong to Regime II (or Regime III). To remove ASVs that could be false-positive nitrate reducers, we similarly performed differential abundance analysis to identify ASVs that are enriched in no-nitrate controls (nitrate⁻) by comparing endpoint chloramphenicol-untreated (−Chl, nitrate⁻) samples with endpoint chloramphenicol-treated (+Chl, nitrate⁻) samples. This filtering was done when inferring nitrate reducer biomass (Extended Data Fig. 4e,f) and inferring the Regime III strains (Extended Data Fig. 6c–e). For each native soil, we only had nitrate⁻ controls for the condition without acidic/basic perturbation. We assumed that these enriched ASVs in no-nitrate conditions (NNresponders) without acid/base perturbation would also be false-positive nitrate reducers in other acidic or basic perturbation levels. For each native soil, we filtered out these false-positive NNresponders from the aggregated list of Regime II (or Regime III) enriched ASVs.

To identify the ASVs enriched for each perturbed pH level, it was necessary to determine what change in recorded abundance constitutes a significant change, relative to what might be expected for purely stochastic reasons. The relevant null model would combine sampling and sequencing noise with the stochasticity of ecological dynamics over a four-day incubation, and cannot be derived from first principles. However, since all measurements were performed in triplicate with independent incubations, the relevant null model can be determined empirically. The deviations of replicate–replicate comparisons from 1:1 line were well-described by an effective model combining two independent contributions, a Gaussian noise of fractional magnitude $c_{\mathrm{frac}}$ and a constant Gaussian noise of magnitude $c_0$ reads, such that repeated measurements (over biological replicates) of an ASV with mean abundance $n$ counts are approximately Gaussian-distributed with a standard deviation of $\sigma(c_0, c_{\mathrm{frac}}) = \sqrt{(c_{\mathrm{frac}}n)^2 + c_0^2}$ counts. In this expression, $c_{\mathrm{frac}}$ was estimated from moderate-abundance ASVs (>50 counts) for which the other noise term is negligible; and $c_0$ was then determined as the value for which 67% of replicate–replicate comparisons are within $\pm\sigma(c_0, c_{\mathrm{frac}})$ of each other, as expected for 1-sigma deviations. This noise model was inferred separately for each soil and each perturbed pH level, as the corresponding samples were processed independently in different sequencing runs. For example, the parameters in Soil 11 were $c_{\mathrm{frac}} = 0.21 \pm 0.04$ and $c_0 = 4.5 \pm 0.7$ counts (Extended Data Fig. 9i).

The model was used to compute the $z$-scores for the enrichments of absolute ASV abundances in non-chloramphenicol treatments against chloramphenicol-treated controls (three independent $z$-scores from three replicate pairs; rep1–rep1, rep2–rep2 and rep3–rep3). The median $z$-score was assigned to each ASV for each perturbed condition. In consideration of ASVs with 0 read count in samples with or without chloramnphenicol treatment, all raw ASV counts were augmented by a pseudocount of 0.5 and divided by the per-sample spike-in counts, yielding values that can be interpreted as the absolute biomass of each taxon (up to a factor corresponding to the copy number of the 16S operon), measured in units where 1 means as many 16S fragments as the number of DNA molecules in the spike-in. Significantly enriched ASVs were identified in each perturbed condition as those with $z$-scores greater than $z = \Phi^{-1}(1 - \alpha/2n_{\mathrm{ASV}})$, where $\Phi^{-1}(x)$ is the inverse cumulative distribution function of the standard normal distribution, $\alpha = 0.05$, and $n_{\mathrm{ASV}}$ is the number of non-zero ASVs in a given sample. This critical $z$-score ($z = 4.2$, when $n_{\mathrm{ASV}} = 2,000$ for enriched ASVs and $z = 4.3$, when $n_{\mathrm{ASV}} = 2,500$ for filtering no-nitrate responders (NNresponders)) corresponds to a two-tailed Bonferroni-corrected hypothesis test at significance level $\alpha$ under the null hypothesis that counts in the chloramphenicol-treated and untreated conditions are drawn from

the same distribution. These analyses were performed using custom MATLAB (Mathworks) and R scripts, which are deposited at the Open Science Framework (https://doi.org/10.17605/OSF.IO/CTF8K); for additional technical details, the reader is referred to the detailed comments in these scripts.

**NMF analysis on phylum-level growth folds.** To analyse the abundance change at the phylum level, we compute the growth fold of each phylum at each condition. For each phylum, we compute the absolute abundance by aggregating the abundances of all ASVs within that phylum. Taking chloramphenicol-treated abundance (Abs⁺) as the reference abundance and non-chloramphenicol abundance (Abs⁻) as the endpoint abundance (where Abs denotes taxon abundance normalized to internal standard), the logarithm of the growth fold for phylum $i$ and condition $j$ is given by $g_{ij} = \log(\mathrm{Abs}_{ij}^- + 10^{-3}) - \log(\mathrm{Abs}_{ij}^+ + 10^{-3})$. Note that we use chloramphenicol-treated abundance as reference instead of the initial abundance (at $T_0$), to account for any effects on read counts unrelated to growth which would be common between chloramphenicol-treated and untreated conditions (for example, direct effect of acid/base addition), allowing us to focus only on growth-mediated abundance changes. We also set all negative $g_{ij}$ to 0 since we are focusing on growth. For all 130 conditions (10 soils × 13 perturbations) and 40 phyla, the phylum-level growth folds $G$ is a 40 × 130 matrix. For each phylum, the row vector $\vec{g}_i$ represents how it grows under different conditions (see Extended Data Fig. 5b for the growth vectors of the first 10 phyla). In order to reduce the dimensionality of the growth matrix and extract the main features of the growth vectors, we use NMF to decompose the growth matrix $G = W \times H$ into 2 factors, which retain 93.36% of the original $G$ matrix variation. Here, the feature matrix $H$ is of size 2 × 130, and the 2 rows $\vec{h}_1$ and $\vec{h}_2$ are 2 modes of growth vectors (shown in Extended Data Fig. 5c). Therefore, the growth vector of phylum $i$ is thus decomposed as $\vec{g}_i \approx w_{i1}\vec{h}_1 + w_{i2}\vec{h}_2$, while the weights $w_{i1}$ and $w_{i2}$ are from the 40 × 2 weight matrix $W$. The weights of all 40 phyla are plotted in Extended Data Fig. 5c, showing that Bacillota is mostly composed of the second mode $\vec{h}_2$ and other phyla are mostly composed of the first mode $\vec{h}_1$. Additionally, Bacteroidota and Pseudomonadota show the most significance in the first mode.

**Genotyping enriched ASVs with PICRUSt2.** To understand what traits make resurgent growth strains unique, we used PICRUSt2 v.2.5.2[28] to infer putative genotypes of the enriched ASVs in the resurgent growth regime (Regime III) (Extended Data Fig. 6g). Using the script place_seqs.py from PICRUSt2, we matched the representative 16S rRNA sequences of each ASV to PICRUSt2's curated reference genome database (multiple sequence alignment). Then, using the hsp.py script from PICRUSt2 with default parameters, we predicted KEGG orthologues (KO) abundance of each ASV with the matched reference genome (hidden-state prediction). To narrow down to KOs or genes related to denitrification and DNRA, we focused on nitrate reductase in denitrification (*narG*/K00370, *narH*/K00371, *narI*/K00374, *napA*/K02567 and *napB*/K02567) and nitrite reductase to ammonium (*nirB*/K00362, *nirD*/K00363, *nrfA*/K03385 and *nrfH*/K15876). To track which KOs were enriched at which pH in the 89 families used in the peak pH analysis (see Supplementary Information, 'Determination of peak pH for each family'), we summed the relative abundance (reads/total reads of each perturbed pH level in non-chloramphenicol condition) of the ASVs belonging to each family that possessed at least 1 predicted gene respectively for *narGHI*, *napAB*, *nirBD* and *nrfAH*. Then, we plotted their relative abundance values across pH for all soils, indicated by the intensity of the point's colours (Extended Data Fig. 6g).

**Taxonomy of growing strains in Regime III varies with soil native pH.** To further investigate whether the taxonomic identity of resurgent growth (Regime III) strains varies across natural pH environments, we performed a regularized regression analysis to see if we can predict

the native pH level of the source soil from the presence or absence of taxa that grow in Regime III at the ASV, species, genus, family, or higher taxonomic levels. The resurgent growth strains were determined by the differential abundance analysis as described previously. We used the sequencing data to build a matrix where the rows are samples (including three biological replicates) belonging to the resurgent growth regime (Regime III), where each row has a corresponding native pH value of the original soil. There are ten source soils with different native pH levels, and each soil has 3–6 pH-perturbed samples (replicates) of which metabolite dynamics are classified as Regime III. The matrix's columns are different taxa belonging to the identified Resurgent growth strains, either in ASV, species, genus, family or higher levels. Each element of the matrix is 0 if absent and 1 if present in the sequencing data of the sample. Because the presence and absence of taxa can randomly depend on the random sampling depth of each sample, we test varying threshold values (0, 0.001 or 0.005 relative abundance) to call the taxa present if their relative abundance is greater than the threshold (effects shown in Extended Data Fig. 8k).

The regularized regression was performed to predict the native pH of the source soil of the samples from the presence and absence of taxa using only additive terms and LASSO regularization to avoid overfitting[70]. To estimate the regularization hyperparameter, tenfold cross-validation was performed on the samples from ten different soils with different native pH levels. All models were fit using the package glmnet in R v.4.1.4. To make predictions of the native pH, we used two strategies. First, 'in-sample' predictions used all available data points to fit the regression coefficients and predicted native pH using those coefficients. Second, to ask whether we can still predict the native pH without the model seeing the samples belonging to that specific native pH level, we implemented a leave-one-soil-out (LOSO) procedure, where all the perturbed samples from one native soil were left out as a test set, and the model was trained on the remaining data to fit the regression coefficients. Then, we used the model to predict the native pH of the left-out samples (out-of-sample prediction). The observed versus predicted pH values are shown in the scatter plots (Extended Data Fig. 8h). The prediction quality ($R^2$) was computed using the mean predicted and mean observed native pH levels for each soil (for different taxonomic levels and prediction strategies, see Extended Data Fig. 8j,k; negative $R^2$ values indicate the predictions are worse than just predicting the pH as the mean predicted pH). To ascertain that our high prediction quality was not a random artefact, we randomly permuted the native pH values of our soils 1,000 times and then predicted in-sample the native pH to obtain 1,000 $R^2$ values under permutation and used this distribution to compute a $P$ value (0.012) for the observed $R^2$ obtained with our true native pH values (black arrow, Extended Data Fig. 8l).

**Testing the effect of different bases and salts on nutrient release**
To see the effects of different bases (NaOH and KOH) on nitrate reduction dynamics, we added different concentrations of NaOH and KOH (final concentration of 0, 8, 16 and 24 mM in the slurry), following the same protocol previously described (without chloramphenicol), to measure the nitrate and nitrite dynamics (Extended Data Fig. 2g, using Soil 6; Supplementary Table 1). In addition, to test the effects of $Na^+$, $K^+$ and $Cl^-$ separately, we added different concentrations of salts (NaCl and KCl) (without chloramphenicol and without adding any acid or base) and measured the metabolite dynamics (Extended Data Fig. 2g).

**Nutrient amendment experiments with slurries**
To experimentally determine what nutrient was limiting growth in the nutrient-limiting regime, we conducted nutrient amendment experiments respectively with glucose, succinate, sodium acetate, ammonium chloride ($NH_4Cl$), monosodium phosphate ($NaH_2PO_4$) and sodium sulfate ($Na_2SO_4$) (for results, see Fig. 5d and Extended Data Fig. 2h). Among them, succinate ($pK_a$ = 4.21 and 5.64, 25 °C), acetate ($pK_a$ = 4.76, 25 °C), and phosphate ($pK_a$ = 2.2, 7.2 and 12.4, 25 °C) were strong candidates for

the limiting nutrient according to our soil nutrient release hypothesis, due to their anionic nature in mid-range pH (5–7). The incubation was conducted following the same protocol using Soil 6 (Supplementary Table 1) without chloramphenicol and without adding any acid/base. Concentrations were either in mM C, mM N, mM S or mM P with final concentrations in slurry varying from 0 to 5 mM, each in triplicate. Because we have previously tested the effect of $Na^+$ and $Cl^-$ to be negligible in nitrate dynamics, the effect of these amendments can be attributed solely to C, N, S or P nutrients other than $Na^+$ and $Cl^-$.

**Reporting summary**
Further information on research design is available in the Nature Portfolio Reporting Summary linked to this article.

## Data availability
Raw sequence reads associated with this manuscript are deposited under NCBI BioProject ID PRJNA1205727. Data tables are available on the Open Science Framework (https://doi.org/10.17605/OSF.IO/CTF8K). Source data are provided with this paper.

## Code availability
R and Matlab codes associated with this manuscript are deposited at the Open Science Framework (https://doi.org/10.17605/OSF.IO/CTF8K).

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

**Acknowledgements** The authors thank I. Leslie and D. Huggins for coordinating the soil sampling in the Cook Agronomy Farm; M. L. Friesen and J. M. Parks for providing laboratory space and facilities; T. Paulitz, I. Leslie and D. Huggins for advice regarding soils; F. M. Abasiyanik and H.-N. Shim in the Single Cell Immunophenotyping Core and C. DeValk in the Ranganathan laboratory for their guidance on operating the sequencer; K. Pajdzik and the C. He laboratory for the use of their quantitative PCR machine; A. Flamholz, members of the Kuehn laboratory, Mani laboratory, J. Thornton laboratory and T. Hwa laboratory for discussions; and M. Chakraverti-Wuerthwein for processing the global topsoil microbiome data. This research was a contribution from the Long-Term Agroecosystem Research (LTAR) network. LTAR is supported by the US Department of Agriculture. We acknowledge the Duchossois Family Institute at the University of Chicago for providing *Parabacteroides* sp. TM425 strain to use as internal standard during sequencing. This work was supported by the National Science Foundation Division of Emerging Frontiers EF 2025293 (S.K.) and EF 2025521 (M.M.) and by National Science Foundation PHY 2310746 (M.T.). S.K. acknowledges the National Institute of General Medical Sciences R01GM151538, and support from the National Science Foundation through the Center for Living Systems (grant 2317138). S.K. and M.T. acknowledge CAREER awards from the National Science Foundation (BIO/MCB 2340416 and PHY-2340791). S.K. and M.M. acknowledge financial support from the National Institute for Mathematics and Theory in Biology (Simons Foundation award MP-TMPS-00005320 and National Science Foundation award DMS-2235451). M.M. was supported by The National Science Foundation-Simons Center for Quantitative Biology at Northwestern University and the Simons Foundation grant 597491. M.M. is a Simons Investigator. This project has been made possible in part by grant DAF2023-329587 from the Chan Zuckerberg Initiative DAF, an advised fund of the Silicon Valley Community Foundation. Any opinions, findings, conclusions, or recommendations expressed in this material are those of the authors and do not necessarily reflect the views of the National Science Foundation.

**Author contributions** K.K.L., M.M. and S.K. conceptualized the research. K.K.L. and S.K. designed the experiments. K.K.L. conducted field soil sampling in the Cook Agronomy Farm under the supervision of D.R.H. K.K.L. performed soil processing, characterization, soil pH perturbation experiments, metabolite assays, gDNA extraction and sequencing, advised by S.K. J.W. performed strain isolation from soils, screening phenotypes and identification. K.K.L., S.L. and M.T. performed statistical analysis of the metabolite dynamics and sequencing data, advised by M.T., M.M. and S.K. S.L. performed consumer-resource model fits and simulations on the metabolite dynamics, advised by M.M. and S.K. K.C. conducted field soil sampling at the Sedgwick Preserves and performed the monoculture experiments to recapitulate linear nitrate dynamics. K.K.L. performed the nutrient amendment experiments in soils. K.K.L. and S.K. wrote the manuscript with input from S.L., M.M. and M.T.

**Competing interests** The authors declare no competing interests.

**Additional information**
**Correspondence and requests for materials** should be addressed to Mikhail Tikhonov, Madhav Mani or Seppe Kuehn.

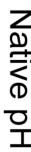

# pH perturbations (20 Soil x 13 incremental pH levels, CHL- vs. CHL+)

**[NO₃⁻] [mM]** ( CHL- vs. CHL+ )

$[NO_3^-]$ [mM] ( CHL- vs. CHL+ )

Native pH

Time (hr)

**Extended Data Fig. 1 | Nitrate dynamics of chloramphenicol untreated (CHL−) and treated (CHL+) conditions in the dataset.** Time series measurements of nitrate in chloramphenicol-untreated (CHL−, black points) and treated (CHL+, red points) across 4 days are shown. Each row is from the identical soil sample of a native pH level ($pH_{H_2O}$), indicated at the right end of each row in the order of most acidic (top) to most basic (bottom). Each row has 13 columns, which are the 13 different levels of short-term pH perturbations. The targeted perturbed pH levels were determined by constructing a soil pH titration curve before the experiment and computing how much acid (HCl) or base (NaOH) to add to the slurries. Perturbed pH levels are indicated inside each panel, which are obtained by measuring the stabilized pH values at the endpoint of the experiment (see Methods). Each line connects the points of measurements of a replicate, constituting the 3 replicates per perturbed condition. The pink-colored box for each row indicates the condition without any acid/base addition, where the pH of these conditions also changes with incubation. The total number of independent biological replicates in the slurry experiment is $n = 1,704$, including no-nitrate ($n = 120$) and cycloheximide controls ($n = 120$). Soil 19 and Soil 20 are not shown due to having different numbers of perturbed pH levels (7 and 3, respectively). CHL+/− slurry experiments ($n = 1,404$) from Soil 1–18 (18 soils × 13 perturbations × 2 CHL+/− × 3 biological replicates) are shown, totaling 14,011 nitrate time-series data points after excluding 29 data points due to experimental errors.

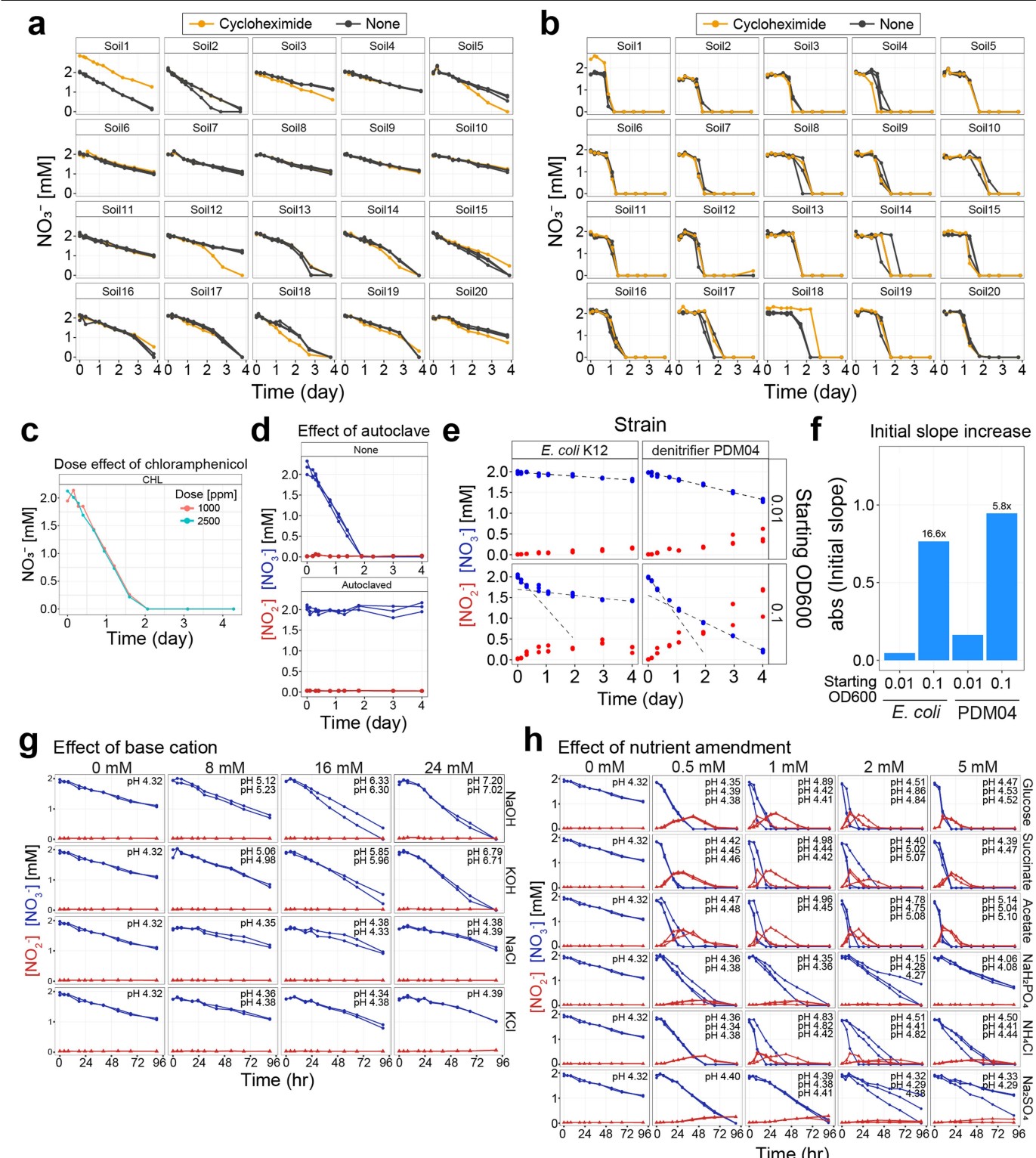

**Extended Data Fig. 2** | See next page for caption.

**Extended Data Fig. 2 | Sanity checks for slurry experiments and monoculture experiment without external carbon source. (a)** Nitrate dynamics without (black, $n = 3$ biological replicates) and with 200 ppm of antifungal cycloheximide (orange, $n = 1$ replicate) treatment for CAF soils ($n = 20$ soils, Table S1). **(b)** illustrates the same experiment but with basic perturbation (<100 mM NaOH). Most of the dynamics were not affected by the application of cycloheximide, indicating that fungi do not play a significant role in nitrate reduction. **(c)** Comparing the nitrate dynamics under 1000 ppm and 2500 ppm ($n = 1$ replicate) of chloramphenicol (CHL) in LaBagh soil (Table S1). If CHL degradation or resistance were an issue, one would expect a dose effect, where lower/higher levels of CHL result in different nitrate reduction dynamics. **(d)** Nitrate (blue) and nitrite (red) dynamics with (bottom, Autoclaved) and without (top, None) autoclaving procedure in LaBagh soil ($n = 3$ biological replicates for each condition). Nitrate reduction was absent in autoclaved soil, confirming it was not abiotic (chemical) nitrate reduction. **(e)** Linear nitrate dynamics in soils can be recapitulated from monoculture experiments without carbon. We measured nitrate (blue) and nitrite (red) dynamics of monoculture experiments using *E. coli* K12 and the denitrifier *Pseudomonas sp*. PDM04 strains over 4 days with no external carbon provided in the culture media. We varied the starting OD600 (optical density at 600nm) to be 0.01 (top) and 0.1 (bottom) ($n = 3$ biological replicates for each condition). The linear dynamics demonstrate that nitrate reduction can occur in the absence of external carbon. The dashed lines represent linear regression of the dynamics: for the top panels, we used all data points, while in the bottom panels, we used the first three points to fit the initial slope and the last four points for late slopes. **(f)** The bars indicate the initial slopes from using different starting OD600 values in two strains, suggesting the effect of starting biomass on initial nitrate reduction rates. The factor increase of initial slopes is annotated on top of the bars. **(g)** To see the effects of different bases (NaOH and KOH) on nitrate dynamics, we added different concentrations of NaOH and KOH (final concentration of 0, 8, 16, 24mM in the slurry) without CHL using Soil 6 (Table S1). To test the effects of $Na^+$, $K^+$, and $Cl^-$ separately, we added different concentrations of salts (NaCl, KCl) (without CHL and without adding any acid/base) and measured the nitrate (blue) and nitrite (red) dynamics ($n = 2$ biological replicates for each condition). Endpoint pH values (1M KCl method) are noted in each panel, and noted once if replicates have identical pH. **(h)** To experimentally determine what nutrient was limiting growth in the Nutrient-limiting regime (Regime II), we conducted nutrient amendment experiments using Soil 6 varying respectively the concentrations of glucose, succinate, sodium acetate, ammonium chloride ($NH_4Cl$), monosodium phosphate ($NaH_2PO_4$), and sodium sulfate ($Na_2SO_4$) ($n = 3$ biological replicates for each condition). 0 mM amendment condition is the same for all nutrients, repeated in the far-left column for each row ($n = 2$ replicates). We observed a transition from linear dynamics to exponential depletion of nitrate, when we amended the soil with a carbon source.

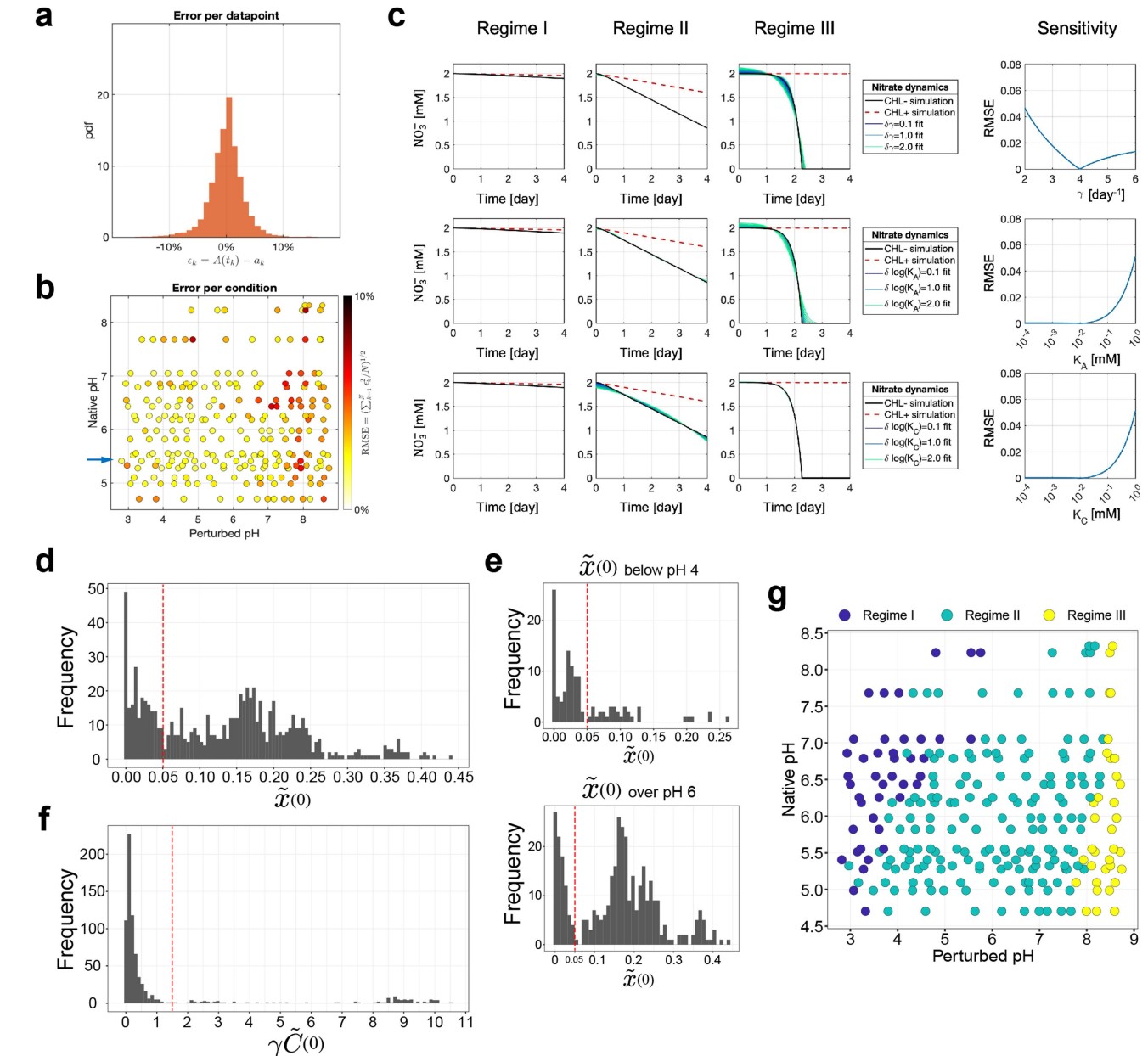

**Extended Data Fig. 3** | See next page for caption.

**Extended Data Fig. 3 | Model fitting error, sensitivity analysis, and determining regime boundary. (a)** Quantification of error in model fitting. The probability density function represents the distribution of errors for individual data points of nitrate measurements at time point $k$ ($n = 14,611$ data points). Errors are calculated as the difference between the model's predicted nitrate concentration $A(t_k)$ and the observed nitrate amounts $a_k$ for either the chloramphenicol-untreated(CHL−) or treated(CHL+) conditions, normalized by dividing by the input nitrate concentration (2 mM) to be expressed as a percentage. **(b)** Each dot represents the error for a specific experimental condition (triplicate), with the native pH of the sample on the y-axis and the perturbed pH on the x-axis ($n = 244$ conditions). The error per condition, indicated by the color of each point, is the square root of the mean-squared error loss function minimized during parameter optimization of both CHL−/+ conditions in triplicate, normalized by the input nitrate concentration (2 mM) to be expressed as a percentage (refer to Methods for the error computation). The arrow indicates the row of the soil that is shown as an example for the bottom panel. **(c)** To justify fixing parameters $\gamma$, $K_A$, and $\widetilde{K}_C$ for model fitting, we analyzed the sensitivity of $\gamma$, $K_A$, and $\widetilde{K}_C$ by simulating dynamic data. To reflect the three typical dynamics (regimes) observed from the measurement, we simulated three nitrate curves by setting up the initial conditions to be $\tilde{x}(0) = 0.01, 0.1, 0.001$ mM/$day$ and $\widetilde{C}(0) = 0.005, 0.05, 2$ mM, respectively. Other parameters are given by $A_0 = A_0^c = 2$ mM, $K_A = \widetilde{K}_C = 0.01$ mM, $\gamma = 4\, day^{-1}$. Black and red dashed lines represent simulated CHL− and CHL+ conditions,

respectively. In the first row, we used different fixed $\gamma$ values - from $\gamma = 2\, day^{-1}$ to $\gamma = 6\, day^{-1}$ - to fit three simulations. We demonstrate very small mismatches (square root of the mean-squared error (RMSE) < 5%, see Methods for loss function) from these variations of parameter values. Purple lines indicate fitted results from $\gamma = 4 \pm 0.1\, day^{-1}$, blue from $\gamma = 4 \pm 1\, day^{-1}$, green from $\gamma = 4 \pm 2\, day^{-1}$). In the second and the third row, we used different fixed $K_A$ and $\widetilde{K}_C$ values - from $10^{-4}$ mM to 1 mM - to fit three simulations. When $K_A < 0.1$ mM or $\widetilde{K}_C < 0.1$ mM, the mismatches were again very small (RMSE < 1%) and invisible, indicating that the fixed values of $\gamma$, $K_A$, and $\widetilde{K}_C$ are insensitive in large ranges. **(d-g)** To determine regime boundary thresholds with distributions of the parameters $\tilde{x}(0)$ and $\gamma\widetilde{C}(0)$, we examined the distributions of parameters fitted to the functional data ($n = 732$ nitrate dynamics). **(d)** $\tilde{x}(0)$ had a bimodal frequency distribution. **(e)** This bi-modality becomes more evident when we separately observe its distribution from the left half (perturbed pH < 4) and right half (perturbed pH > 6) of the parameter space displayed in the perturbed pH vs. native pH grid in Fig. 3c. We set the threshold for the $\tilde{x}(0)$ boundary where these two modes are separated ($\tilde{x}(0) = 0.05$). **(f)** $\gamma\widetilde{C}(0)$ showed an uni-modal frequency distribution. We set the threshold ($\gamma\widetilde{C}(0) = 1.5$) at the tail of the distribution, where the $\gamma\widetilde{C}(0)$ threshold also separated the Regime III samples in the top-left quadrant of the $\tilde{x}(0)$ vs. $\gamma\widetilde{C}(0)$ scatter plot (Fig. 3a). **(g)** With these thresholds of two parameters, we can define the three different functional regimes across native pH and perturbed conditions ($n = 244$ conditions).

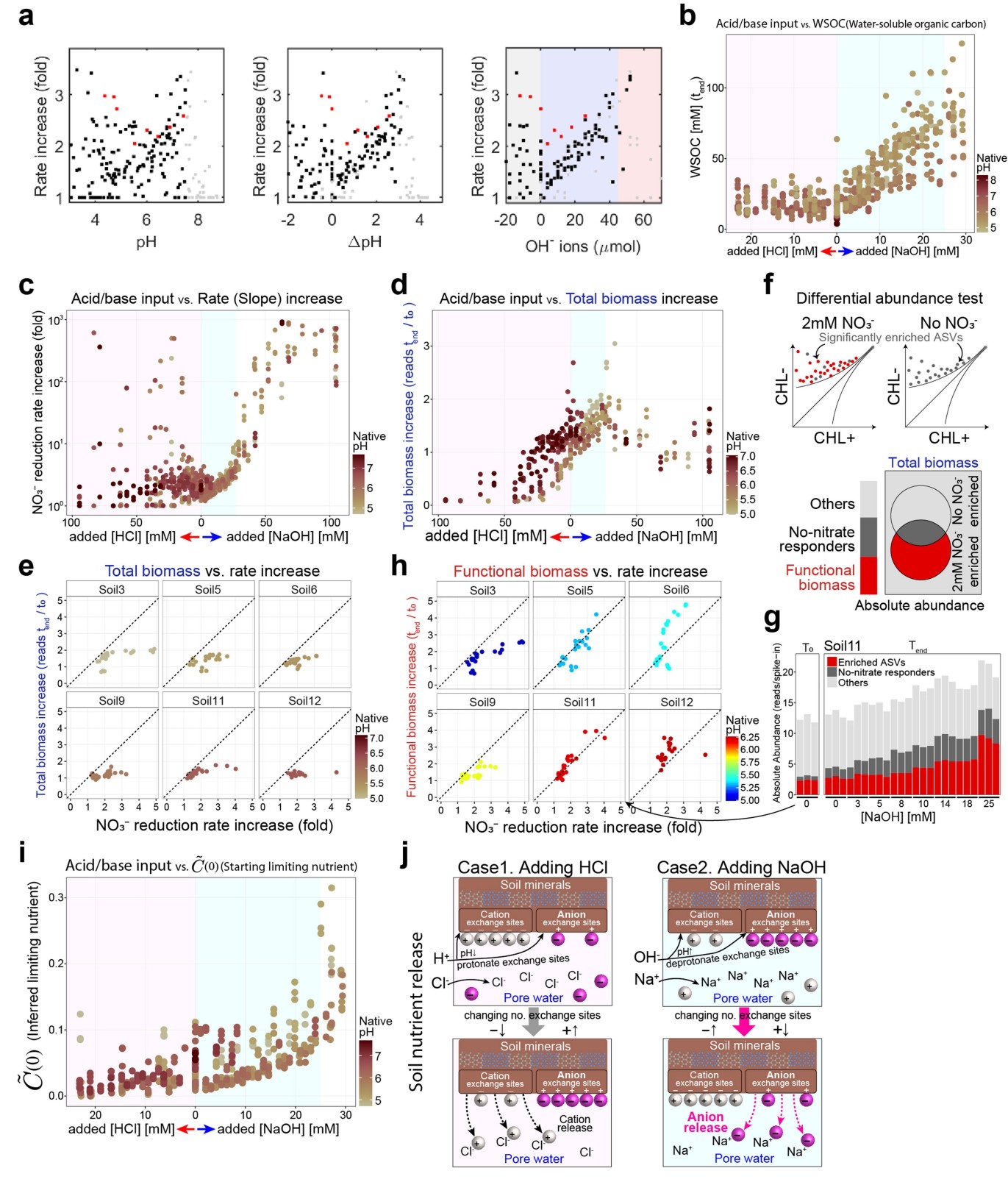

**Extended Data Fig. 4** | See next page for caption.

**Extended Data Fig. 4 | Mechanism of Nutrient-limiting regime (Regime II).**
**(a)** NaOH input shows a more consistent linear relationship with nitrate reduction rate increase than pH or $\Delta$pH. To confirm this, we independently calculated the rate increase (y-axis) by determining the slope ratio (CHL−/CHL+) from linear nitrate dynamics under chloramphenicol-treated (CHL+) and untreated (CHL-) conditions ($n = 3$ biological replicates per condition). Rate increases were plotted against perturbed pH, $\Delta$pH (perturbed pH - native pH), and OH− input (in $\mu$moles, with negative values for H+). Progressing from left to right plots, the data collapse into a stronger linear relationship with OH− input, confirming NaOH as the most reliable descriptor for nutrient-driven growth across soils with varying native pH levels. Soil 12 points (red) deviate from the general trend (black), while pH perturbations of pH > 7.5 (grey) align with the Resurgent growth regime (Regime III), where linearity is not expected. The blue background in the rightmost plot highlights perturbations within the Nutrient-limiting regime (Regime II). **(b)** Adding NaOH linearly increased the water-soluble organic carbon (WSOC) concentrations present in the slurry at the endpoint, while adding HCl did not ($n = 495$ endpoint samples of < 30 mM OH−, < 25 mM H+). The result suggests most water-soluble organic carbon (WSOC) may be negatively charged (anion). Native soil pH is indicated by color. **(c-h)** A more detailed analysis, accounting for individual Amplicon sequence variants (ASVs) that responded to the amendment of nitrate, further confirmed the linear dependence between functional biomass and acid/base added. **(c)** Acid/base input (x-axis) vs. $NO_3^-$ reduction rate increase (y-axis) for CHL− relative to CHL+ conditions across soils with varying native pH ($n = 692$ samples; 38 data points with fold-change > 1000 were excluded for visualization). The rate increase, derived from model parameters $(1 + \gamma \widetilde{C}_0 / \overline{x}_0)$, shows a linear relationship within 0–25 mM NaOH (Regime II, blue background) but not for HCl addition or NaOH > 25 mM (pink background). **(d)** Acid/base input vs. total

biomass growth (fold), calculated as $Tend/T_0$ ($n = 390$ sequencing samples). Biomass increases with NaOH and decreases with HCl within the same range, although total biomass growth need not reflect growth driven solely by nitrate reduction. **(e)** Total biomass fold increase vs. nitrate reduction fold increase shows total biomass growth is less than the nitrate reduction increase (below 1:1 line), indicating not all biomass reduces $NO_3^-$ ($n = 129$ samples of 0–25 mM OH−). **(f)** Functional biomass determination: Differential abundance analysis compares CHL- to CHL+ (baseline). ASVs enriched without nitrate (dark grey) are excluded to identify true nitrate reducers (red). Summing the abundance of these ASVs gives functional biomass per condition. **(g)** Example for Soil 11 showing functional biomass distribution (red, dark grey, light grey bars in $n = 3$ biological replicates). **(h)** Increase fold of functional biomass (end/start) vs. nitrate reduction rate fold increase (CHL−/CHL+) aligned roughly with a 1:1 line showing equal changes in functional biomass and nitrate reduction ($n = 129$ samples of 0–25 mM H+). **(i)** Change in inferred $\widetilde{C}(0)$ with acid/base addition ($n = 498$ samples of < 30 mM OH−, < 25 mM H+): In acidic soils, increasing NaOH raises $\widetilde{C}(0)$ (light blue region), while in neutral soils, increasing HCl lowers $\widetilde{C}(0)$ (light pink region). The proposed mechanism of soil nutrient release is illustrated in the cartoon **(j)**. When adding HCl (Case 1), H+ protonates exchange sites, increasing the number of anion exchange sites and decreasing the number of cation exchange sites. This promotes anion adsorption and cation release into pore water, where cations are stabilized by the added Cl− in the pore water (pink region). When adding NaOH (Case 2) OH− deprotonates exchange sites, increasing the cation exchange sites and decreasing the anion exchange sites. This promotes cation adsorption and anion release into the pore water, where anions are stabilized by Na+, facilitating microbial access to anions in the pore water (blue region).

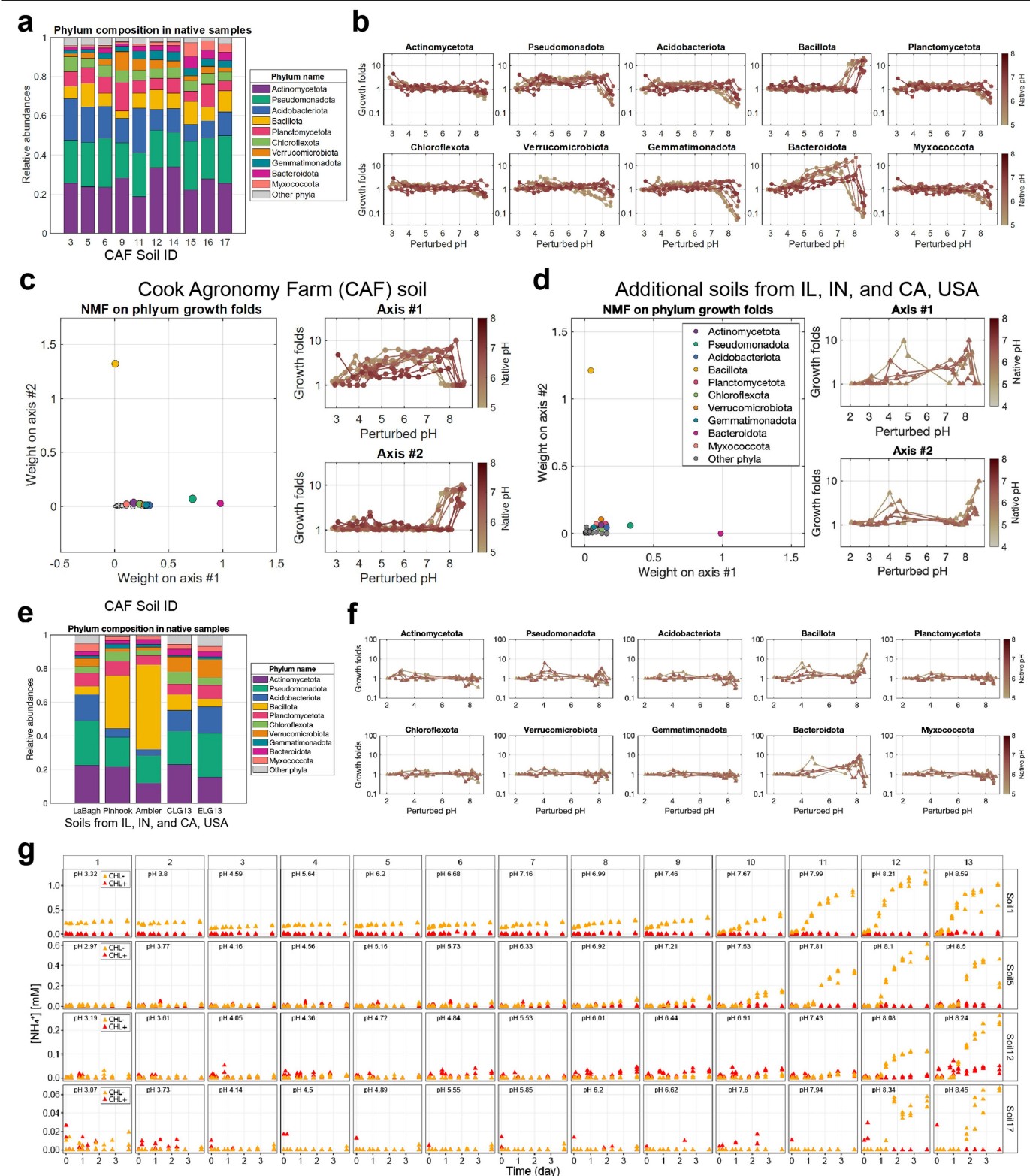

**Extended Data Fig. 5** | See next page for caption.

**Extended Data Fig. 5 | NMF (Non-negative matrix factorization) reveals low-dimensional shifts in growth at the phylum level. (a)** Initial community composition ($T_0$) of CAF soils at the phylum level. The x-axis indicates soils with different native pH levels ($n = 10$ soils: Soil 3, 5, 6, 9, 11, 12, 14, 15, 16, 17, see Table S1 for their properties). The y-axis represents the relative abundance (summed to 1) of the top 10 phyla out of 40, with the cumulative abundance of the remaining phyla depicted in grey as 'Other phyla'. **(b)** For each soil, at each pH perturbed condition ($n = 130$ conditions, 10 soils × 13 perturbations) we computed the growth fold at the phylum level by summing absolute abundance of all ASVs within that phylum in both CHL+ and CHL- and computing $Abs_{CHL-}/Abs_{CHL+}$ (y-axis). Each panel shows growth folds for one phylum as labeled in the title. Each line corresponds to a soil with color indicating native pH (color bar to right). y-axis values above 1 indicate growth in that phylum at that pH. **(c)** To systematically identify the underlying lower-dimensional growth response to pH, we used non-negative matrix factorization (NMF) on the growth fold values to decompose the growth response of all phyla into two modes (Axis #1 and Axis #2, see Methods). The growth folds of each phylum are the linear combination of two modes whose weights are plotted on the left panel (points are colored by phylum as in **(a)**). Mode #2 is composed of the Bacillota phylum (Regime III). Mode #1 is composed of Pseudomonadota and Bacteroidota (Regime II). **(d-f)** The same analysis was performed with growth fold values of phyla in the pH perturbation experiments on the additional soils from IL, IN, and CA, USA (Table S1) ($n = 55$ conditions, 5 soils × 11 perturbations). **(g)** Ammonium dynamics in CHL− and CHL+ samples were measured for 20 CAF soils using the Salicylate-hypochlorite assay[64], plots only showing data for Soils 1, 5, 12, and 17 out of 20 CAF soils. $NH_4^+$ accumulation (3–50% of 2 mM $NO_3^-$) in Regime III suggests activation of the dissimilatory nitrate to ammonia (DNRA) pathway by Regime III strains. The NaOH concentration in perturbed samples also impacted $NH_4^+$ measurements, because the Salicylate-hypochlorite assay includes a step where $OCl^-$ reacts with the N-H moiety, resulting in N-Cl and $OH^-$. Higher NaOH addition results in slightly lower detection of chloramphenicol in the CHL+ samples. We used the constant $NH_4^+$ levels in the controls without 2 mM $NO_3^-$ (No-Nitrate controls) in the CHL+ conditions for each soil to offset the NaOH effect in the CHL- samples (by computing the conversion factor ratio of $NH_4^+$ levels of No-Nitrate controls in CHL+ conditions to the initial $NH_4^+$ levels of each condition with different NaOH additions in CHL+ samples) and subtracted $NH_4^+$ levels caused by chloramphenicol in CHL+ samples. pH in each panel indicates the endpoint pH of the experiment.

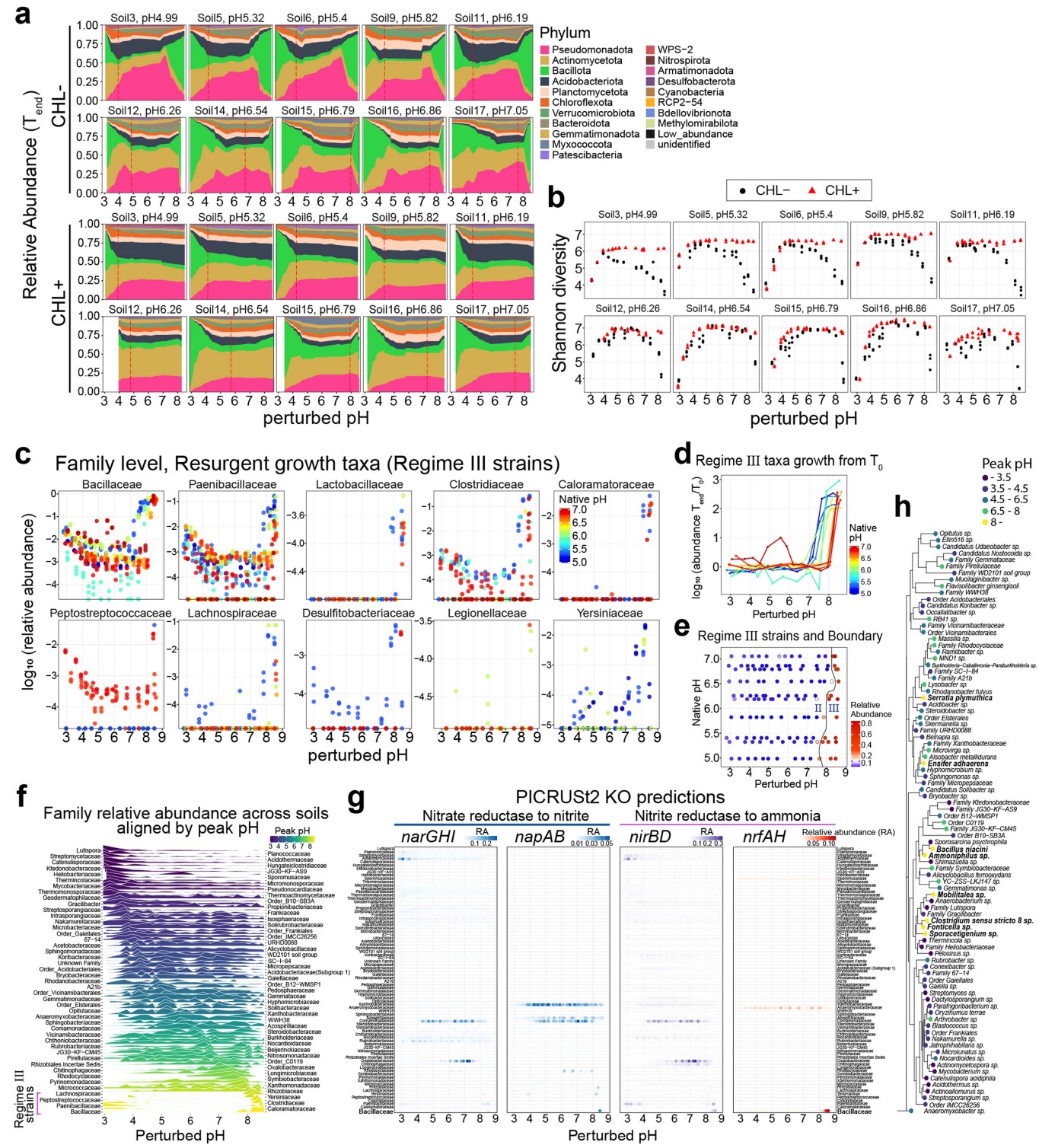

**Extended Data Fig. 6** | See next page for caption.

**Extended Data Fig. 6 | Taxonomy, diversity, and traits of Resurgent growth (Regime III) strains. (a)** Endpoint relative abundance in the phylum level is plotted across the perturbed pH for ten different CAF soils with (CHL+) and without chloramphenicol (CHL-) treatment ($n$ = 258 conditions). The alluvial plots were constructed by connecting the average relative abundance values from biological triplicates of 13 different pH perturbed levels. Red vertical dashed lines indicate the stabilized endpoint pH (1M KCl) of the pH-unperturbed samples. **(b)** Shannon diversity of the endpoint community is plotted across the perturbed pH for ten different soils in CHL+/− conditions ($n$ = 772 replicates). **(c-e)** To identify the taxa accountable for the emergence of Regime III at a finer taxonomic level, we conducted a differential abundance analysis (see Methods, $z > 4.2$) that statistically determined which Amplicon sequence variants (total $n$ = 34,696 ASVs) were significantly enriched in CHL-conditions compared to the CHL+ in Regime III. **(c)** After aggregating those Regime III ASVs at the family level, we plotted relative abundance ($\log_{10}$ scale) of 10 families against perturbed pH (x-axis), colors indicating the sample's native soil pH. **(d)** We computed the growth of absolute abundance from T0 ($T_{end}/T_0$) of the aggregated absolute abundance of all ASVs belonging to the 10 families identified as Regime III family and plotted it against perturbed pH. **(e)** The relative abundance of aggregated Regime III taxa as a heatmap with perturbed pH on the x-axis and native pH on the y-axis. Regime II-III boundary is shown in black. **(f)** To observe the pH niche of each taxon, we aggregated the relative abundance of significantly enriched ASVs within each family ($n$ = 89 families) for every pH-perturbed condition. We then calculated the median relative abundance across soils for each pH bin (see Methods). Families were arranged by their peak pH, with acidic peak pH families (dark blue) at the top and basic peak pH families (yellow) at the bottom. Notably, families (e.g., Bacillaceae) associated with resurgent growth had a peak pH above 8. **(g)** To infer Regime III strain genotypes, we used PICRUSt2 to predict KEGG ortholog (KO) gene abundance from the 16S rRNA sequence of each ASV (see Methods). We focused on denitrification and DNRA-related KOs: nitrate reductases (*narGHI*, *napAB*) and nitrite reductases to ammonium (*nirBD*, *nrfAH*). For the 89 families analyzed in **(f)**, we summed the relative abundance (reads/total reads at each pH level in CHL- samples) of ASVs in each family that had at least one gene for *narGHI*, *napAB*, *nirBD*, and *nrfAH*, respectively. Relative abundance values were plotted across pH for all soils, with color intensity indicating abundance. Among the Resurgent growth taxa, the *Bacillaceae* family showed notable enrichment in *nrfAH* genes (red points), which are DNRA-related genes, producing ammonium from nitrite. **(h)** To assess phylogenetic convergence among taxa with similar pH niches, we constructed a phylogenetic tree using the 16S rRNA V3-4 region of one representative ASV with the highest relative abundance per family (Methods). Nodes are labeled by genus or species and colored by peak pH. Resurgent growth families (yellow) were dispersed throughout the tree, indicating no phylogenetic convergence.

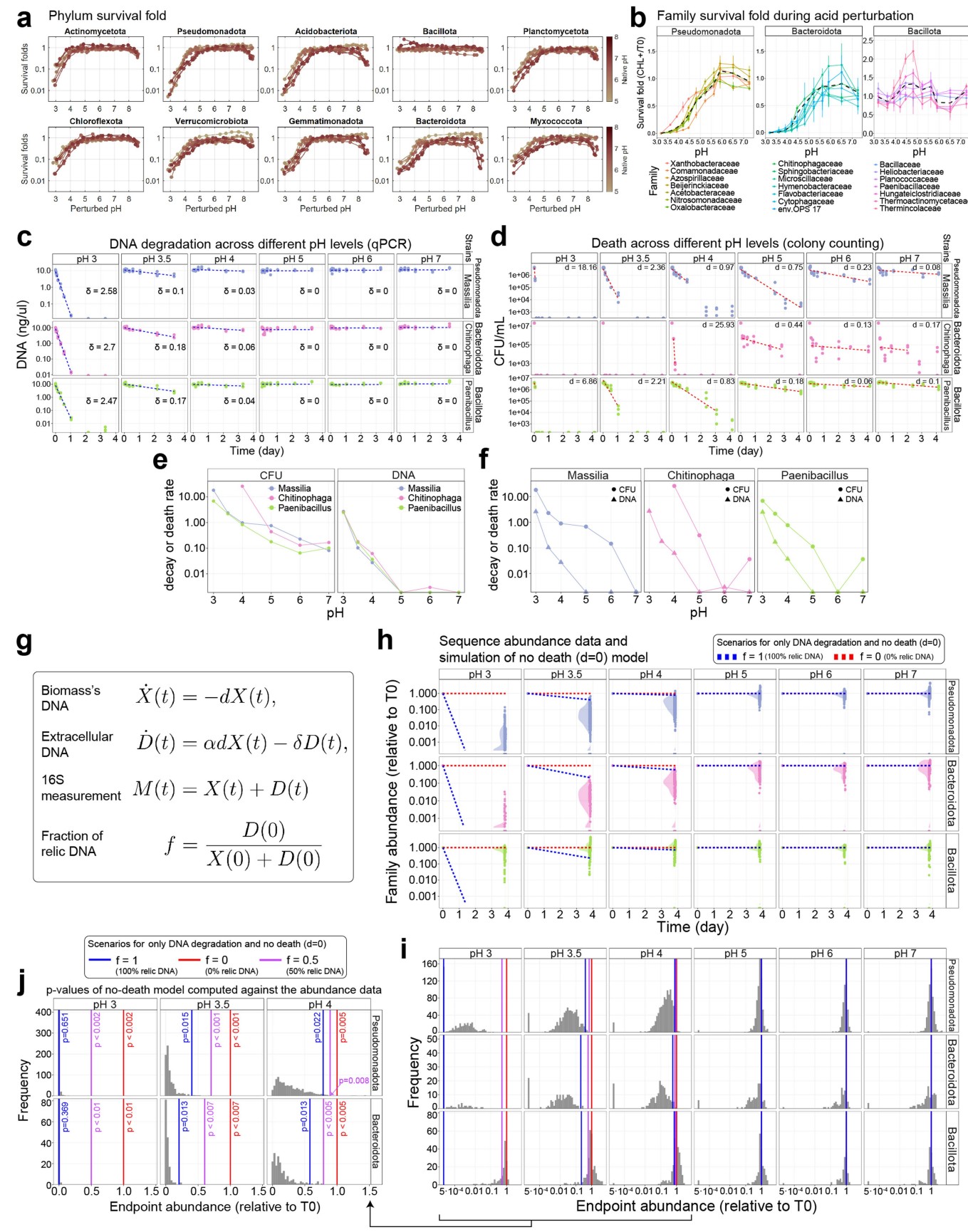

**Extended Data Fig. 7** | See next page for caption.

**Extended Data Fig. 7 | Widespread death explains Acidic death regime (Regime I). (a)** To infer putative death, we computed the Survival folds of the top 10 phyla (abundance-wise) among 40 phyla for each perturbed pH condition by computing the fold difference in the endpoint absolute abundance of each phylum under CHL+ conditions relative to their baseline levels at the initial time point ($Abs_{CHL+}/Abs_{T0}$) ($n = 130$ mean fold values for each phylum). Using CHL+ conditions ruled out growth, isolating the death effect of pH. Each line represents a soil, with native pH shown by color, and each panel represents a phylum. A consistent drop in survival folds under acidic perturbation was observed for all phyla except Bacillota. **(b)** Survival fold in the family level across perturbed pH levels for seven major families respectively within Pseudomonadota, Bacteroidota, and Bacillota phyla. Samples were from acidic perturbation in Soil 16 (Table S1). Error bars represent standard deviations of 3 slurry replicates. The black dotted line shows the phylum-level survival fold from **(a)**. **(c-i)** Justifying widespread death as the mechanism of Regime I rather than DNA degradation caused by acidic pH (see SI). **(c)** DNA degradation rates across pH levels were measured by qPCR over a time series using genomic DNA (gDNA) extracted from three isolates: *Massilia* sp. (Pseudomonadota), *Chitinophaga* sp. (Bacteroidota), and *Paenibacillus* sp. (Bacillota). The y-axis shows DNA concentration (ng/$\mu$L) in log10 scale. 10 ng/$\mu$L of gDNA was incubated in phosphate buffers at pH 3, 3.5, 4, 5, 6, and 7 in triplicate. DNA degradation rate $\delta$ (1/day) was calculated via linear regression on log10-transformed non-zero DNA concentrations (dotted blue lines). **(d)** Death rates of the same isolates across pH levels were measured by colony-forming units (CFUs) over time. Strains were anaerobically incubated in pH-buffered media without growth nutrients and plated for CFU counts ($n = 3$ biological replicates). Death rates $d$ (1/day) were determined by linear regression on log10-transformed CFU counts above $10^3$ CFU/mL (dotted red lines). **(e)** The three strains had similar DNA degradation rates $\delta$ across pH levels (left) but distinct death rates $d$ (right). The y-axis shows degradation or death rates. **(f)** Death rates $d$ (CFU, circle) were consistently higher than DNA degradation rates $\delta$ (DNA, triangle) across all strains at each pH. Baseline death effects at neutral pH were subtracted from $d$. **(g-j)** Death is necessary to explain sequencing measurements. **(g)** ODE equations describe biomass death and DNA degradation. Sequencing measurements ($M(t)$) include DNA from functional biomass ($X(t)$) and relic DNA from dead cells ($D(t)$). **(h)** Endpoint family abundances in three phyla (rows) across Regimes I and II (columns) ($n = 5,765$ family endpoint abundance data points). A sample was included if its endpoint pH was within $\pm 0.2$ of the labeled. The y-axis shows family abundance normalized to initial abundance ($M(t = endpoint)/M(0)$, equation (5) in the Supplementary Information, **(g)**). Points represent normalized abundances for each slurry sample across soils in each pH bin. Half-violin plots show endpoint normalized abundance distributions ($n = 5,765$ points). Dotted lines show predicted abundance assuming no death ($d = 0$). Red assumes no relic DNA ($f = 0$), keeping $M(t)$ constant at $M(0)$. Blue assumes all sequences are relic DNA ($f = 1$), with $M(t)$ decaying at $\delta$ (**(c)**). **(i)** Histograms of endpoint abundance from **(h)** for each pH and phylum, with vertical lines for no-death model predictions under varying $f$ (blue: $f = 1$, red: $f = 0$, purple: $f = 0.5$). **(j)** P-values for each hypothesized $f$ in Pseudomonadota and Bacteroidota for pH $\leq 4$. Vertical lines (blue: $f = 1$, red: $f = 0$, purple: $f = 0.5$) show no-death model predictions. P-values $\leq 0.05$ indicate that assuming no death and only DNA degradation cannot explain sequence abundances, rejecting the no-death hypothesis.

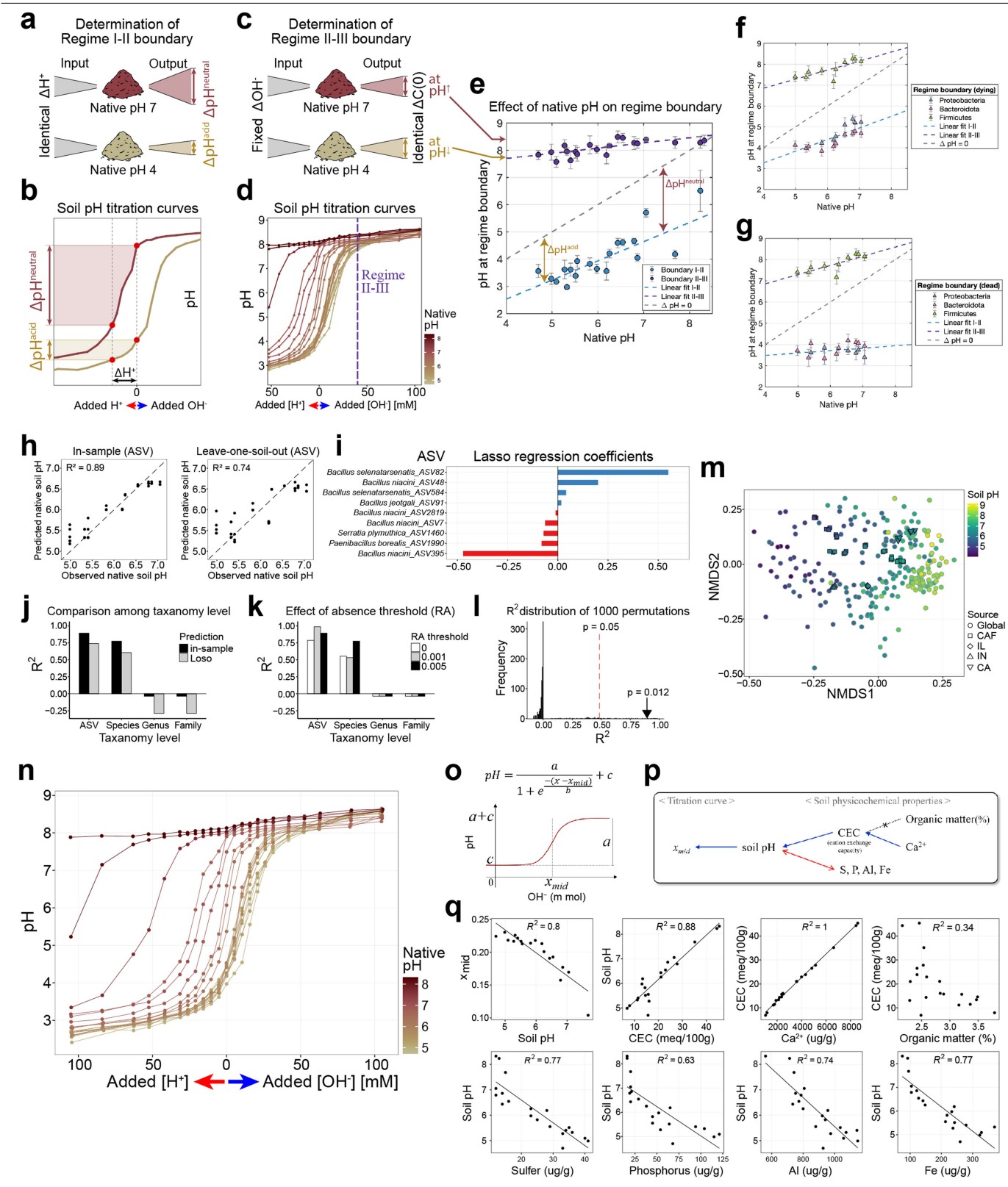

**Extended Data Fig. 8 |** See next page for caption.

**Extended Data Fig. 8 | Impact of native pH on regime boundaries, community composition, and soil physicochemical properties. (a)** Cartoon illustrating how native pH impacts the Regime I-II boundary. Identical acid concentrations ($\Delta H^+$) cause larger pH changes in neutral soils ($\Delta pH^{neutral}$) compared to acidic soils ($\Delta pH^{acidic}$), as explained by their positions on the titration curve in **(b)**. **(b)** Titration curves showing endpoint pH (y-axis) versus acid/base added (x-axis) for neutral (dark brown) and acidic (light brown) soils. The dashed vertical line at 0 marks the unperturbed pH. For the same $\Delta H^+$, neutral soils undergo greater pH changes (shaded regions). This suggests acidic soils transition from Regime II to I after smaller pH perturbations (blue points, **(e)**). **(c)** Cartoon showing how native pH affects the Regime II-III boundary. Fixed NaOH additions release identical carbon ($\Delta C(0)$, Fig. 5), with larger $C(0)$ driving transitions to Regime III. Neutral soils reach higher pH values for the same NaOH additions due to titration curves **(d)**. **(d)** Soil pH titration curves of soils from varying native pH levels (full titration curve shown in **(n)**). The vertical dashed line shows NaOH added to transition from Regime II to III. Neutral soils (darker colors) achieve higher pH values, aligning with the increasing Regime II-III boundary pH (purple points, **(e)**). **(e)** Regime transition pH levels (y-axis) for soils with different native pH levels (x-axis). Blue points show Regime II-I boundaries, and purple points show Regime II-III. Boundary pH is the midpoint between the last Regime I/II sample's pH and the first Regime II/III sample's pH. Error bars represent pH differences between these samples. The dashed blue line (Regime I-II boundary) and dashed purple line (Regime II-III boundary) are weighted least squares fits, with the weights inversely proportional to the error of each point. The dashed black line (slope 1) indicates a constant change in pH from native to the regime boundary for all soils. Slopes differing from 1 show that this pH change depends on native soil pH. The blue dashed line has a slope of 0.7 (95% CI: [0.44, 0.97]). **(f-g)** Using sequencing data, we examined whether taxa adaptation to native pH influences regime boundary pH levels. **(f)** To analyze the transition to the Acidic death regime (Regime I), we calculated survival folds for Pseudomonadota and Bacteroidota across perturbed pH levels as $Abs_{CHL+}/Abs_{T_0}$ (endpoint abundance in CHL+ vs. initial time point, $T_0$). Using a uniform survival fold threshold, we determined the pH where survival folds fell below the threshold, defined as either (1) "dying" (<1) or (2) "dead" ($\rightarrow 0$). For Resurgent growth regime (Regime III), growth folds for Bacillota were calculated as $Abs_{CHL-}/Abs_{CHL+}$, with a growth fold threshold of 3 to find the boundary pH. **(f)** Boundaries using the "dying"

definition show a Regime I-II slope (0.56 ± 0.09), indicating neutral soils tolerate larger $\Delta pH$. **(g)** The "dead" definition for Pseudomonadota and Bacteroidota shows Regime II-I pH transition points with a slope (0.11 ± 0.08) near 0, suggesting death occurs at a fixed pH (-3.5). Error bars reflect pH differences around threshold crossings; points represent midpoint pH. Linear fits were computed using least squares (blue/purple dashed lines). The grey dashed line represents $y = x$. **(h-l)** Taxonomy of growing strains in Regime III varies with soil native pH (see Methods). **(h)** Predicted vs. observed soil pH using LASSO-regularized regression based on ASV presence/absence (threshold: relative abundance > 0.005) to predict soil pH. Left: in-sample predictions; right: 'Leave-one-soil-out' (Loso, Methods). Prediction quality ($R^2$) is calculated from mean predicted vs. observed pH. **(i)** Bar plots of non-zero ASV regression coefficients from in-sample predictions in **(h)**. **(j)** $R^2$ of regression by taxonomic level for presence/absence. Predictions fail above the genus level. Bars: dark = in-sample, grey = Loso. **(k)** Impact of relative abundance threshold on presence/absence definition. **(l)** A permutation test was conducted to evaluate the significance of ASV-level predictions. Native pH values were permuted 1000 times to generate $R^2$ distribution. The observed $R^2$ in **(h)** (black arrow) has p = 0.012. **(m)** The community composition of our soils at T0 samples effectively spans the pH gradient of the global topsoil microbiome[19]. To compute the compositional difference between our soils used for the experiment and global topsoils ($n = 237$ global samples)[19], we performed non-metric multidimensional scaling (NMDS; Vegan v2.5.7) on the Bray-Curtis distance matrix at the family level (k=3, stress=0.11). Points were colored by soil pH. Global: global topsoil data, CAF: Cook Agronomy Farm, IL: LaBagh in Illinois, IN: Pinhook in Indiana, CA: Sedgwick in California (see Table S1). **(n-q)** pH titration curves and physicochemical properties across soils (see SI). Fitting a logistic function **(o)** to soil pH titration curves **(n)** for 20 CAF soils across native pH levels. **(n)** Soil pH titration curves of CAF soils (colored by native pH). The y-axis shows endpoint pH after 4 days of incubation following pH perturbation ($n = 457$ endpoint samples). **(o)** Logistic function and parameters. Titrations with $H^+$ and $OH^-$ were unified on an $OH^-$ x-axis by shifting curves 0.2 mmol right, aligning them at 0 mmol $OH^-$. **(p)** Summary of correlations between soil physicochemical properties and titration curve characteristics. Native soil pH shifts the titration curves horizontally, moving $X_{mid}$. **(q)** Correlations supporting diagram **(p)** with $R^2$ values ($n = 20$ CAF soils; Soil 1–18 used for $X_{mid}$ values).

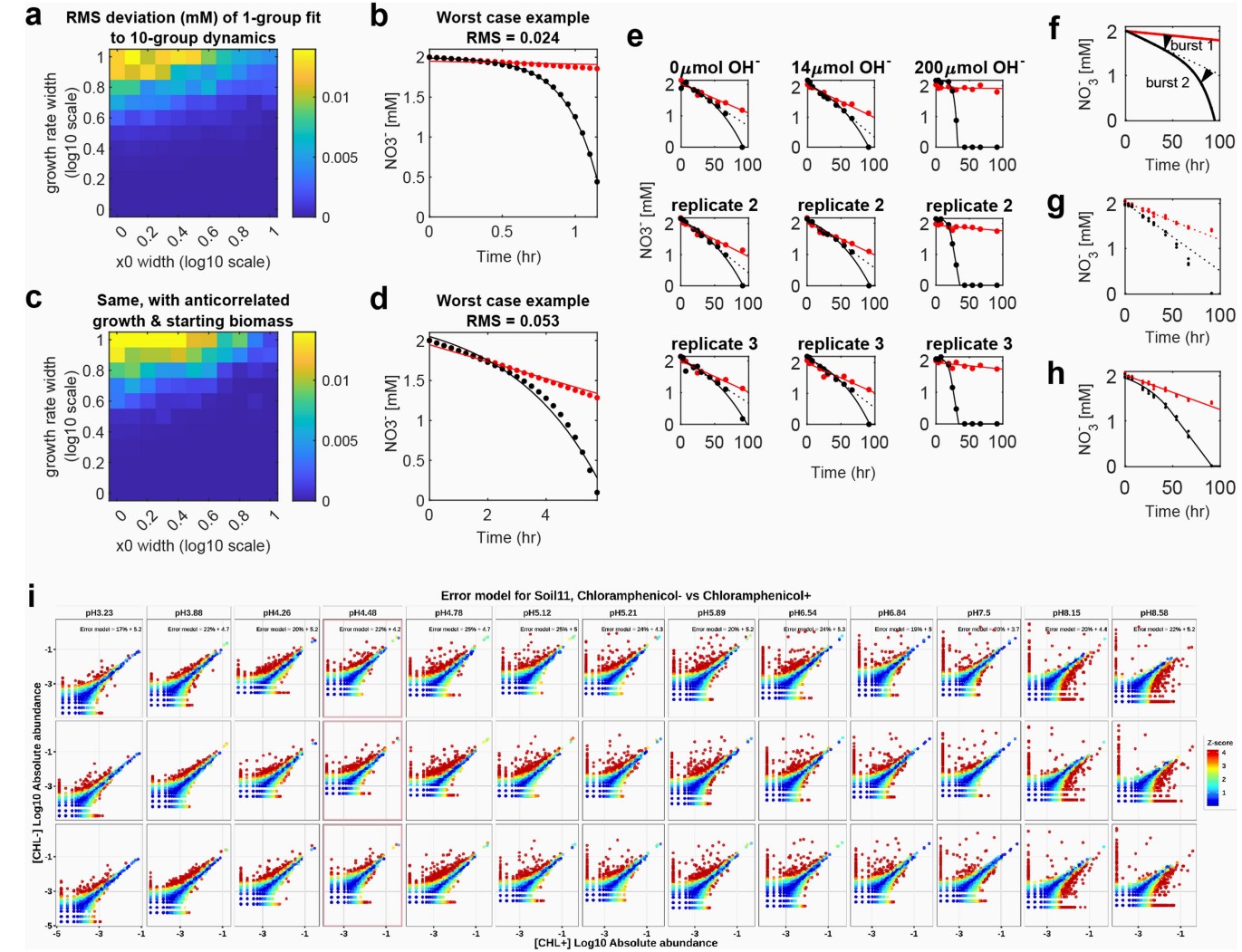

**Extended Data Fig. 9 |** See next page for caption.

**Extended Data Fig. 9 | Justifying the effective 1-biomass model despite the diversity of denitrifying taxa (see SI). (a)** Single-biomass approximation quality for simulations when the "true" underlying dynamics involve 10 taxa. The species' growth rates and initial abundances are drawn randomly from distributions whose widths are varied (see SI). The community is initialized with 2 mM nitrate, the dynamics are simulated according to Eq.(9) in the Supplementary Information, and the results are fit to the single-biomass effective model. The heatmap reports the root-mean-square deviation of the fit (units mM nitrate), averaged over 100 random instances. Experimental errors are estimated to be 0.01-0.02 mM (assessed via replicate-replicate variability of early-time measurements in CHL$^+$ samples). **(b)** An example of a "poor fit" (corresponding to the top-left corner of panel A heatmap). Points, simulated data (10-taxon dynamics); solid lines, 1-biomass model fit. Black and red correspond to CHL$^-$ and CHL$^+$, respectively. Deviations of this scale would be undetectable within the precision of the experiment. **(c, d)** Same as panels **(a, b)**, but the 10 random species are drawn with anti-correlated growth rate and initial biomass, which is expected to worsen the performance of the single-biomass model (see SI). The RMS deviations show a weak increase. For simulations in this figure, both the simulations and the model fits assume no carbon limitation. **(e)** Examples from data exhibiting deviations from the single-biomass model. Shown are the three most basic conditions in soil 16 (pH increasing from left to right). The three rows correspond to the three replicates (independent incubations), shown separately for clarity. In all 3 panels in the right column (extreme basic perturbation), dynamics are consistent with the rapid growth of a single denitrifying taxon rising from low abundance (Regime III). The 6 other panels exhibit (weak) deviations that are arguably consistent with the "failure mode" predicted by the analysis in **(a-d)**: The curve in the CHL$^-$ slope is ever so slightly underestimated, and at 50-60 hours, the fit curve consistently passes below the data points. This suggests that in these conditions (which lie at the boundary of Regime II and Regime III),

the two strongly distinct groups (initially abundant slow growers and initially negligible fast growers) might both be contributing to denitrification. However, the effect is very weak, so even in this case, the single-biomass model remains an excellent approximation. **(f-h)** Failure of the 1-biomass model due to multiple carbon sources. **(f)** A simulated example of a nitrate utilization curve with two distinct growth bursts (one early, one late, as labeled). **(g)** The strongest example in our data (Soil 14, weak acidic perturbation with 4 $\mu$M $H^+$ ions) shows all three replicates. Dotted lines (linear fits to first few datapoints of CHL$^-$ and CHL$^+$) are guides for the eye. Note the key difference between this example and those shown in **(e)**: in both cases, we see some late-time growth, but previously, the initial utilization slopes in CHL$^-$ and CHL$^+$ were the same. Here, the initial slopes differ strongly between CHL + and CHL − . In our model, differing early slopes and a late time speed up in CHL − is only possible if some denitrifiers grow in the first few hours, stop growing, and then a second phase of growth must occur at later times. In our model, this requires two carbon sources utilized by different taxa. **(h)** Even in this case, the single-biomass model provides an excellent fit of dynamics. **(i)** ASV abundance error model. To identify ASVs enriched for each perturbed pH level, we constructed a null model using three biological replicates per condition (see Methods). For each soil (e.g., Soil 11) and perturbed pH (titles), we plot log-scale ASV abundance in CHL+ samples (x-axis) vs. CHL− samples (y-axis) across replicates (rows). Deviations from the 1:1 line follow a Gaussian noise model with fractional magnitude $c_{frac}$ and constant magnitude $c_0$, such that replicate measurements of an ASV with mean abundance $n$ counts have a standard deviation $\sigma(c_0, c_{frac}) = \sqrt{(c_{frac}n)^2 + c_0^2}$. $c_{frac}$ was estimated from moderate-abundance ASVs (>50 counts), and $c_0$ was set so 67% of comparisons are within $\pm \sigma(c_0, c_{frac})$. This model was inferred for each soil and pH level. Noise parameters are shown in the panel. Points are colored by z-scores from the error model. The pink box indicates no acid/base addition.

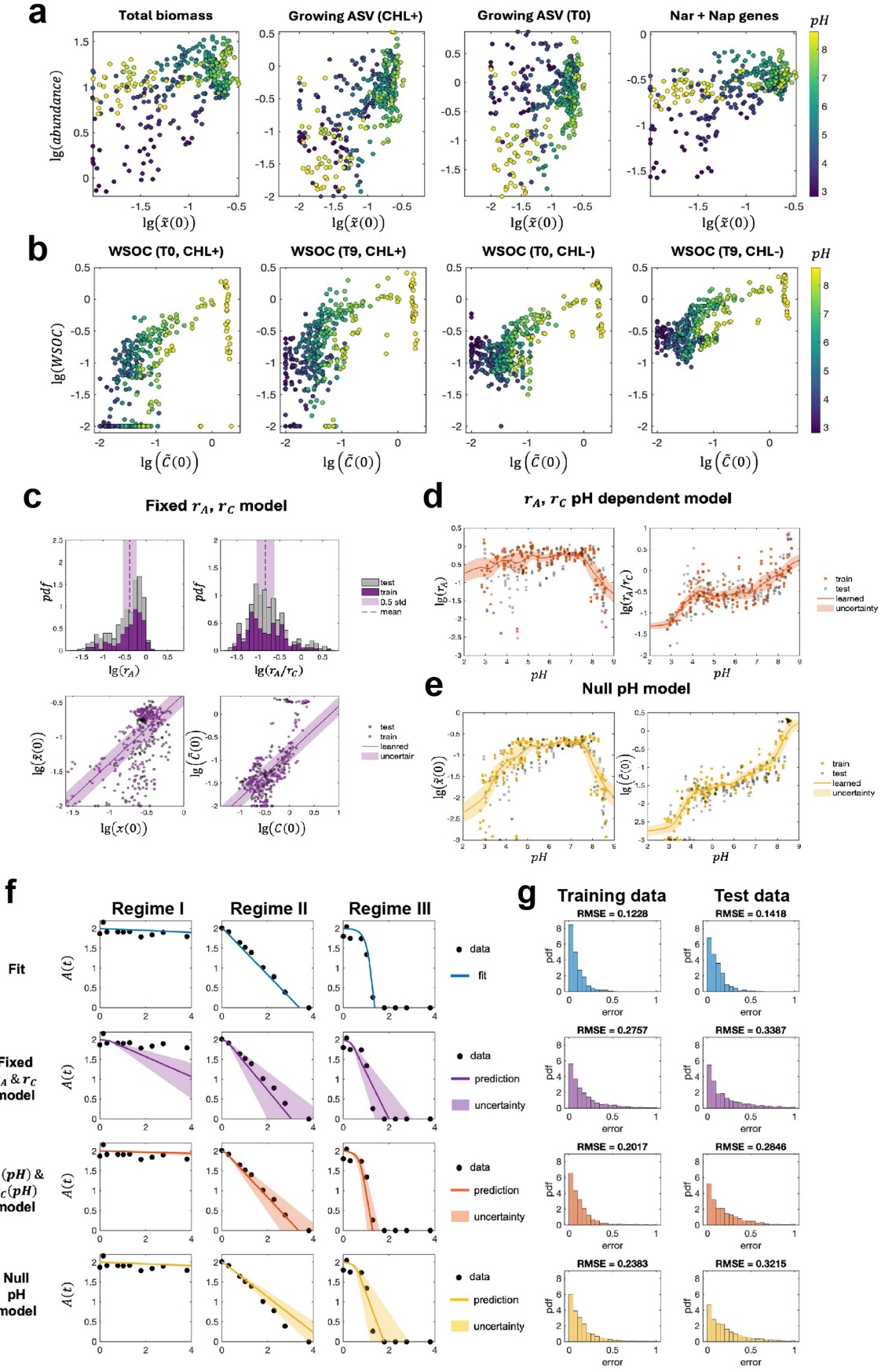

**Extended Data Fig. 10** | See next page for caption.

**Extended Data Fig. 10 | Prediction of nitrate dynamics from community structure (see SI). (a-b)** Candidate variables for feature selection to estimate $x(0)$ and $C(0)$ from sequencing and WSOC (water-soluble organic carbon) measurements. **(a)** Scatter plots of $\widetilde{x}(0)$ against (1) total absolute abundance (total sequencing reads, first panel), (2) summed absolute abundance of significantly enriched ASVs in CHL − relative to CHL+ (second panel), (3) summed abundance of significantly enriched ASVs in CHL − relative to T0 (third column), and (4) summed abundances of all ASVs that possess Nar and Nap genes in CHL+ samples inferred via PICRUST2[28] (last panel). **(b)** Scatter plots between $\widetilde{C}(0)$ and WSOC measurements in CHL+ or CHL- samples at T0 or T9 (endpoint) as indicated in the panel titles. Colors show endpoint perturbed pH in both **(a, b)**. Note the log scale. **(c-e)** Models for predicting $r_A$ and $r_A/r_C$ from estimates of $\widetilde{x}(0)$, $\widetilde{C}(0)$, $x(0)$, $C(0)$. Shows estimates of: (1) $r_A = \widetilde{x}(0)/x(0)$ where $\widetilde{x}(0)$ and $x(0)$ come from nitrate dynamics and the variables identified in **(a, b)** respectively. (2) $r_A/r_C = \widetilde{C}(0)/C(0)$ again with $\widetilde{C}(0)$ from nitrate dynamics and $C(0)$ from WSOC measurements in **(a, b)**. **(c)** Assumes fixed values of $r_A$ and $r_C$. The top two panels present the log-distribution of $r_A$ and $r_A/r_C$ computed by $lg(\widetilde{x}(0)/x(0))$ and $lg(\widetilde{C}(0)/C(0))$, where dashed lines and shades show the mean values and 0.5 standard deviations. The bottom two panels present the learned $\widetilde{x}(0)$ and $\widetilde{C}(0)$ (from nitrate dynamics) and the uncertainty of learning (0.5 standard deviation) using purple curves and shaded region. x-axis values are Nar+Nap gene abundances and WSOC data estimated in **(a, b)**. **(d)** The pH-dependent $r_A$ and $r_A/r_C$ model (see SI). The y-axes are $lg(r_A)$ and $lg(r_A/r_C)$, computed by $lg(\widetilde{x}(0)/x(0))$ and $lg(\widetilde{C}(0)/C(0))$. Orange curves and shades represent the learned $lg(r_A) = f_1(pH)$ and $lg(r_A/r_C) = f_2(pH)$ functions (lines) and learning uncertainty (shaded regions). **(e)** The null pH model which predicts $\widetilde{x}(0)$ and $\widetilde{C}(0)$ as a function of pH without estimating $r_A$ and $r_A/r_C$. Yellow curves and shades represent the learned $lg(\widetilde{x}(0))$ and $lg(\widetilde{C}(0))$ as pH-dependent functions and learning uncertainty. The rightmost column of each row was used as an estimate of $x(0)$ and $C(0)$. **(f-g)** Predicting nitrate dynamics out of sample from sequencing and WSOC measurements. **(f)** The examples of fit with our effective 1-biomass consumer resource model (Fig. 3, top row) and the three models presented in **(c-e)**, bottom three rows) in each of the three functional regimes (three columns). The pH-dependent model (orange, third row) has the best prediction performance, which is close to the fit for the 1-biomass consumer-resource model (blue, first row). **(g)** The fitting and prediction errors on the training dataset (left column) and the test dataset (right column). The pH-dependent model (orange) has the smallest errors among the three prediction models.

# Reporting Summary

Please do not complete any field with "not applicable" or n/a.  Refer to the help text for what text to use if an item is not relevant to your study.
For final submission: please carefully check your responses for accuracy; you will not be able to make changes later.

## Statistics

For all statistical analyses, confirm that the following items are present in the figure legend, table legend, main text, or Methods section.

| n/a | Confirmed | |
|---|---|---|
| ☐ | ☑ | The exact sample size (*n*) for each experimental group/condition, given as a discrete number and unit of measurement |
| ☐ | ☑ | A statement on whether measurements were taken from distinct samples or whether the same sample was measured repeatedly |
| ☐ | ☑ | The statistical test(s) used AND whether they are one- or two-sided<br>*Only common tests should be described solely by name; describe more complex techniques in the Methods section.* |
| ☐ | ☑ | A description of all covariates tested |
| ☐ | ☑ | A description of any assumptions or corrections, such as tests of normality and adjustment for multiple comparisons |
| ☐ | ☑ | A full description of the statistical parameters including central tendency (e.g. means) or other basic estimates (e.g. regression coefficient) AND variation (e.g. standard deviation) or associated estimates of uncertainty (e.g. confidence intervals) |
| ☐ | ☑ | For null hypothesis testing, the test statistic (e.g. *F*, *t*, *r*) with confidence intervals, effect sizes, degrees of freedom and *P* value noted<br>*Give P values as exact values whenever suitable.* |
| ☑ | ☐ | For Bayesian analysis, information on the choice of priors and Markov chain Monte Carlo settings |
| ☐ | ☑ | For hierarchical and complex designs, identification of the appropriate level for tests and full reporting of outcomes |
| ☑ | ☐ | Estimates of effect sizes (e.g. Cohen's *d*, Pearson's *r*), indicating how they were calculated |

*Our web collection on statistics for biologists contains articles on many of the points above.*

## Software and code

Policy information about availability of computer code

| Data collection | Raw sequence reads under NCBI BioProject ID PRJNA1205727. Data tables are at OSF (doi.org/10.17605/OSF.IO/CTF8K) |
|---|---|
| Data analysis | All codes deposited at Open Science Framework doi.org/10.17605/OSF.IO/CTF8K |

For manuscripts utilizing custom algorithms or software that are central to the research but not yet described in published literature, software must be made available to editors and reviewers. We strongly encourage code deposition in a community repository (e.g. GitHub). See the Nature Portfolio guidelines for submitting code & software for further information.

## Data

Policy information about availability of data

All manuscripts must include a data availability statement. This statement should provide the following information, where applicable:
- Accession codes, unique identifiers, or web links for publicly available datasets
- A description of any restrictions on data availability
- For clinical datasets or third party data, please ensure that the statement adheres to our policy

Raw sequence reads are under NCBI BioProject ID PRJNA1205727. Data tables are at OSF (doi.org/10.17605/OSF.IO/CTF8K)

## Research involving human participants, their data, or biological material

Policy information about studies with human participants or human data. See also policy information about sex, gender (identity/presentation), and sexual orientation and race, ethnicity and racism.

| | |
|---|---|
| Reporting on sex and gender | |
| Reporting on race, ethnicity, or other socially relevant groupings | |
| Population characteristics | |
| Recruitment | |
| Ethics oversight | |

Note that full information on the approval of the study protocol must also be provided in the manuscript.

## Field-specific reporting

Please select the one below that is the best fit for your research. If you are not sure, read the appropriate sections before making your selection.

☐ Life sciences          ☐ Behavioural & social sciences          ☑ Ecological, evolutionary & environmental sciences

For a reference copy of the document with all sections, see nature.com/documents/nr-reporting-summary-flat.pdf

## Life sciences study design

All studies must disclose on these points even when the disclosure is negative.

| | |
|---|---|
| Sample size | For metabolites, we had 20 soils each with 13 pH-perturbed conditions (except Soil19, 20). We used 10 soils for sequencing. |
| Data exclusions | For time series metabolite sampling or endpoint sequencing, we discarded the replicates with experimental errors. |
| Replication | All time series metabolite sampling and sequencing were done in three biological replicates. |
| Randomization | For field sampling, we used soil samples with desired native pH. We homogenized each soil and subsampled for experiment. |
| Blinding | |

## Behavioural & social sciences study design

All studies must disclose on these points even when the disclosure is negative.

| | |
|---|---|
| Study description | |
| Research sample | |
| Sampling strategy | |
| Data collection | |
| Timing | |
| Data exclusions | |
| Non-participation | |
| Randomization | |

# Ecological, evolutionary & environmental sciences study design

All studies must disclose on these points even when the disclosure is negative.

| | |
|---|---|
| Study description | Sampled topsoil across a pH gradient in the Cook Agronomy Farm, WA, USA |
| Research sample | Topsoil samples (depth 0-20cm) |
| Sampling strategy | Targeted search of different pH ranges at the site |
| Data collection | Metabolic measurements and 16S rRNA sequence data |
| Timing and spatial scale | 4-day incubation for the metabolic measurements |
| Data exclusions | NA |
| Reproducibility | Included replicates and controls for reproducible results |
| Randomization | Three biological replicates for metabolic measurements |
| Blinding | NA |

Did the study involve field work?  ☑ Yes   ☐ No

## Field work, collection and transport

| | |
|---|---|
| Field conditions | Dry season, September 8-12, 2022, topsoil sampled post-harvest of spring crops |
| Location | Cook Agronomy Farm, Pullman, WA, USA / Additional soil sampling from IL, IN, CA, USA |
| Access & import/export | |
| Disturbance | |

# Reporting for specific materials, systems and methods

We require information from authors about some types of materials, experimental systems and methods used in many studies. Here, indicate whether each material, system or method listed is relevant to your study. If you are not sure if a list item applies to your research, read the appropriate section before selecting a response.

### Materials & experimental systems

| n/a | Involved in the study |
|---|---|
| ☑ | ☐ Antibodies |
| ☑ | ☐ Eukaryotic cell lines |
| ☑ | ☐ Palaeontology and archaeology |
| ☑ | ☐ Animals and other organisms |
| ☑ | ☐ Clinical data |
| ☑ | ☐ Dual use research of concern |
| ☑ | ☐ Plants |

### Methods

| n/a | Involved in the study |
|---|---|
| ☑ | ☐ ChIP-seq |
| ☑ | ☐ Flow cytometry |
| ☑ | ☐ MRI-based neuroimaging |

## Antibodies

| | |
|---|---|
| Antibodies used | |
| Validation | |

# Eukaryotic cell lines

Policy information about cell lines and Sex and Gender in Research

Cell line source(s)

Authentication

Mycoplasma contamination

Commonly misidentified lines
(See ICLAC register)

# Palaeontology and Archaeology

Specimen provenance

Specimen deposition

Dating methods

☐ Tick this box to confirm that the raw and calibrated dates are available in the paper or in Supplementary Information.

Ethics oversight

Note that full information on the approval of the study protocol must also be provided in the manuscript.

# Animals and other research organisms

Policy information about studies involving animals; ARRIVE guidelines recommended for reporting animal research, and Sex and Gender in Research

Laboratory animals

Wild animals

Reporting on sex

Field-collected samples

Ethics oversight

Note that full information on the approval of the study protocol must also be provided in the manuscript.

# Clinical data

Policy information about clinical studies
All manuscripts should comply with the ICMJE guidelines for publication of clinical research and a completed CONSORT checklist must be included with all submissions.

Clinical trial registration

Study protocol

Data collection

Outcomes

# Dual use research of concern

Policy information about dual use research of concern

## Hazards

Could the accidental, deliberate or reckless misuse of agents or technologies generated in the work, or the application of information presented in the manuscript, pose a threat to:

|    | No | Yes |                              |
|----|----|-----|------------------------------|
|    | ☑  | ☐   | Public health                |
|    | ☑  | ☐   | National security            |
|    | ☑  | ☐   | Crops and/or livestock       |
|    | ☑  | ☐   | Ecosystems                   |
|    | ☑  | ☐   | Any other significant area   |

### Experiments of concern

Does the work involve any of these experiments of concern:

| No | Yes |                                                                      |
|----|-----|----------------------------------------------------------------------|
| ☑  | ☐   | Demonstrate how to render a vaccine ineffective                      |
| ☑  | ☐   | Confer resistance to therapeutically useful antibiotics or antiviral agents |
| ☑  | ☐   | Enhance the virulence of a pathogen or render a nonpathogen virulent |
| ☑  | ☐   | Increase transmissibility of a pathogen                              |
| ☑  | ☐   | Alter the host range of a pathogen                                   |
| ☑  | ☐   | Enable evasion of diagnostic/detection modalities                    |
| ☑  | ☐   | Enable the weaponization of a biological agent or toxin              |
| ☑  | ☐   | Any other potentially harmful combination of experiments and agents  |

# Plants

| Seed stocks | |
|---|---|
| Novel plant genotypes | |
| Authentication | |

# ChIP-seq

### Data deposition

☐ Confirm that both raw and final processed data have been deposited in a public database such as GEO.

☐ Confirm that you have deposited or provided access to graph files (e.g. BED files) for the called peaks.

| Data access links
*May remain private before publication.* | |
|---|---|
| Files in database submission | |
| Genome browser session
(e.g. UCSC) | |

### Methodology

| Replicates | |
|---|---|
| Sequencing depth | |
| Antibodies | |
| Peak calling parameters | |
| Data quality | |

| Software | |
|---|---|

# Flow Cytometry

## Plots

Confirm that:

☐ The axis labels state the marker and fluorochrome used (e.g. CD4-FITC).

☐ The axis scales are clearly visible. Include numbers along axes only for bottom left plot of group (a 'group' is an analysis of identical markers).

☐ All plots are contour plots with outliers or pseudocolor plots.

☐ A numerical value for number of cells or percentage (with statistics) is provided.

## Methodology

| Sample preparation | |
|---|---|
| Instrument | |
| Software | |
| Cell population abundance | |
| Gating strategy | |

☐ Tick this box to confirm that a figure exemplifying the gating strategy is provided in the Supplementary Information.

# Magnetic resonance imaging

## Experimental design

| Design type | |
|---|---|
| Design specifications | |
| Behavioral performance measures | |

| Imaging type(s) | |
|---|---|
| Field strength | |
| Sequence & imaging parameters | |
| Area of acquisition | |

Diffusion MRI    ☐ Used    ☐ Not used

## Preprocessing

| Preprocessing software | |
|---|---|
| Normalization | |
| Normalization template | |
| Noise and artifact removal | |
| Volume censoring | |

## Statistical modeling & inference

| Model type and settings | |
|---|---|
| Effect(s) tested | |

Specify type of analysis:  ☐ Whole brain  ☐ ROI-based  ☐ Both

Statistic type for inference

(See Eklund et al. 2016)

Correction

## Models & analysis

| n/a | Involved in the study |
|---|---|
| ☐ ☐ | Functional and/or effective connectivity |
| ☐ ☐ | Graph analysis |
| ☐ ☐ | Multivariate modeling or predictive analysis |

Functional and/or effective connectivity

Graph analysis

Multivariate modeling and predictive analysis

