## [Peer Review file · Nature]

Functional regimes define soil microbiome response to environmental change

Corresponding Author: Professor Seppe Kuehn

Version 2:

Reviewer comments:

Referee #1

(Remarks to the Author)

The main aim of the study is to identify how natural microbial communities respond to environmental changes. The authors formulate and analyze a minimal consumer resource model based on high-throughput measurements of soil nutrient dynamics in the lab. The samples stem from 20 locations that vary in their initial pH from a single agricultural field, and soil slurries are subjected to pH perturbations with and without the addition of chloramphenicol, a translation inhibiting antibiotic that suppresses the growth of many lineages. The main claim is that the approach that combines modeling and experiments allows identification of key mechanisms that dictate the response of microbiomes to pH perturbations.

While the study is scientifically sound – measurements and models are well formulated and controlled – the insights are somewhat limited and it is not evident that the findings are as general as the authors propose.

Major issues:

1. An important claim of the study is that the finding of regimes is general, but this is insufficiently substantiated with a single site. Even if this might be credible for pH perturbations, it is much less clear for other environmental variables as the authors imply in the discussion. But even for pH, other environments with more acidic or basic conditions may have adapted communities that respond very differently.
2. The regimes themselves are somewhat difficult to interpret in terms of their relevance even at the site under study since their boundaries largely lie outside the natural pH variation. In fact, the natural pH variation is mostly encompassed by regime II and the natural short-term pH variations the authors mention are unlikely to push the system outside regime II. Additionally, regime I represents a death zone, which seems a somewhat trivial finding from the modelling perspective.
3. The claim that the study solves issues with previous interpretation of pH variation is difficult to judge based on the information provided. The evidence is presented as paper summaries in Tables S1 and S2 without a systematic review and leaves the question of whether different conditions and places may show different responses. Take for example, the suggestion that DNRA increases in regime III. In the literature, there are reports that this is due to increased C:N or higher pH which this study does not solve since these parameters covary.
4. The sequencing data are very superficially analyzed. The variability at the phylum levels seems very limited compared to the full range of variability found in natural habitats. Given that the authors find strong phylum level responses, or at least mostly emphasize phylum level responses in the main text, it is unclear how different initial communities with vastly different phylum level compositions would respond to pH perturbations. To strengthen the generalizability claim, it would be very helpful to compare the data in Figure S15 panel A to other published soil datasets and discuss how similarities or differences might influence the far-reaching conclusions proposed in the current study.
5. Resistance to chloramphenicol is widespread in soils, and the deactivation mechanisms may lead to breakdown of the antibiotic over time leading to population growth. The data presented in Figure S16, especially panel B raises serious concerns about the interpretation of CHL+ conditions as a growth arrested state compared to CHL- which includes growth. For example, soils 12, 14, and 15 only show a difference in Shannon diversity at very basic pH levels.
6. An alternative explanation to the results that may be relevant in some of the conditions tested is that pH does not perturb lineage abundances but rather the activity of the resident microbiome. This can be addressed with transcriptomics coupled to 16S taxonomic abundance measurements.
7. Because the authors measure microbial abundance only through DNA sequencing, and extracellular DNA can be very abundant in soils (leading to up to 3 orders of magnitude difference in CFU estimates compared to microscopy counting

techniques) the effect of pH on extracellular DNA for each phylum should be assayed.

8. The explanation for linear nitrate reduction that is given in the paper is internal carbon storage, but the authors fail to explain how internal C storage could arise under carbon limitation, which they postulate.

Minor issues:

Line 49: It is a bit of a leap to pull climate change into the argument since the paper is dealing with shortterm perturbations while climate change is gradual.

Line 139 – why were the specific 10 samples chosen for 16S rRNA sequencing?

Interpreting C, the limiting resource – The authors present evidence that C is cationic carbon sorbed to soil minerals. Should we think of this quantity, C as the average rC / rA of sorbed carbon, weighted by their abundance? Would we have to assume carbon, although variable and composed of numerous molecular species, is released with a similar r ratio across different pH perturbations?

Figure S7 - To understand the performance of the model better, it would be very helpful to plot the error in prediction vs measured nutrient level.

The role of death – For the linear consumption dynamics to hold, the death rate of the “functional population” must be very small compared to the standing population size. The authors should quantify and describe the role of death in their slurry experiments.

Diversity of nutrient limitation – How should we reconcile the idea that different lineages are limited by different nutrients at different levels? Does a mean-field model give rise to the author’s simple single population model in some way?

Fig. S11: The biomass increases within the 0-20 mM NaOH that is highlighted as the linear range is at best 2-fold. Is this consistent with the observed rates within that range? It is hard to tell from panel C where that range would fall.

Fig. 5A is difficult to glean the patterns due to the complexity of the dots and heatmap combination. The description of the dynamics in Fig. 5A seems overly simplistic since there appear to be much more nuanced responses.

Referee #2

(Remarks to the Author)

The authors present an extensive experimental and mathematical analysis of nitrate respiration dynamics in soils in response to perturbations of pH and nitrate amendments, over the time scales of a few days. Their mathematical model seems to capture these dynamics quite well, and offers insight into various mechanisms underlying the microbial community's nitrate respiration and growth dynamics. Overall the study tackles an important topic, and the experimental breadth is impressive! I did not notice major methodological issues. One broader concern that I have is that I found the story rather convoluted and hard to follow at times, with many moving parts and large logical jumps in the text. I also found it strange that the authors do not make their code available for review and do not provide accession numbers for their data.

Lines 81-85: I found the introduction of the model confusing. For example, it was not clear until much later that the model was a differential equation model, what the dynamic variables of the model were, and that pH essentially dictates the parameters and initial conditions of the model. In my view these key points would have been helpful to have earlier on. Further, in their introduction the authors say that the model only considers two variables, but their differential equation model (Fig. 2) has 3 dynamic variables (A, C and x). Further, on line 84 the authors also seem to use “parameters” interchangeably with “variables”, which further confuses the reader. These issues do not per se undermine the validity of their math, however there is considerably room for improving the clarity about their model early on.

Line 119: The authors mention that the sampled soils had “similar characteristics”, despite their pH ranging from 4.7 to 8.3, and they mention that variation in pH arose from agricultural practices and erosion. I'm surprised that soils covering such a wide range of pH would show otherwise similar characteristics. Wouldn't those agricultural practices and erosion that caused the variation in pH also likely cause other important properties of the soils to vary, thus leading to potentially confounding factors? For example, did the authors measure phosphorus (a common component of fertilizers) or moisture content in those sampled soils?

Line 151: Couldn't the growth also be limited top-down by phages or eukaryotic predators, rather than by other nutrients? In heavily top-down controlled populations increasing nutrient availability for the prey only has a weak effect on the prey population.

Line 886: Data associated with this manuscript will be made publicly available at NCBI
887 BioProject upon publication. Please clarify the accession numbers for the data. The data may be locked currently, but the accession numbers should be in the reviewed manuscript. This is also what the editorial policy checklist requests.

Line 888: I cannot access and therefore cannot assess the validity of the authors' code. In the editorial policy checklist the

authors say "Full code will be available before publication", but in the manuscript they say that it will be made available after publication. How can reviewers possibly check the code?

(Remarks on code availability)

I was unable to access the code! The authors have NOT made the code available. Their code availability statement says that they will supposedly make it available upon publication.

Referee #3

(Remarks to the Author)

Summary of findings & major comments:

Predicting the functional responses of microbial communities to changing environmental conditions is a critical challenge in microbial ecology. Here, Lee et al. address this problem by quantifying the response of soil microbiota to environmental shifts in terms of nitrate reduction at different pH levels. The authors performed these assessments at an unprecedented scale and across pH gradients (rather than a handful of conditions), which enabled them to unify observations across disparate studies of microbially-mediated nitrate reduction in soils. Specifically, the authors found that communities' nitrate reduction rates can be described according to three general levels and that these levels correspond quantitatively to shifts in community compositions and nutrient availability, both as a function of pH. Importantly, the authors develop a mathematical model that elegantly describes communities' nitrate reduction dynamics in terms of two key variables (essentially, how many cells are able to reduce nitrate and the nutrient levels available to them). This model provides a roadmap both for this specific system (in terms of how we might predict nitrate reduction in soils), as well as for microbial ecology generally (in terms of the power of identifying coarse-grained variables that can help us distill meaning from complex systems).

Overall, this is a very impressive body of work. I also strongly agree with the authors' philosophy that microbial ecology can, step by step, move from a descriptive to a predictive field by systematically and quantitatively characterizing microbial communities through combined experimentation and modeling, and I believe this manuscript is an excellent demonstration of this approach. However, before publication, I have some major comments/concerns to which I would appreciate the authors' responses.

Firstly, the significant advance to the field claimed by the authors is the use of their model to reveal the key mechanisms underlying the functional responses of soil microbial communities to environmental change. While I believe using the model is a powerful way to generate hypotheses about which coarse-grained variables underlie community function, I would appreciate an analysis of whether the model accurately captures a structure-function mapping in this system. If this model does capture the important underlying variables for this system, then it should predict nitrate reduction rates from the $x(0)$ and $C(0)$ values of a given community. It is remarkable that the inferred values of $C(0)$ are correlated with measurements of water-soluble organic carbon (WSOC) (Fig. 4C). What about the other key model parameter of biomass activity, $x(0)$? If you pair estimates of $x(0)$ from the microcosms with the WSOC measurements, can you predict the nitrate utilization dynamics? The authors have the data to try to estimate $x(0)$ from the sequencing-based absolute abundance estimates and the community compositions (e.g. through an estimate of functional biomass as used in Fig. S11, and/or through PICRUSt2 estimates of ASV's nitrate reduction potential as used in Fig. S18). I would appreciate it if the authors could perform this test of their model.

Secondly, I am skeptical of the evidence presented that the mechanism underlying Regime I is widespread death at low pH (lines 283-296). The death curves (Fig. S20A) look strikingly similar across diverse taxonomic groups, and the observation that members of the phylum Acidobacteriota are dying at low pH runs counter to many studies demonstrating that Acidobacteriota grow well in low pH soils (e.g. Jones ISMEJ 2009). While there tends to be lower bacterial biomass in low pH soils, the difference is not nearly as dramatic as in Fig. S20A (e.g. Rousk ISMEJ 2010). Instead, I have a concern that the 2-week wet-up of the soils at a high gravimetric water content with Milli-Q water, performed prior to setting up the pH perturbations, resulted in widespread cell lysis, with a bias in survival of Firmicutes (many of which are spore formers). This phenomenon has been observed previously (e.g. Blazewicz ISMEJ 2020). DNA is much less stable at acidic pHs than at neutral pH (e.g. An PLoS ONE 2014), so there is a chance that the "death" curves reflect the increased degradation of extracellular DNA at low pH of already-lysed organisms, rather than death due to low pH itself. It is of course still possible that there is widespread death due to low pH in these microcosms, but without further information, it is difficult to tell. Therefore, I recommend one of two options:

(i) Please demonstrate experimentally that there is widespread death in how you set up your microcosms at low pH (not just due to cell lysis upon wet-up followed by increased DNA degradation with decreasing pH). If the original soils are no longer available (which is very understandable), I think it would be okay to use other soils. A biomarker other than sequenced DNA would be convincing (though if you do use sequencing, I recommend extracting DNA from only the cellular fraction to avoid potential noise from extracellular DNA). Otherwise...

(ii) I recommend revising the language of your manuscript so that death is not the primary stated mechanism behind lower functional biomass activity at lower pH. There are numerous other mechanisms by which this could occur, and which precise mechanism it does not strongly affect your general conclusions. However, without such an abrupt biological difference between the low and neutral pH microcosms to further differentiate Regimes I and II, I would caution against emphasizing the qualitative separation between these regimes and would acknowledge that it is a continuum.

References for literature cited above:

Jones, R. T. et al. A comprehensive survey of soil acidobacterial diversity using pyrosequencing and clone library analyses.

The ISME Journal 3, 442–453 (2009).

Rousk, J. et al. Soil bacterial and fungal communities across a pH gradient in an arable soil. The ISME Journal 4, 1340–1351 (2010).

Blazewicz, S. J. et al. Taxon-specific microbial growth and mortality patterns reveal distinct temporal population responses to rewetting in a California grassland soil. The ISME Journal 14, 1520–1532 (2020).

An, R. et al. Non-Enzymatic Depurination of Nucleic Acids: Factors and Mechanisms. PLoS ONE 9, e115950 (2014).

Minor comments and suggestions:

I believe there are some instances in the discussion where it would be appropriate to temper the language and claims. For example:

- Lines 357-359: "This demonstrates that understanding the community response to perturbation may not require grappling with every metabolic process or interaction in the community, but only with a handful of key features." I believe it would be worthwhile to acknowledge (perhaps in the "Limitations" section, which I appreciate) that this statement applies to the authors' system likely because they focused on one metabolic function that is catalyzed by a single enzymatic step, and so it is perhaps easier to find a model here than for a process that is dependent on more metabolic steps and/or interspecies interactions.

- Lines 392-393: "... the metabolism of a single strain can mirror the metabolism of the soil microbiome." It is not surprising that there is a strain that has the same coarse-grained nitrate reduction rates as some communities, and the focus of this study is not on how to build up an understanding of microbial community function from the physiology of its individual members.

- Lines 394-398: "More broadly, the discovery of the three functional regimes, including the nutrient-limiting regime, is notable because it reflects a potential duality between the physiology of an ecosystem and the three phases of a cell: growth (Regime III), stationary (Regime II), and stress (Regime I). This duality suggests the possibility that cellular physiology might provide a conceptual framework for understanding the ecosystem." I understand that discussions are a space for large-scale thinking and speculation, but I believe this statement is a great oversimplification. Cells are experiencing all three of these phases across the functional regimes, and the relative contribution of each growth phase to the make-up of the cells in each regime is not quantified. (This statement is also in contradiction to the authors' own interpretation of the shift from Regime II to Regime III involving Regime II taxa experiencing stress; see line 377).

My apologies if you have included this information and I missed it: could you please provide values for the increase in the rate of nitrate reduction for all of the samples, indicating to which regime the sample belongs? Can the distribution of these values also be used to define the regimes (in addition to the thresholds for the inferred biomass activity and available limiting nutrient)? To what extent is there overlap in these values for Regimes I vs. II?

Fig. 3A: The y-axis is strange since it's not really on a log scale - the represented distance between $10e-2$ and $10e-1$ is much smaller than $10e-1$ to $10e0$ to $10e1$. Why did you choose to represent it like this?

Fig. 5D caption: It would be helpful to note that the points in Fig. 5D are a subset of those in Fig. 3A because only half of the samples were sequenced (rather than saying it is the same plot).

Fig. S7 (right): The model seems to be biased to have the highest error for the samples that are at the Regime II/III boundary. Could you please comment on the direction of the error and why this might be?

There are some instances where $x(0)$ is referred to as biomass (e.g. lines 182 and 193), rather than biomass activity, which was confusing on a first read. It would help with the clarity of the writing if the latter were used consistently.

(Remarks on code availability)

There is no code provided.

Version 3:

Reviewer comments:

Referee #1

(Remarks to the Author)

The authors have presented an impressive, expanded analysis in response to the criticisms voiced by the previous reviews. We appreciate that the regimes hold – at least qualitatively – at different sites and that the simple model can capture these dynamics. However, the analysis of community dynamics at the additional sites shows very high variation. While in Fig. 5, Pseudomonadota and Bacteriodota are lumped and presented as growing in regime II, Fig. 6 actually shows very high variation and much less clear trends. Pseudomonadota are highly variable and appear to also respond at high pH, and Bacteriodota mostly respond at high pH, contradicting the results presented in Fig. 5. Bacillota only appear to respond in 2 samples and appear to have a peak at low and high pH in the same samples. This high variation questions how the data are presented in Fig. 5 and the generality of the taxonomic analysis.

Other than that, we are happy with the changes made to the paper.

Referee #2

(Remarks to the Author)

The authors have addressed all of my concerns!

(Remarks on code availability)

The code seems well documented and well-structured. I have not attempted to run it.

Referee #3

(Remarks to the Author)

The authors have satisfactorily addressed my concerns and comments, and I appreciate the extensive additional experiments and analyses that the authors performed to strengthen their manuscript.

(Remarks on code availability)

All code is available and is reasonably readable/accessible to readers.

Version 4:

Reviewer comments:

Referee #1

(Remarks to the Author)

The additional changes have addressed our remaining concerns. We thank the authors for a comprehensive and easy to read response.

January 31, 2025

Dear Dr. Editor –

We would like to thank you and the referees for your thoughtful evaluation of our work. We have now revised the manuscript according to the queries from the referees. Below you will find a point-by-point response to the questions from the referees (responses in regular text, comments from referees in *italic*). In the revised manuscript changes are marked in red. Here we provide a high-level summary of the most impactful and substantial revisions that we made. We believe these revisions significantly improve our study, making it appropriate for publication in *Nature*.

Sincerely,

Sepp Kuehn (on behalf of all authors)

1. **Generality:** For the question of the generality of our study from referee #1 and the editors:
 - a. We performed the entire experiment on soils from three additional sites sampled from the midwest and California. These experiments include pH perturbations, dynamic measurements of metabolites, and sequencing. **These measurements confirm our finding of functional regimes.**
 - b. We re-analyzed a dataset from a previous study (Simek *et al.* 2002) performed on soils from the Czech Republic that enabled us to estimate functional biomass and nutrient levels and again **confirmed the existence of functional regimes.**
 - c. We performed a meta-analysis of more than nine studies spanning ~70 years from soils sampled across the globe and a spectrum of soil types. These studies confirmed the fundamental findings of our work: the decline of $x(0)$ at extreme acidic or basic pH, the increase of reduction rates with carbon amendment, the rapid rise of Firmicutes (Bacillota) in basic conditions, and linear dynamics of nitrogen compound utilization near native pH.
 - d. We replaced Figure 6 with a figure presenting the results from (a)-(c) and added a section to the results presenting these results.
2. **Death:** With regard to the question from referees #1 and #3 concerning cell death in Regime I. We performed the experiments proposed by both reviewers.
 - a. Isolated strains from our soils representative of the dominant phyla.
 - b. Measured the degradation of gDNA as a function of pH via qPCR on 16S rRNA genes.
 - c. Measured taxon-specific death rates as a function of pH for isolates from the dominant phyla in our experiment.

- d. From (b-c) developed a model that enabled us to show that death must contribute to the change in sequencing counts we observe.
 - e. Tempered the language in the manuscript to indicate that while death is involved in Regime I, other mechanisms may also be at play.
3. **Prediction:** Referee #3 asked if we could extend our method to enable predictions of nitrate dynamics from sequencing and carbon measurements. In an extensive new supplementary information section we show that such predictions do in fact work. This section increases the impact of our study substantially by quantitatively linking structure to function.
 4. **Readability:** Referee #2 pointed out that the flow of the manuscript was hard to follow. We undertook a careful revision of the manuscript paying close attention to the coupling between the presentation of the model, the explication of regimes, and the discussion of underlying mechanisms.

Note for reviewers: To align with the updated taxonomy (2021), we revised the manuscript to change 'Firmicutes' to 'Bacillota' and 'Proteobacteria' to 'Pseudomonadota' in the text and figures. However, for clarity and ease of communication, we have retained the terms 'Firmicutes' and 'Proteobacteria' in this response letter.

Referees' comments:

Referee #1 (Remarks to the Author):

The main aim of the study is to identify how natural microbial communities respond to environmental changes. The authors formulate and analyze a minimal consumer resource model based on high-throughput measurements of soil nutrient dynamics in the lab. The samples stem from 20 locations that vary in their initial pH from a single agricultural field, and soil slurries are subjected to pH perturbations with and without the addition of chloramphenicol, a translation inhibiting antibiotic that suppresses the growth of many lineages. The main claim is that the approach that combines modeling and experiments allows identification of key mechanisms that dictate the response of microbiomes to pH perturbations.

While the study is scientifically sound – measurements and models are well formulated and controlled – the insights are somewhat limited and it is not evident that the findings are as general as the authors propose.

Major issues:

Rev1-Comment1.1. *An important claim of the study is that the finding of regimes is general, but this is insufficiently substantiated with a single site.*

Response: We appreciate this comment from the referee and we recognize the importance of demonstrating the generality of our results in other soils. We have undertaken a serious and wide-ranging effort to test the generality of the regimes we observed in our original 20 soil samples. In the response that follows, we have addressed the question of generality with a range of approaches from (1) **new experiments** with additional soils, (2) **a meta-analysis** of

functional data in previous denitrification studies spanning many sites, and (3) analysis of **existing sequencing datasets**.

As the referee can see below, these results provide striking evidence for the generality of our findings and of our approach. **To present these results we have added an entirely new main text figure (new Fig. 6).**

1. New experiments with additional soils

From August of 2024, we collected topsoil samples from multiple additional sites spanning natural preserves in Illinois (LaBagh Woods, Orland grasslands, Moraine), Indiana (Pinhook Bog, Ambler Flatwoods), and the Sedgwick Reserve (CL, EL, SCL grassland samples) in California (managed by the University of California, Santa Barbara). We performed pH perturbation, dynamic measurements of metabolites, and 16S amplicon sequencing on the same scale as the experiments were done on the original 20 soils from our first submission (Cook Agronomy Farm (CAF)). We conducted analysis on four soils (LaBagh, Pinhook, CLG13, and ELG13) (see Methods: last paragraph in “Sample collection, site description, and soil characterization” for details).

Modeling reveals regimes in additional sites: We found that our model fits the nitrate dynamics on these soils well (see new Fig. S40 where the fitting error is reported). **The results confirmed our findings by revealing the same functional regimes** observed in the Cook Agronomy Farm (CAF) soils: the indigenous biomass activity ($\tilde{x}(0)$ parameter) decreasing and available limiting nutrient ($\tilde{C}(0)$ parameter) increasing during basic perturbations (shift towards Regime III), large indigenous biomass activity ($\tilde{x}(0)$) in Regime II, and diminishing $\tilde{x}(0)$ and $\tilde{C}(0)$ during acidic perturbations (shift towards Regime I) (new Fig. 6B, reproduced here). Note that these soils all display qualitatively similar trends in $\tilde{x}(0)$ and $\tilde{C}(0)$ as observed in Fig. 3A of the main text).

Reproduced from **Figure 6B, D** to facilitate reading. Refer to the main text figure for captions.

Sequencing reveals similar taxonomic response to pH perturbations in new soils: As with our original 20 soils, we used sequencing measurements to quantify growth in these four additional soil samples. When we performed dimension reduction on the phylum growth fold with NMF (non-negative matrix factorization, see Methods) on the new soils, **we observed the same 3 phyla capturing the structural variation during pH perturbations:** Proteobacteria (Pseudomonadota) and Bacteroidota (NMF Axis 2), Firmicutes (Bacillota) (NMF Axis 1) (new **Fig. S27**, captured here). New soils reproduced the same pattern of **Firmicutes (Bacillota) abundance increasing** during basic perturbations (Regime III) as we observed in the CAF soils (new **Fig. 6D**, captured here). These results strongly suggest that the finding of regimes is generalizable to other soils.

Reproduced from the new **Figure S27** to facilitate reading. Results of NMF (non-negative matrix factorization, see Methods) on growth fold for the new additional soils from IL, IN, and CA, USA.

2. Meta-analysis of functional data in previous studies confirms the existence of regimes

The referee correctly pointed out that our original Tables S1 and S2 were hard to understand and did not contain a clear articulation of how our results unified past studies of a similar nature. To remedy this we undertook a thorough meta-analysis of denitrification studies in the

past 70 years, which we summarize in the **new main text Fig. 6** and a **new Table S2**. Here we highlight the key findings of this meta-analysis.

First, we re-analyzed a dataset from a previous study (Simek *et al.*, 2002) performed on soils from the Czech Republic. Their measurements of early rates of denitrification enzyme activity (early DEA) correspond to our model parameter for indigenous biomass activity ($\bar{x}(0)$), as both measure the initial rate of denitrification. The difference between the early rates and the late rates measured as denitrification potential (DP) can be used as an estimate of the available nutrient levels $\tilde{C}(0)$ in our model, because the change of denitrification activity signals the growth of biomass activity from nutrients (see new **Fig. S26** and SI: “*Meta-analysis of functional data in previous denitrification studies in other soils*” for analysis details). Remarkably, these dynamics also adhere to **functional regimes at three different sites** with native soil pH of 4.8, 6.4, and 7.8. For all three soils, we qualitatively observe the same transitions between the three regimes under pH perturbations as seen in our $\bar{x}(0)$, $\tilde{C}(0)$ parameter space (Fig. 3A) in the scatter plots of early denitrification enzyme activity (DEA) (x-axis) and rate change (DP - DEA) (y-axis) (new **Fig. 6C**).

Second, we found that the decline in indigenous biomass activity $\bar{x}(0)$ under extreme acidic or basic pH perturbations away from the soil’s native pH (Fig. 3C, E) is **also observed in at least 10 other soils across four different studies** (Parkin *et al.*, 1985; Simek & Hopkins, 1999; Simek *et al.*, 2002; Khalifa & Folz, 2024). In these studies, denitrification enzyme activity (early DEA) measurements corresponded to indigenous biomass activity $\bar{x}(0)$ (y-axis in new **Fig. 6F**).

Third, as we observe in Regime II, we found linear dynamics of nitrogen compound utilization near the native pH in two studies from sites in Norway and the UK (Nömmik, 1956; Bremner & Shaw, 1958) (new **Fig. 6G**). In addition, when these authors amended their soils with carbon, they also observed an increase in reduction rates (new **Fig. 6G**), confirming nutrient limitation of denitrification activity in Regime II.

Taken together, this meta-analysis provides independent strong evidence for the fidelity of our modeling approach and the existence of regimes.

3. Comparison to existing sequencing datasets

In a study similar to ours Anderson *et al.*, 2018 perturbed a soil sample to different pH levels and sequenced the samples dynamically in time. In this study, as in ours, they observed a rapid rise of Firmicutes (Bacillota) abundance in basic perturbations (right two panels in new **Fig. 6E**) and growth of Bacteroidota during relatively acidic perturbations (left two panels in new **Fig. 6E**), as we saw in our CAF soils (**Fig. 5B**). **These results further confirm the taxonomically conserved response of soils to perturbations that we observe in our soils.**

Finally, to compare the initial soil microbiome composition of our CAF soils to other soils, we used the global soil microbiome dataset (Delgado-Baquerizo *et al.*, 2018) and showed that the CAF soils are representative of global soil microbial communities across diverse geographic regions and ecosystems in the NMDS embedding space (see **Rev1-Comment 4** for details).

Actions taken:

To summarize the results of the meta-analysis of more than 15 studies spanning ~70 years from soils sampled across the globe and a spectrum of soil types, we replaced the previous Fig. 6 with a figure to present these findings. Furthermore, in the new **Table S2**, we

documented how each study supports our results, along with details of soil characteristics and experiments. In light of this new figure we revised the Discussion section “unifying decades of prior work in a quantitative framework” to not be redundant with the main text. The following bullet points serve as a guide to where changes are reflected in the main text, supplementary text, figures, and throughout this response letter.

(1) New experiments with additional soils:

- Main text: we added in the Results section: “*Functional regimes generalize to other soils*” (Lines 326-354) describing how our experiments demonstrate the generality of our findings.
- Main text: In Methods, we added a paragraph at the end of the section “*Sample collection, site description, and soil characterization*” (Lines 542-553) to briefly explain what we did as described above.

(2) Meta-analysis of the functional data on denitrification from other soils:

- Main text: we added a results section: “*Functional regimes generalize to other soils*” describing how soils from other studies reproduced the fundamentals of different regimes (Lines 326-354).
- Supplementary Information: we added a section “*Meta-analysis of functional data in previous denitrification studies in other soils*” (Lines 1529-1547) to explain what we did as described above.
- **Rev1-Comment 3:** Meta-analysis of past literature summarized in new **Table S2**.

(3) Analysis of sequencing data of other soils from published data:

- See new **Fig. 6** and the associated caption.
- See also new **Fig. S25** for a comparative analysis of the global topsoil microbiome.
- **Rev1-Comment 4:** we analyzed global topsoil sequencing data from Delgado-Baquerizo *et al.* (2018) to show that our soils from a single site still capture much of the structural variation driven by natural pH variations.
- **Fig. 6E:** we analyzed sequencing data from Anderson *et al.* (2018) to show that, as we observed in CAF soils, Firmicutes (Bacillota) abundance increases during basic perturbations.

References:

Anderson, C. R., Peterson, M. E., Frampton, R. A., Bulman, S. R., Keenan, S., & Curtin, D. (2018). Rapid increases in soil pH solubilise organic matter, dramatically increase denitrification potential and strongly stimulate microorganisms from the Firmicutes phylum. *PeerJ*, 6, e6090.

Bremner, J. M., & Shaw, K. J. J. A. (1958). Denitrification in soil. II. Factors affecting denitrification. *The Journal of Agricultural Science*, 51(1), 40-52.

Delgado-Baquerizo, M., Oliverio, A. M., Brewer, T. E., Benavent-González, A., Eldridge, D. J., Bardgett, R. D., ... & Fierer, N. (2018). A global atlas of the dominant bacteria found in soil. *Science*, 359(6373), 320-325.

Khalifah, S., & Foltz, M. E. (2024). The ratio of denitrification end-products were influenced by soil pH and clay content across different texture classes in Oklahoma soils. *Frontiers in Soil Science*, 4, 1342986.

Nömmik, H. (1956). Investigations on denitrification in soil. *Acta agriculturae scandinavica*, 6(2), 195-228.

Parkin, T. B., Sexstone, A. J., & Tiedje, J. M. (1985). Adaptation of denitrifying populations to low soil pH. *Applied and Environmental Microbiology*, 49(5), 1053-1056.

Šimek, M., & Hopkins, D. W. (1999). Regulation of potential denitrification by soil pH in long-term fertilized arable soils. *Biology and Fertility of Soils*, 30, 41-47.

Šimek, M., Jíšová, L., & Hopkins, D. W. (2002). What is the so-called optimum pH for denitrification in soil?. *Soil Biology and Biochemistry*, 34(9), 1227-1234.

Rev1-Comment 1.2. *Even if this might be credible for pH perturbations, it is much less clear for other environmental variables as the authors imply in the discussion.*

Response: We agree that it is not entirely clear whether functional regimes might govern the response of soil microbiomes to other environmental perturbations. We regard this as an important avenue for future work. Also, a new paper published during our revision process shows conserved taxonomic and functional responses to extreme perturbations (Knight *et al.* Nature 2024). This study provides some evidence of conservation and simplicity in the response of complex communities to diverse perturbations.

Action taken: We have emphasized this more clearly in the Discussion section now titled “*Functional regimes and other environmental perturbations*” (Lines 415-423). We added a sentence and citation to Knight *et al.* Nature 2024 to the same section.

References: Knight, C.G., Nicolitch, O., Griffiths, R.I. *et al.* Soil microbiomes show consistent and predictable responses to extreme events. *Nature* 636, 690–696 (2024).

Rev1-Comment 1.3. *But even for pH, other environments with more acidic or basic conditions may have adapted communities that respond very differently.*

Response: There are several important points to make in response to this comment. First, it is certainly the case that for soils with pH well outside those measured here or analyzed in our meta-analysis the dynamics of nitrate utilization could depart from what we report here.

Second, however, is the fact that the native pH range of the soils studied here spans the range of pH values for topsoils observed in global topsoil surveys. As a result, we would argue that *most* topsoils are at a similar pH to those observed here. Specifically, our soils (and those in other studies) range in pH from about 4.8 to 8.3 (pH H₂O - measured 1:5, soil:water). A global topsoil dataset that we analyzed for the revision (Delgado-Baquerizo *et al.*, 2018) had a similar range of pH ranging from 4 to 9 (pH H₂O - measured 1:2.5 soil:water). Similarly, another recently published global topsoil microbiome survey (Bahram *et al.*, 2018) includes soils with a similar pH KCl range of 3 to 7 (see Extended Data Fig. 5 of Bahram). However, pH KCl values are systematically lower than pH H₂O values (due to ion exchange mechanisms) by approximately 1 pH unit (Thomas, 1996). Therefore, on a global scale, most topsoils have a pH range of approximately 4-8. As a result, the pH values of the soils we study are representative of the global topsoil surveys (also please refer to our new analysis results in response to your comment #4).

Lastly, pH perturbations in basic soils are challenging, and hence studying pH perturbations in soils with a pH higher than 8 is often impractical or impossible. Although we sampled CAF soils with native pH up to 8.32, we could not analyze Soils 19 (pH 8.23) and 20 (pH 8.32) due to

their strong buffering capacity, making them highly resistant to pH perturbations (see soil pH titration curves in **Fig. S31 A & F**). In contrast, Soil 18 (pH 7.68) was amenable to pH perturbation with up to 100 mM of H⁺ or OH⁻ addition. To test its limits during an earlier pilot experiment with one soil starting from native pH of 8, adding 500 mM of HCl reduced the pH only to 7, 1000 mM reduced it to 6, and over 2000 mM was required to lower the pH to 5. Such extreme acid or base additions make the biological relevance questionable. Consequently, for highly basic soils with extreme buffering capacity, the story of functional regimes due to pH perturbations is less valid. We have acknowledged this in the revision.

Action taken: To point this out to the reader, we have added these statements in the “*Limitations of the study*” section of the Discussion (Lines 397-404).

References:

Delgado-Baquerizo, M., Oliverio, A. M., Brewer, T. E., Benavent-González, A., Eldridge, D. J., Bardgett, R. D., ... & Fierer, N. (2018). A global atlas of the dominant bacteria found in soil. *Science*, 359(6373), 320-325.

Bahram, M., Hildebrand, F., Forslund, S. K., Anderson, J. L., Soudzilovskaia, N. A., Bodegom, P. M., ... & Bork, P. (2018). Structure and function of the global topsoil microbiome. *Nature*, 560(7717), 233-237.

Thomas, G. W. (1996). Soil pH and soil acidity. *Methods of soil analysis: part 3 chemical methods*, 5, 475-490.

Rev1-Comment 2.1. *The regimes themselves are somewhat difficult to interpret in terms of their relevance even at the site under study since their boundaries largely lie outside the natural pH variation. In fact, the natural pH variation is mostly encompassed by regime II and the natural short-term pH variations the authors mention are unlikely to push the system outside regime II.*

Response: We agree with the referee that natural fluctuations in soil pH are typically small enough that they do not drive regime shifts (e.g. Zhang *et al*, 2022; Zhao *et al.*, 2021). As a result, most pH fluctuations would move soils around within Regime II. In that case, our results show how those pH perturbations can be *quantitatively* expected to impact rates of respiration. We also agree that it is important to make the reader aware of this fact. However, for soils that are fertilized with urea or pasture soils subjected to urine perturbations, large increases in pH are relatively common, and these could drive transitions from Regime II to Regime III (e.g. Anderson *et al.*, 2018 for a summary).

More broadly, however, we would propose that functional regimes and the mechanisms that govern them teach us a bigger lesson about soils and complex microbiomes. In particular, our pH perturbations reveal how the extant structure of these very complex systems governs their response to perturbations. In doing so, these perturbations reveal the set of functional dynamics that are possible for the system to access given its structure. We believe this is a key insight given the vast complexity of these communities. In essence, even though the largest perturbations in our experiments do not occur every day in every soil, they provide us a lens through which we can understand the structure and its impact on the function of these systems.

Action taken: We have expanded the comment in the “*Limitations of the study*” section of the Discussion which already mentioned our unnaturally large perturbations of pH to emphasize this point more clearly (Lines 397-400). Second, we have added a statement at the end of the

first paragraph of the Discussion (Lines 391-393) stating the utility of perturbations to understand system structure and function.

References:

Zhang, X., Xiang, D. Q., Yang, C., Wu, W., & Liu, H. B. (2022). The spatial variability of temporal changes in soil pH affected by topography and fertilization. *Catena*, 218, 106586.

Zhao, K., Ma, B., Xu, Y., Stirling, E., & Xu, J. (2021). Light exposure mediates circadian rhythms of rhizosphere microbial communities. *The ISME Journal*, 15(9), 2655-2664.

Anderson, C. R. *et al.* Rapid increases in soil pH solubilise organic matter, dramatically increase denitrification potential and strongly stimulate microorganisms from the Firmicutes phylum. *PeerJ* 6, e6090 (2018).

Rev1-Comment 2.2. *Additionally, regime I represents a death zone, which seems a somewhat trivial finding from the modelling perspective.*

Response: We agree that the mechanism can be “trivial” at very low pH ($\text{pH} < 3$) because it is likely that death would occur for most organisms in extremely acidic environments. However, we believe the Acidic death regime (Regime I) is non-trivial for three reasons.

The first reason is that the pH at which the transition between Regime II and I reflects the long-term adaptation of the community. Our model reveals that the native pH of the soil determines when (at what pH) it shifts from Regime II to Regime I, suggesting there is long-term adaptation to native pH levels, and thus acquiring acid tolerance for different pH ranges. Our last section of the paper (Result section: “*Long-term soil pH defines regime boundaries*” and in the SI: “*Details of argument that long-term soil pH defines regime boundaries*”) goes into detail to demonstrate this phenomenon where the pH of death in Regime I depends on their native pH levels. See **Fig. S28** (formerly Fig. 6 in the main text).

The second reason that Regime I is non-trivial is the taxa-dependent response to low pH. Each taxon's varying susceptibility to low pH can differentially affect how much individuals contribute to the community's collective function (nitrate dynamics). We measure the survival-fold (**Fig.S20**) by comparing abundance at T0 to acid-perturbed CHL+ (chloramphenicol-treated) conditions. As **Fig. S20A** shows, **Firmicutes (Bacillota)** exhibit very different dynamics at low pH compared to all other phyla, namely they survive, while other phyla reduce in abundance apparently coherently across phyla. However, zooming in to finer taxonomic levels (e.g., family level), we find that death rates vary across pH in a family-dependent fashion. Thus, the degree of death across pH levels varied among families within each phylum, with families in Proteobacteria (Pseudomonadota) showing pronounced differences (new **Fig. S23**). This suggests that the response of taxa to acidic perturbations reflects physiology, again a non-trivial finding.

Reproduced from **Figure S23** in the revised manuscript to facilitate reading. Survival folds for seven major families within 3 phyla. Survival folds are computed by absolute abundance in CHL+ (endpoint) samples divided by the absolute abundance in T0 sample. We focused on samples under acidic perturbation in Soil16. Error bars are standard deviations of 3 slurry sample replicates. The black dotted line is the survival fold in the phylum level from Fig. S20A.

We sought to confirm this result experimentally. To independently measure death rates without using the sequencing data, we **isolated representatives from different phyla in our soils, we then performed time-series CFU death assays and confirmed these taxa-dependent death rates in pH 3, 3.5, 4, 5, 6, and 7.** (Fig. S21B in the revised manuscript, refer to the new SI section: “*Justifying widespread death as the mechanism of Regime I*”). These pH- and taxa-dependent death rates confirm that the process of death under acidic perturbations is not a trivial outcome of lowering pH but reflects to some extent physiology of the organisms involved.

The third reason that Regime I is non-trivial is the asymmetry between Regimes I and III. Both basic and acidic perturbations are toxic to bacteria (Russell & Dombrowski, 1980; Antoniou et al., 1990; Presser et al., 1997; Fernández-Calviño & Bååth, 2010). Therefore, one might naively expect death for both acidic and basic perturbations (Regimes I and III). However, we observe the death zone in Regime I and not in Regime III. We would argue that this asymmetry is a soil’s response to pH perturbations that also makes Regime I non-trivial.

Action taken: In response to this comment, your Comment #7, and Rev. 3’s comment #2, we added a new section (SI: “*Justifying widespread death as the mechanism of Regime I*”) (Lines 1547-1767) discussing death in Regime I. Here, we added the results from the new experiments from our death rate quantification (CFU assays) and DNA degradation rate (qPCR) in 6 different pH levels (new Fig. S21) as well as death rate modeling results (new Fig. S22). Family-level analysis of the survival folds was also added to the SI (new Fig. S23).

Reference:

Russell, J. B., & Dombrowski, D. B. (1980). Effect of pH on the efficiency of growth by pure cultures of rumen bacteria in continuous culture. *Applied and Environmental Microbiology*, 39(3), 604-610.

Antoniou, P., Hamilton, J., Koopman, B., Jain, R., Holloway, B., Lyberatos, G., & Svoronos, S. A. (1990). Effect of temperature and pH on the effective maximum specific growth rate of nitrifying bacteria. *Water research*, 24(1), 97-101.

Presser, K. A., Ratkowsky, D. A., & Ross, T. (1997). Modelling the growth rate of *Escherichia coli* as a function of pH and lactic acid concentration. *Applied and environmental microbiology*, 63(6), 2355-2360.

Fernández-Calviño, D., & Bååth, E. (2010). Growth response of the bacterial community to pH in soils differing in pH. *FEMS microbiology ecology*, 73(1), 149-156.

Rev1-Comment 3. *The claim that the study solves issues with previous interpretation of pH variation is difficult to judge based on the information provided. The evidence is presented as paper summaries in Tables S1 and S2 without a systematic review and leaves the question of whether different conditions and places may show different responses. Take for example, the suggestion that DNRA increases in regime III. In the literature, there are reports that this is due to increased C:N or higher pH which this study does not solve since these parameters covary.*

Response: We agree with the referee that Tables S1 and S2 were not sufficient to judge whether or not our study unified previous work on the topic of denitrification and pH. We regard this as a serious omission in our original submission and have therefore undertaken a comprehensive and systematic review of the past literature as it pertains to the present work. Please see also the response to Comment 1.1 above (our response “2. Meta-analysis of functional data”). With respect to DNRA and C:N ratio vs. pH, we agree that we have not solved the issue of causality to determine which of these environmental variables controls the dominance of DNRA vs denitrification. However, we were still able to separately examine the effects of C:N ratio and higher pH on nitrate reduction (i.e., transition from Regime II to III) through soil carbon amendment experiments in our study, as well as in 2 previous studies with similar amendment experiments (**Fig. 6G**). Thus, our claim is that our finding of regimes unified a host of previous studies under a common quantitative framework, with specific aspects of these studies now summarized quantitatively in the new **Fig. 6** and new **Table S2**.

Action taken: First, we have made an entirely new main text figure (new **Fig. 6**) which re-analyzes and summarizes data from previous studies on pH and denitrification. To accomplish this, we have extracted data from figures of papers going back to 1958 or used available data. As **Fig. 6** demonstrates, studies undertaken on soils across the globe either directly reveal the existence of similar functional regimes, or the hallmarks of functional regimes from the rise of Firmicutes (Bacillota) in alkaline conditions to the pH dependence of the indigenous biomass activity ($\bar{x}(0)$). Second, we have removed Tables S1 and S2 in favor of a **new Table S2** that more clearly summarizes the key findings of these studies. These results are described in detail in the new Results section: “Functional regimes generalize to other soils” (Lines 326-354). Finally, we have tempered the first paragraph of the Discussion section titled “Unifying prior work in a quantitative framework” (Lines 462-470) to reflect the fact that our approach cannot disambiguate all mechanisms underlying soil response to pH perturbations (e.g. C:N ratio versus pH controlling DNRA).

Rev1-Comment 4. *The sequencing data are very superficially analyzed. The variability at the phylum levels seems very limited compared to the full range of variability found in natural habitats. Given that the authors find strong phylum level responses, or at least mostly emphasize phylum level responses in the main text, it is unclear how different initial communities with vastly different phylum level compositions would respond to pH perturbations. To strengthen the generalizability claim, it would be very helpful to compare the data in Figure S15 panel A to other published soil datasets and discuss how similarities or differences might influence the far-reaching conclusions proposed in the current study.*

Response: We appreciate this comment from the referee and agree that it is an important point to think more carefully about the taxonomic composition of our soils with respect to the question of generality posed above. In particular, we appreciate the statement “*it would be very helpful to compare the data in Figure S15 panel A to other published soil datasets and discuss how similarities or differences might influence the far-reaching conclusions proposed in the current study.*” Therefore, it would be useful to compare the initial taxa composition (T0) of the Cook Agronomy Farm (CAF) soils, and the soils analyzed as part of this revision, with that of other sampled soils or a global soil dataset. We have performed this analysis that compares the taxonomic composition of our soils with a global topsoil microbiome dataset.

To do so, we plotted the relative abundance at the phylum level for the 10 Cook Agronomy Farm (CAF, WA, USA) soils at T0, and other sampled soils from LaBagh Woods (IL, USA), Pinhook Bog (IN, USA), and Sedgwick Preserves (CLG13 and ELG13 samples, CA, USA), averaging the phylum relative abundance across triplicates (**Fig S25A** in revised manuscript, captured here). We displayed their measured pH values (H₂O suspension method) on the x-axis. As a reference soil dataset, we used the global topsoil data from Delgado-Baquerizo *et al.* (2018), which were collected from diverse ecosystems (forests, grasslands, and shrublands) across eighteen countries and six continents. In the 273 samples, soil pH varied from 4 to 9, also measured by H₂O suspension method. To compare global topsoils with our dataset we undertook two analyses.

First, we visualized the phylum level composition of our soils with those of soils across a similar pH range in the global dataset. We find that our soils are broadly similar taxonomically to the global topsoil microbiome dataset (see panel A here and **Fig. S25** in the revision). We next sought to make this comparison more quantitative.

Reproduced from **Figure S25** in the revised manuscript to facilitate reading. Refer to the figure in the revised manuscript for captions.

To accomplish this, we performed non-metric multidimensional scaling (NMDS, Bray-Curtis distance matrix at the family level) on all soils in the global topsoil microbiome and the soils from our study. The NMDS plot illustrates that CAF soils (green) and our newly sampled soils (IL, IN, CA) are well-integrated within the broader diversity of global topsoils (gray, panel B above), suggesting that our soils span the first and largest NMDS dimension of global soil microbiome diversity across diverse geographic regions and ecosystems (new **Fig. S25B**, captured here). For example, these soils are not simply clustered around a single point within the NMDS plot. When we color the samples by soil pH, we observe a clear pH gradient, with acidic soils on the left and basic soils on the right, illustrating the known strong influence of pH on microbial community structure. The CAF soils align along this gradient horizontally in the NMDS embedding (new **Fig. S25C**, see above), capturing the pH trend observed in the global dataset and demonstrating their representativeness of the broader soil pH spectrum, in line with our response to comment 1.3.

However, we agree with the sentiment of the referee that it is important to discuss the limitations of the diversity found within our samples. In particular, looking at **Fig. S25C** we can see that our samples do not push into the very basic regime (soils around pH 9) and also the extreme acidic regime (dark navy, **Fig. S25C**). Indeed, during our revision, we found that some basic soils had such strong buffering capacity that regimes could not be accessed via pH perturbations. In our revision, we now mention this in the discussion section (Limitations). We believe this is important to point out to the readers in service of the question of generality.

Action taken: We performed the requested analysis of a global topsoil microbiome dataset and directly compared the diversity of our soils to the global topsoil microbiome (new **Fig.S25**). In the Discussion's Limitations section, we added sentences: "Third, while the taxonomic diversity of the soils studied here does span the diversity of soils globally to a reasonable extent, it may not represent the diversity of very basic soils of native pH above 8 or acidic soils of pH below 4 (Fig S25C). Moreover, basic soils above pH 8 with strong

buffering capacity resist pH changes, thus rendering functional regimes due to pH perturbations less applicable in these basic soils.” (Line 400-404)

Reference: Delgado-Baquerizo, M., Oliverio, A. M., Brewer, T. E., Benavent-González, A., Eldridge, D. J., Bardgett, R. D., ... & Fierer, N. (2018). A global atlas of the dominant bacteria found in soil. *Science*, 359(6373), 320-325.

Rev1-Comment 5. *Resistance to chloramphenicol is widespread in soils, and the deactivation mechanisms may lead to breakdown of the antibiotic over time leading to population growth. The data presented in Figure S16, especially panel B raises serious concerns about the interpretation of CHL+ conditions as a growth arrested state compared to CHL- which includes growth. For example, soils 12, 14, and 15 only show a difference in Shannon diversity at very basic pH levels.*

Response: We acknowledge the potential unwanted side effects of the chloramphenicol treatment and the challenges associated with resistance or breakdown. However, we believe there is very strong evidence that CHL+ dramatically inhibits *growth* in the system (see below).

First, concerning specifically Fig. S16: Recall that there is, in fact, very little growth in these microcosms in Regime I and II. Thus, we **respectfully disagree with the referee’s assessment that Shannon diversity in Fig. S16B is a cause for concern.** The lack of change in Shannon diversity over most of the pH range is in fact *consistent* with our understanding that growth is limited by a lack of carbon in the system. The diversity curves observed in Fig. S16 are shaped by a variety of mechanisms (e.g. acid-induced death, Regime 1). Among these, growth (arrestable by CHL) is only a minor contributor, except in Regime III. Thus, the fact that CHL- and CHL+ curves look so similar is not alarming, and should not be seen as an indication of the failure of CHL+ to inhibit growth. It is precisely the limited growth in all regimes except III that gives rise to the observation by the reviewer of changing diversity only in basic pH levels. (See also an additional comment below regarding total biomass.)

Separately from Fig. S16, let us now address the possibility of chloramphenicol resistance. For the purposes of our study, we would only be concerned by CHL resistance within the nitrate-reducing population, and only to the extent that the growth of any resistant taxa would increase the rate of nitrate reduction (which would then be incompatible with our model). However, this is directly measurable; and in our soil pH perturbation experiments, we did not observe any increase in nitrate reduction rates under CHL+ (chloramphenicol-treated) conditions. For example, if we look at the nitrate dynamics during pH perturbation experiments of a soil (**Fig. 1B**, main text), the slopes (derivatives) of the nitrate dynamics in CHL+ samples (red lines) are constant. In some instances, the slope slightly decreases at later time points (slopes becoming smaller and lines curving upward, e.g. 5th column, first row in panel B), suggesting that long-term exposure to CHL may slightly **decrease the nitrate reduction rates** of the denitrifying population. Crucially, we do not observe any cases where the rates increased through time (slopes becoming steeper and lines curving downward) in the presence of CHL. Thus, the mechanism of CHL degradation leading to the growth of nitrate reducers proposed by the referee is not supported by the data. Moreover, in every experiment, we observe a *faster* nitrate reduction rate in CHL- relative to CHL+ (except Regime I), which is very strong evidence that CHL is inhibiting growth.

Further, if CHL degradation or resistance were a serious issue one would expect a dose effect, where lower/higher levels of CHL result in different nitrate reduction dynamics. Therefore, we tested a higher dose of CHL and found no evidence to suggest it significantly influenced the results. If CHL is broken down enough to release the inhibition then this effect should be inhibited by increasing the concentration of CHL. Contradicting this, we see little to no difference in nitrate dynamics in soils with 1000 ppm vs 2500 ppm CHL (**Fig. S4 new B**), where 1000 ppm (1g/L) is the concentration we used for our pH perturbation experiments.

Reproduced from **Figure S4B** to facilitate reading. Comparing the nitrate dynamics (concentration, mM) under 1000 ppm and 2500 ppm of chloramphenicol. The soil used was LaBagh Woods soil (IL, USA, see Table S1).

Finally, given the challenges of assessing growth via Shannon diversity plots (**Fig. S16**), we plotted the endpoint total biomass (absolute abundance of each sample) of Soil 12, 14, and 15 obtained by sequencing, where the absolute abundance is acquired by using internal standards (**Fig. Rev1-5**). We observe higher total biomass by this metric in CHL- relative to CHL+, further supporting the claim that CHL inhibits growth. In addition, in samples with higher pH than the native pH (right of vertical dotted lines), we observe a gradual increase in total biomass in the CHL- samples (None column) up until the last or second-last highest pH level where the community becomes dominated by the Firmicutes (Bacillota). This gradual growth is due to the nutrient release mechanism associated with base addition in Regime II. This further supports the claim that CHL inhibits growth.

Figure Rev1-5 Endpoint total biomass (absolute abundance of each sample) at each perturbed pH in Soil12, 14, and 15. The total biomass is acquired by taking the total sequencing reads of each sample and dividing it by the reads from the spike-ins (internal standards). The vertical dotted line is the soil's native pH (H₂O method) at T₀. CHL: chloramphenicol-treated samples. None: no-chloramphenicol.

Action taken: To show that CHL is working as intended, we added the plot (**Fig. S4 new B**) showing the absence of the dose effect of CHL by comparing 1000 ppm vs. 2500 ppm to our original **Fig. S4**. We incorporated this argument into a new SI section: “*Effectiveness of chloramphenicol in preventing microbial growth in the soil*” (Lines 1209-1230) and referenced this section in the Methods section (Line 574).

Rev1-Comment 6. *An alternative explanation to the results that may be relevant in some of the conditions tested is that pH does not perturb lineage abundances but rather the activity of the resident microbiome. This can be addressed with transcriptomics coupled to 16S taxonomic abundance measurements.*

Response: This is an excellent point, and underscores the importance of clarity in the definition of the biomass activity $\bar{x}(t)$, which our original manuscript failed to emphasize appropriately. Our model and analysis indicate that the indigenous biomass activity ($\bar{x}(0)$) changes with pH. However, this need not be mediated by changes in specifically lineage abundances. As the reviewer correctly points out, changes in $\bar{x}(0)$ could also reflect enzyme expression, or physiological responses to changing pH. In Regime I, changes in $\bar{x}(0)$ likely do reflect the death of particular lineages (as confirmed by sequencing; see questions from

referees #2, #3 regarding this mechanism). However, in other regimes, changing $\bar{x}(0)$ may simply reflect the differing pH sensitivity of different lineages in the population. For example, increasing pH can give rise to a decline in $\bar{x}(0)$ if only a low-abundance lineage is capable of nitrate reduction at higher pH. One way to further interrogate these possibilities would be via proteomics or transcriptomics, and we indeed regard these efforts as useful directions for future work. However, they lie beyond the scope of the present manuscript. Here, our intent is to demonstrate the existence and generality of functional regimes, which are defined directly in terms of the **activity** $\bar{x}(t)$ and are fully supported by the data presented.

Action taken: We have made two edits to the description of the model to clarify this understanding. First, at the end of the Results section “*Simple consumer-resource model captures metabolite dynamics across pH perturbations*” (Lines 183-188) we have re-stated the interpretation of $\bar{x}(0)$ as. “*These parameters retain the same interpretations: $\bar{x}(0)$ reflects the indigenous biomass activity of all taxa that can perform nitrate reduction in a given condition, and $\gamma \bar{C}(0)$ the available limiting nutrient,...*”. Second, at the end of the section “*Model reveals functional regimes*” (Lines 203-207) we now state “*It is important to recognize that the pH-dependent changes in $\bar{x}(0)$ (Fig. 3C, E) reflect shifts in indigenous biomass activity. These changes in $\bar{x}(0)$ (Fig. 3C, E) reflect shifts in indigenous biomass activity. These changes may result from variations in the abundance of nitrate-reducing taxa, differential expression of relevant enzymes, or changes in enzymatic activity, all influenced by pH. These observations motivate us to harness the model to identify the mechanisms underlying functional regimes.*” We also paid special attention in revision to consistently distinguish “biomass activity ($\bar{x}(t)$)” from “functional biomass ($x(t)$),” to avoid confusion.

Rev1-Comment 7. *Because the authors measure microbial abundance only through DNA sequencing, and extracellular DNA can be very abundant in soils (leading to up to 3 orders of magnitude difference in CFU estimates compared to microscopy counting techniques) the effect of pH on extracellular DNA for each phylum should be assayed.*

Response: We share the concerns with the reviewer regarding the possibility of relic DNA contributing to our signal. Early on we attempted to use the photoreactive DNA-intercalating dye propidium monoazide (PMA) to remove the relic DNA (Carini *et al.*, 2016), but these efforts were discontinued due to (1) technical difficulties of achieving significant compositional difference between PMA+/- treatments, (2) possibility of PMA interfering with the quantification of internal standards (gDNA spike-ins), and (3) learning that PMA has varying relic DNA removal efficacy depending on soil matrices’ physicochemical properties (Bairoliya *et al.*, 2022).

Therefore, **we addressed this comment by performing the experiments requested by the reviewer by measuring the effect of pH on extracellular DNA.** We isolated strains from our soils from each phylum: Proteobacteria (Pseudomonadota), Bacteroidota, and Firmicutes (Bacillota) (3 phyla that correlated well with the regimes), and **added a new section in our manuscript to address this issue** (SI: “*Justifying widespread death as the mechanism of Regime I*”) (Lines 1547-1767). In this response, we will provide an abbreviated description of the experiments and analysis to show that despite DNA degradation in low pH, cell death is still prevalent in Regime I. (A similar comment was also made by Rev.3; see their Comment 2)

Nucleic acids are less stable and more prone to degradation at acidic pH than at neutral pH due to proton-mediated depurination (An *et al.*, 2014). Therefore, we experimentally and

quantitatively tested whether this decrease in sequence-based abundance during acidic perturbation is an artifact derived solely from the degradation of relic DNA in low pH rather than death. To do so, we (1) measured under varying pH conditions the degradation rates of DNA extracted from isolates belonging to three different phyla, (2) we measured the death rates of the same strains by performing time-series CFU assays in monocultures in defined media with varying pH levels, then (3) built a simple model (**Fig. S22D**) of cell death and DNA degradation to predict how much decrease in sequence-based abundance should we expect to see given the measured DNA degradation but without cell death at low pH. We show that without cell death the sequence-based abundance data cannot be explained in the Acidic death regime (Regime I).

Experiment:

1. Strain isolation: We aimed to isolate strains from the three major phyla that correlated with the regimes (Fig. 5): Proteobacteria (Pseudomonadota), Bacteroidota, and Firmicutes (Bacillota). Endpoint slurry samples from the pH perturbation experiments, stored in 25% glycerol, were thawed and streaked onto TSA and R2A plates adjusted to the slurry's endpoint pH. Next, the isolates' nitrate reduction capability was verified in an anaerobic chamber using succinate-defined media (SDM). Then, full 16S rRNA sequences of isolates were Sanger sequenced to compare with ASV V3-V4 sequences classified as Regime II and III strains in prior analysis. Isolates chosen for the following experiments were Proteobacteria (Pseudomonadota) strain JW70530 (*Massilia* sp., Regime II strain), Firmicutes (Bacillota) strain JW50604 (*Paenibacillus* sp., Regime III strain), and Bacteroidota strain sic0106 (*Chitinophaga* sp.).

2. qPCR quantification of DNA degradation rate at varying pH levels: To measure DNA degradation at different pH levels, we cultured strains from three phyla and extracted genomic DNA (gDNA). 10ng/ul of gDNA was incubated in pH 3, 3.5, 4, 5, 6, and 7 phosphate buffers at 25°C for 4 days, with samples collected over 9 time points and stored at -20°C. Using the same qPCR primers as in our 16S rRNA sequencing, we quantified DNA concentrations with standard curves constructed for each pH buffer ($R^2 = 0.97-0.99$). To account for positional effects, we included 10 gDNA ng/ul controls on every plate row. DNA degradation rate constants (δ) were calculated from log-transformed concentration data, revealing pH-specific trends (new **Fig. S21A**, captured here).

A

Reproduced from **Figure S21A** to facilitate reading. Refer to the figure for captions.

3. CFU measurements to quantify isolate's death rate at different pH levels: To determine death rates of three strains at different pH levels, we pre-cultured them in 1/10x TSB (aerobic), then succinate-defined medium (SDM, pH 7.3, aerobic), followed by anaerobic preculture in SDM (with the exception of *Chitinophaga* strain). After preculture, we incubated them in SDM with a phosphate buffer at pH levels of 3, 3.5, 4, 5, 6, and 7. Samples of 9-time points were taken over 4 days, serially diluted, and plated on R2A or 1/10x TSA agar for CFU counting. Colony counts were used to calculate death rate constants (d) via linear regression on log-transformed CFU data (**Fig. S21B**, captured here). This approach revealed pH-dependent death rate trends across pH.

B

Reproduced from **Figure S21B** to facilitate reading. Refer to the figure for captions.

Analysis:

1. Modeling cell death and DNA degradation to predict sequence abundance dynamics:

We modeled microbial biomass ($X(t)$, DNA ng) and extracellular DNA ($D(t)$, DNA ng) dynamics over time, where measured DNA ($M(t)$) with sequencing is the sum of X and D . Biomass dies at death rate (d), while extracellular DNA degrades at a rate (δ). Dead cells release DNA into the extracellular pool at a rate proportional to the biomass, described by a factor α (ng of DNA released per unit of biomass). Lastly, the f parameter is the fraction of relic DNA out of initial sequence reads (new **Fig. S22D**, captured here).

D

Biomass's DNA	$\dot{X}(t) = -dX(t),$
Extracellular DNA	$\dot{D}(t) = \alpha dX(t) - \delta D(t),$
16S measurement	$M(t) = X(t) + D(t)$
Fraction of relic DNA	$f = \frac{D(0)}{X(0) + D(0)}$

Reproduced from **Figure S22D** to facilitate reading. Refer to the figure for captions.

2. Testing the odds of no death against the sequence abundance data:

To test whether DNA degradation alone (without death) could explain sequence abundance changes, we simulated a no-death model ($d = 0$) using measured DNA degradation rates (δ) for each phylum and pH condition. The no-death model predictions of $M(t)$ had only one free parameter f , after normalizing $M(t)$ by the initial abundance $M(0)$. We superimposed the no-death model predictions for normalized $M(t=\text{endpoint})$ with varying f ($f = 0, 0.5, 1$) onto the family-level normalized absolute abundance data within each phylum from chloramphenicol-treated (CHL+) samples, where growth was inhibited. At neutral and mildly acidic pH (5–7), no significant deviations were observed between predictions and data, confirming that chloramphenicol does not induce cell death. However, in acidic conditions (pH 3–4), significant deviations of no death predictions from data for Proteobacteria (Pseudomonadota) and Bacteroidota indicated widespread acid-induced death, particularly at pH 3.5 and 4 ($p < 0.05$ in all f scenarios). Even at pH 3, where DNA degradation rates were highest, sequence abundance changes required death to explain the data (when $f=0.5$). These results **provide strong evidence of acid-induced death in Regime I** (new **Fig. S22C**, captured here).

Reproduced from **Figure S22C** to facilitate reading. Refer to the figure for captions.

3. Inferring the fraction of relic DNA (f) with the model and sequence data:

Because estimating f can provide insight into how much of the signal in the sequencing data is coming from the actual microbial biomass, we attempted to estimate the fraction of relic DNA (f) using our model (new **Fig. S24**) with experimentally measured death (d) and DNA degradation (δ) rates for Proteobacteria (Pseudomonadota). We fit the f parameter for each soil, family, and pH combination by minimizing the root-mean-squared error (RMSE) between model predictions and observed normalized endpoint abundances. Since f estimates are sensitive to the DNA release parameter (α), we fixed α at values between 0 and 1 (in 0.1 intervals). To summarize the f estimates for each soil, we computed the median f from the best fits (minimum RMSE) across varied α . The median fraction of relic DNA in different soils was estimated to range from 0 to approximately 0.4 (new **Fig. S24B**, captured here).

B

Reproduced from **Figure S24B** to facilitate reading. Refer to the figure for captions.

Action taken: In response to this comment, your Comment #2.2, and Rev. 3's Comment #2, we added a new section (SI: "*Justifying widespread death as the mechanism of Regime I*" (Lines 1547-1767) with new supplementary **Figs. S21, S22, S23, and S24**) confirming death in Regime I despite DNA degradation. In the main text, we added a paragraph addressing the possibility of degradation of relic DNA in soil (Lines 315-325).

Reference:

Carini, P., Marsden, P. J., Leff, J. W., Morgan, E. E., Strickland, M. S., & Fierer, N. (2016). Relic DNA is abundant in soil and obscures estimates of soil microbial diversity. *Nature microbiology*, 2(3), 1-6.

Bairoliya, S., Koh Zhi Xiang, J., & Cao, B. (2022). Extracellular DNA in environmental samples: Occurrence, extraction, quantification, and impact on microbial biodiversity assessment. *Applied and Environmental Microbiology*, 88(3), e01845-21.

An, R., Jia, Y., Wan, B., Zhang, Y., Dong, P., Li, J., & Liang, X. (2014). Non-enzymatic depurination of nucleic acids: factors and mechanisms. *PloS one*, 9(12), e115950.

Rev1-Comment 8. *The explanation for linear nitrate reduction that is given in the paper is internal carbon storage, but the authors fail to explain how internal C storage could arise under carbon limitation, which they postulate.*

Response: We thank the reviewer for pointing out this confusion. We agree that in our original submission, this point was not clearly spelled out. The way it is written makes it sound like there are internal carbon storage compounds that are utilized under carbon limitation. This is not what we intended to claim, instead, there is robust literature showing that this endogenous respiration arises either from the catabolism of intracellular components (e.g. protein, nucleic acids) or from cell death within the system. So the "internal C storage" is very likely not C-storage compounds such as glycogen but protein, RNA, or other components of biomass. We detail this argument in this response and then lay out revisions to communicate this to the reader.

First, when bacteria are growing in carbon-replete conditions they often store carbon compounds such as glycogen or polyhydroxybutyrate (Sekar *et al.*, 2020; Dawes & Ribbons, 1964). However, these carbon storage compounds are typically utilized very rapidly upon carbon limitation of growth (minutes to hours, Sekar *et al.*, 2020). After carbon storage compounds are utilized, protein is degraded by ATP-dependent proteases (e.g. clpXP) and the degraded protein is utilized for energy and new protein synthesis (Damerau & St John, 1993; Ribbons & Dawes, 1963). Similarly, RNA, which unlike DNA is a significant fraction of biomass, can be degraded during starvation (Boylen & Ensign, 1970, Dawes & Ribbons, 1964). The utilization of protein as an energy source during carbon limitation is phylogenetically widespread (Gronlund & Campbell, 1961). This process of utilizing cellular components in the presence of available electron acceptors is referred to as endogenous respiration (Dawes & Ribbons, 1964) or maintenance.

Second, endogenous respiration in carbon-limited soils is also a well-known phenomenon. For example, in soil microcosms Anderson and Domsch used measurements of endogenous respiration (CO₂ production rates and the carbon amendment required to sustain fixed respiration rates without growth) to measure microbiome biomass (Anderson *et al.*, 2018). This measurement is conceptually similar to our “indigenous biomass activity” $\bar{x}(0)$ which corresponds to measuring biomass in units of nitrate utilization. In the seminal work of Anderson and Domsch the source of electron donors for sustained respiration in carbon limitation was not known, but the physiological studies above point clearly to the degradation of cellular components on long-time scales.

Action taken: We heavily revised the section “*Physiological insights from constant utilization rates in nutrient-limited environments*” in the Discussion highlighting that maintenance or endogenous respiration can occur even in carbon-limited conditions via utilization of cellular components (Lines 452-457).

Reference:

Sekar, K., Linker, S. M., Nguyen, J., Grünhagen, A., Stocker, R., & Sauer, U. (2020). Bacterial glycogen provides short-term benefits in changing environments. *Applied and environmental microbiology*, 86(9), e00049-20.

Dawes, E. A., & Ribbons, D. W. (1964). Some aspects of the endogenous metabolism of bacteria. *Bacteriological Reviews*, 28(2), 126-149.

Damerau, K. E. I. T. H., & St John, A. C. (1993). Role of Clp protease subunits in degradation of carbon starvation proteins in *Escherichia coli*. *Journal of bacteriology*, 175(1), 53-63.

Ribbons, D. W., & Dawes, E. A. (1963). Environmental and growth conditions affecting the endogenous metabolism of bacteria. *Annals of the New York Academy of Sciences*, 102(3), 564-586.

Boylen, C. W., & Ensign, J. C. (1970). Intracellular substrates for endogenous metabolism during long-term starvation of rod and spherical cells of *Arthrobacter crystallopoietes*. *Journal of Bacteriology*, 103(3), 578-587.

Gronlund, A. F., & Campbell, J. J. R. (1961). Nitrogenous compounds as substrates for endogenous respiration in microorganisms. *Journal of Bacteriology*, 81(5), 721-724.

Anderson, C. R., Peterson, M. E., Frampton, R. A., Bulman, S. R., Keenan, S., & Curtin, D. (2018). Rapid increases in soil pH solubilise organic matter, dramatically increase denitrification potential and strongly stimulate microorganisms from the Firmicutes phylum. *PeerJ*, 6, e6090.

Minor issues:

Rev1-Minor 1. Line 49: *It is a bit of a leap to pull climate change into the argument since the paper is dealing with shortterm perturbations while climate change is gradual.*

Response: We agree that climate change is not the *most* obvious connection. However, we do believe this link is relevant and justified, because of the fact that we study soils that have a broad range of native pH. Changes in native pH (e.g., soil acidification events) are sustained over long timescales, so in effect, we are studying how variation in long-term environmental conditions alters the response of soils to short-term perturbations. This relationship is the subject of our original Fig. 6 (now moved to SI as **Fig. S28** in favor of generality supporting new Fig. 6), and we believe it is an aspect of our methodology that is important to highlight. As a result, we have chosen not to revise line 49. However, to clarify this point we took the following action below.

Action taken: In our revision of the section “*Long-term soil pH defines regime boundaries*” (Lines 356-360), we have more clearly explained the adaptation effect of long-term pH changes.

Rev1-Minor 2. Line 139 – *why were the specific 10 samples chosen for 16S rRNA sequencing?*

Response: We performed pH perturbation experiments on 20 soil samples. Soil19 and 20, which were neutral to slightly basic soils, were resistant to pH perturbations, resulting in fewer perturbed conditions. Soils 1 and 2, sampled from a depth of 0–10 cm, differed from the other soils, which were sampled from 10–20 cm depth. To better facilitate comparisons among the soils, we decided not to prioritize these four soils for sequencing. Instead, we opted to sequence a subset of 16 soils because (a) sequencing all of the soils (~2000 samples) would have required excessive time and effort, and (b) a subset of soils distributed across the pH gradient was deemed sufficient to explain the variation observed in the functional dynamics. We manually selected 12 soils from the 16 based on their native pH values, ensuring the pH values were not too similar to each other. Unfortunately, the sequencing of Soils 13 and 18 failed for technical reasons and the samples were not recoverable.

Action taken: We revised Lines 134-135 to express clearly the criteria for soil selection: “Finally, we selected 10 soils spread evenly across the native pH gradient and performed 16S rRNA amplicon sequencing before and after incubation”

Rev1-Minor 3. *Interpreting C, the limiting resource – The authors present evidence that C is cationic carbon sorbed to soil minerals. Should we think of this quantity, C as the average r_C / r_A of sorbed carbon, weighted by their abundance? Would we have to assume carbon, although variable and composed of numerous molecular species, is released with a similar r ratio across different pH perturbations?*

Response: These are insightful questions. We’ll answer the two questions sequentially.

(1) Should we think of C as a weighted sum of different r_C / r_A 's?

Yes, the reviewer's intuition overall is correct, though not precisely mathematically accurate. This may be a similar type of question as "effective one-biomass model vs many groups of denitrifiers", as you asked in comment **Rev1-Minor 6**. In principle, there are many. We approximate with an effective model, where there is only one functional biomass and limited by only one kind of "carbon". In a more microscopically accurate world, there are many species and many carbon compounds, C_i , with the consumption rate r_A/r_{C_i} . The many carbon equations can be mathematically reduced to one equation of only one effective carbon resource C whose r_A/r_C equals to the average r_A/r_{C_i} weighted by carbon abundance C_i .

$$C = \sum_i C_i$$

$$r_A/r_C = (\sum_i C_i r_A/r_{C_i}) / (\sum_i C_i)$$

(2) Do we have to assume that the carbon is released in all forms with similar ratios across pH perturbations?

We do not make this assumption. The $\tilde{C}(0)$ in our paper refers to $C(0)*r_A/r_C$, which is a function of pH. However, it is not necessary to keep r_A/r_C fixed and assume only $C(0)$ is a function of pH. Analogously, for $\tilde{x}(0)=x(0)*r_A$ as a function of pH, and x_0 and r_A can be functions of pH. It is not necessary to assume $x_0(\text{pH})$ and r_A fixed. We acknowledge that we need to be more precise in our language when referring to $\tilde{C}(0)$; using 'effective carbon' would be more appropriate than simply 'carbon'.

Action taken: We added a new section in the SI to show the mathematics of the generalized model and its reduction to our effective model (see SI: "*Justifying the effective 1-biomass model despite the diversity of denitrifying taxa*") (Lines 1768-1848), and referenced that SI section in the Methods (Lines 678-679).

Rev1-Minor 4. *Figure S7 - To understand the performance of the model better, it would be very helpful to plot the error in prediction vs measured nutrient level.*

Action taken: To better illustrate the performance of the model, we added a new panel **C** to show model prediction and observed nitrate data for one soil as an example in **Fig. S7C**.

Rev1-Minor 5. *The role of death – For the linear consumption dynamics to hold, the death rate of the “functional population” must be very small compared to the standing population size. The authors should quantify and describe the role of death in their slurry experiments.*

Response: The referee is correct that the death rate must be small for the functional population for linear dynamics of nitrate utilization. Thanks to the previous comment by the referee (**Rev1-Comment 7** and **Rev3-Comment 2**) we quantified death rates for pH values representative of Regime II and found them to be very low (**Fig. S21, S22**). In contrast, in the Acidic death regime (Regime I), under extreme acidic conditions (pH ~3) where death is rapid, the dynamics could still appear linear, albeit with a near-flat slope (slope ~0). (see **Fig. 1B**). In less extreme Regime I, around pH 4, we observe instances where the death rate is sufficiently high to be evident through diminishing slopes in the dynamics, as seen in both CHL+ and CHL- conditions in **Fig. 1B**.

Action taken: In response to this comment, your Comment #7, and Rev. 3's Comment #2, we quantified and described the role of death in our slurry experiments in the new SI section: "*Justifying widespread death as the mechanism of Regime I*" (Lines 1547-1767) with new supplementary **Figs. S21, S22, S23, and S24**).

Rev1-Minor 6. *Diversity of nutrient limitation – How should we reconcile the idea that different lineages are limited by different nutrients at different levels? Does a mean-field model give rise to the author's simple single population model in some way?*

Response: Thank you for prompting us to clarify this point. Our 1-biomass model is compatible with the microscopic diversity of denitrifying taxa; it should be seen as an effective model summarizing this diversity into a few key parameters. The precision of the experiment often does not allow us to resolve deviations likely attributable to the fact that the set of denitrifiers is not uniform. In rare instances (e.g., Soil 14), we observe cases where we can hypothesize that multiple carbon sources are present (see new **Fig. S36**), indicating a possible failure mode of the mean-field 1-biomass model. However, these deviations are very subtle, and the 1-biomass model provides an excellent summary of dynamics. This is now clarified in the Discussion section: Limitations of the study.

Action taken: We added a new section in the SI to comprehensively address the potential failure modes of the 1-biomass model, including your point where strains are limited by different nutrients at different levels, by simulating metabolite dynamics with multiple biomass. See the new SI section "*Justifying the effective 1-biomass model despite the diversity of denitrifying taxa*" (Lines 1768-1848).

Rev1-Minor 7. *Fig. S11: The biomass increases within the 0-20 mM NaOH that is highlighted as the linear range is at best 2-fold. Is this consistent with the observed rates within that range? It is hard to tell from panel C where that range would fall.*

Response: You are asking a very important question of whether biomass increase corresponds with rate increases. This question is the key to connecting our functionally learned variable x back to the sequencing data, directly tackling the question: "Who are the denitrifiers in the community?". In **Fig. S11B**, the y-axis, labeled as biomass increase, was the fold increase ($T9/T0$) of total biomass (the total sum of absolute abundance per sample). However, since not all biomass would participate in nitrate reduction (e.g. other strains could be doing fermentation), we will call the subset of the total biomass, which actually performs nitrate reduction (function), "functional biomass". This is the true $x(t)$ variable in our model.

Due to other strains growing without utilizing nitrate, we would expect the fold change in total biomass to poorly align with the fold increase of nitrate reduction rate on a 1:1 line in a scatter plot. Indeed, when plotted against each other, this was the case (**new Fig. S11C**). This showed too little change in the biomass to explain the rate change (data points lying below the 1:1 line). This motivates finding the functional biomass.

To find the functional biomass, we used the differential abundance analysis (described in Methods) to filter significantly enriched ASVs when comparing CHL- and CHL+ samples. Among these enriched ASVs, we removed ASVs that were also enriched in No-nitrate conditions (CHL- vs. CHL+), which we regarded as strains that grow without reducing nitrate. By aggregating the absolute abundance of the remaining ASVs, we calculated the functional

biomass. As expected, this results in an improved agreement of biomass fold increase and rate increase on a 1:1 line in the scatter plot (**new Fig. S11F**).

Action taken: To make a clearer distinction between total and functional biomass and to enhance the readability of Fig. S11, we made several changes to the figure and text. First, we changed “biomass” to “total biomass”, highlighting it in blue in **Fig. S11**, and contrasted “functional biomass” with red. We added total biomass increase vs. rate increase scatter plots in the new **Fig. S11C**, so that we can show whether the biomass increases are consistent with the observed rate increases, and thus motivate the finding of functional biomass. We added a cartoon for identifying functional biomass using no nitrate controls in **Fig. S11D and E** for a better understanding of the analysis. We plotted the functional biomass and rate relationship soil by soil in **Fig. S11F** so that we can make comparisons to the total biomass case for each soil. Accordingly, we modified the corresponding SI section (“*Connecting nitrate reduction rate increases to the fold change in functional biomass in Regime II*”) (Lines 1344-1376).

Rev1-Minor 8. *Fig. 5A is difficult to glean the patterns due to the complexity of the dots and heatmap combination. The description of the dynamics in Fig. 5A seems overly simplistic since there appear to be much more nuanced responses.*

Response: We interpret “much more nuanced responses” as the bands of blue dots at the Regime I-II boundary AND the band at the II - III boundary. We agree that these boundaries contain subtleties that are not captured by a heatmap.

Action taken: To better illustrate these nuanced responses, we plotted Proteobacteria (Pseudomonadota) and Bacteroidota separately, as well as for each soil individually (**new Fig. S41**).

Referee #2 (Remarks to the Author):

The authors present an extensive experimental and mathematical analysis of nitrate respiration dynamics in soils in response to perturbations of pH and nitrate amendments, over the time scales of a few days. Their mathematical model seems to capture these dynamics quite well, and offers insight into various mechanisms underlying the microbial community's nitrate respiration and growth dynamics. Overall the study tackles an important topic, and the experimental breadth is impressive! I did not notice major methodological issues.

Response: We thank the referee for the careful review of our work and for recognizing the value of the model and the extent of the experiments performed.

Rev2-Comment 1. *One broader concern that I have is that I found the story rather convoluted and hard to follow at times, with many moving parts and large logical jumps in the text. I also found it strange that the authors do not make their code available for review and do not provide accession numbers for their data.*

Response: We apologize for any difficulty in following the story. On re-reading our manuscript we agree with the reviewer that the story was challenging to follow. You will find a heavily revised manuscript (in red) with the flow and logic of the argument in mind. We did our best to keep this comment in mind during revisions, editing many sections for clarity, particularly the model presentation (more clearly distinguishing between variables and parameters, as suggested; editing **Fig. 2**, etc.; see the file with changes highlighted), and addressing all your specific comments below. Regarding the code availability, there was a miscommunication between us and the editorial staff during submission that resulted in the link to the code repository not being transmitted to the reviewers with the first version of the manuscript. We apologize for this mix-up and will ensure that the referee has access to the code for this round of revision.

Raw sequence reads associated with this manuscript are deposited under NCBI BioProject ID **PRJNA1205727**. Codes and data tables associated with this manuscript are deposited at the [Open Science Framework](https://osf.io/ctf8k/) (<https://osf.io/ctf8k/>).

Rev2-Comment 2. *Lines 81-85: I found the introduction of the model confusing. For example, it was not clear until much later that the model was a differential equation model, what the dynamic variables of the model were, and that pH essentially dictates the parameters and initial conditions of the model. In my view these key points would have been helpful to have earlier on. Further, in their introduction the authors say that the model only considers two variables, but their differential equation model (Fig. 2) has 3 dynamic variables (A, C and x). Further, on line 84 the authors also seem to use "parameters" interchangeably with "variables", which further confuses the reader. These issues do not per se undermine the validity of their math, however there is considerably room for improving the clarify about their model early on.*

Response: This was a significant oversight on our part in the first draft of the manuscript which we agree could confuse many readers. We have made revisions to address this confusion.

Action taken: We have heavily revised the paragraph around line 84 in the original submission to clarify (1) that the model is dynamical (differential equation) and (2) the difference between a

dynamic variable in the model and a parameter. We also corrected several instances where we said “variable” but meant “parameter”. We have also clarified that pH perturbations are reflected as changes in *parameters* in the model (Lines 74-77).

To further clarify and reduce confusion among readers, we attached a box table below the original **Fig. 2** to show the definitions of every parameter, variable, and rescaled parameters and variables.

Rev2-Comment 3. *Line 119: The authors mention that the sampled soils had “similar characteristics”, despite their pH ranging from 4.7 to 8.3, and they mention that variation in pH arose from agricultural practices and erosion. I’m surprised that soils covering such a wide range of pH would show otherwise similar characteristics. Wouldn’t those agricultural practices and erosion that caused the variation in pH also likely cause other important properties of the soils to vary, thus leading to potentially confounding factors? For example, did the authors measure phosphorus (a common component of fertilizers) or moisture content in those sampled soils?*

Response: We reported all of the measured soil characteristics in Fig. S23 of the original submission (**Fig. S31** in the revised manuscript). As the referee suspects, phosphorus availability does change across pH values (**Fig. S31D**), which we did mention previously in our SI: “pH titration curves and physicochemical properties across soils” (previously the section name was “pH titration curves and soil’s native pH are shaped by soil’s physicochemical properties”). Along with P, we also showed that cation exchange capacity, sulfur, calcium ion, aluminum, and iron are highly correlated with the soil pH values (**Fig. S31D**). However, other key parameters of the soil such as clay:silt:sand ratios and moisture content are stable across pH values. This comment made us realize that we were not specific enough with our terminology when we referred to soil “characteristics”. Our revision reflects this realization. We also note that our amendment experiment indicates that phosphorus availability in the soils does not determine nitrate reduction rates (**Fig. 4E**).

Action taken: To highlight how other important properties of the soils can vary with soil pH, we added the statement regarding soil characteristics at the beginning of the Results section: “While 20 CAF sites had similar soil texture (silty clay loam) (Table S1, **Fig. 6A**), their variation in soil pH correlated well with cation exchange capacity, sulfur, and phosphorus levels (Fig. S31D)” (Lines 115-118). We made a new supplementary table (new **Table S3**) that has all the information about soil’s physicochemical properties and visualized the soil texture in the new **Fig. 6A**.

Rev2-Comment 4. *Line 151: Couldn’t the growth also be limited top-down by phages or eukaryotic predators, rather than by other nutrients? In heavily top-down controlled populations increasing nutrient availability for the prey only has a weak effect on the prey population.*

Response: We considered these alternate hypotheses during the analysis of our data, but the following experiments led us to reject these proposals. First, the top-down limitation of bacterial growth by phages is not consistent with our nutrient amendment experiment (**Fig. 4E**). If phages were the dominant contributor to limiting the growth of bacteria, then amending nutrients should not affect growth. Similarly, top-down control by eukaryotic predators should be impacted in microcosms that are treated with cycloheximide (a eukaryotic inhibitor, **Fig. S4**).

However, this comment made us realize that we did not clearly articulate these results to the reader, thus we made the following changes.

Action taken: First, the text previously at line 151 that the reviewer noted has been changed to read: "...suggests that growth is limited by something other than nitrate, potentially other nutrients (schematic, Fig. 1C)." (Lines 149-150). Second, we added a statement regarding the evidence against the top-down regulation of bacterial growth by phage or eukaryotes at the end of the section "*Metabolite dynamics in Regime II are governed by carbon release*" (Lines 257-259).

Rev2-Comment 5. Line 886: *Data associated with this manuscript will be made publicly available at NCBI887 BioProject upon publication. Please clarify the accession numbers for the data. The data may be locked currently, but the accession numbers should be in the reviewed manuscript. This is also what the editorial policy checklist requests.*

Response: We apologize for the omission (see our comment above regarding the confusion around data availability). We added the accession number (NCBI BioProject ID PRJNA1205727) for the raw data to the manuscript and also Open Science Framework link for processed data tables.

Line 888: I cannot access and therefore cannot assess the validity of the authors' code. In the editorial policy checklist the authors say "Full code will be available before publication", but in the manuscript they say that it will be made available after publication. How can reviewers possibly check the code?

Response: We sincerely apologize (see comment above). All codes for modeling, analysis, and visualization are stored in the Open Science Framework (<https://osf.io/ctf8k/>).

Referee #2 (Remarks on code availability):

I was unable to access the code! The authors have NOT made the code available. Their code availability statement says that they will supposedly make it available upon publication.

Response: We sincerely apologize. See above.

Referee #3 (Remarks to the Author):

Summary of findings & major comments:

Predicting the functional responses of microbial communities to changing environmental conditions is a critical challenge in microbial ecology. Here, Lee et al. address this problem by quantifying the response of soil microbiota to environmental shifts in terms of nitrate reduction at different pH levels. The authors performed these assessments at an unprecedented scale and across pH gradients (rather than a handful of conditions), which enabled them to unify observations across disparate studies of microbially-mediated nitrate reduction in soils. Specifically, the authors found that communities' nitrate reduction rates can be described according to three general levels and that these levels correspond quantitatively to shifts in community compositions and nutrient availability, both as a function of pH. Importantly, the authors develop a mathematical model that elegantly describes communities' nitrate reduction dynamics in terms of two key variables (essentially, how many cells are able to reduce nitrate and the nutrient levels available to them). This model provides a roadmap both for this specific system (in terms of how we might predict nitrate reduction in soils), as well as for microbial ecology generally (in terms of the power of identifying coarse-grained variables that can help us distill meaning from complex systems).

Overall, this is a very impressive body of work. I also strongly agree with the authors' philosophy that microbial ecology can, step by step, move from a descriptive to a predictive field by systematically and quantitatively characterizing microbial communities through combined experimentation and modeling, and I believe this manuscript is an excellent demonstration of this approach. However, before publication, I have some major comments/concerns to which I would appreciate the authors' responses.

Response: We appreciate the reviewer's perspective on the work, and more importantly, on the value of the approach we have taken here for moving the field forward. We especially appreciate that the reviewer recognizes the value of a close dialogue between theory and experiment.

Rev3-Comment 1. *Firstly, the significant advance to the field claimed by the authors is the use of their model to reveal the key mechanisms underlying the functional responses of soil microbial communities to environmental change. While I believe using the model is a powerful way to generate hypotheses about which coarse-grained variables underlie community function, I would appreciate an analysis of whether the model accurately captures a structure-function mapping in this system. If this model does capture the important underlying variables for this system, then it should predict nitrate reduction rates from the $x(0)$ and $C(0)$ values of a given community. It is remarkable that the inferred values of $C(0)$ are correlated with measurements of water-soluble organic carbon (WSOC) (Fig. 4C). What about the other key model parameter of biomass activity, $x(0)$? If you pair estimates of $x(0)$ from the microcosms with the WSOC measurements, can you predict the nitrate utilization dynamics? The authors have the data to try to estimate $x(0)$ from the sequencing-based absolute abundance estimates and the community compositions (e.g. through an estimate of functional biomass as used in Fig. S11, and/or through PICRUSt2 estimates of ASV's nitrate reduction potential as used in Fig. S18). I would appreciate it if the authors could perform this test of their model.*

Response: We very much appreciate this comment as it made us think more deeply about what our model can reveal about the dynamics within our microcosms. We also agree that interrogating the predictive capacity of our model strengthens our paper substantially.

Building on the reviewer's suggestion, we added an additional step to estimate the two parameters r_A and r_C , representing the consumption rates of A and C per unit biomass. These two parameters are not measured in our experiment. However, by separating our samples into 50% training dataset and 50% test dataset, we applied machine learning methods on the training dataset to learn the parameters r_A and r_C (Fig. S38B, captured here). Using the learned r_A and r_C , together with the estimated $x(0)$ and $C(0)$, the model can predict nitrate dynamics $A(t)$ in the test set.

Reproduced from Fig. S37 to facilitate reading. **Candidates for feature selection to estimate $x(0)$ and $C(0)$ from sequence data and WSOC (water-soluble organic carbon) measurements.** (A) Scatter plots of $\tilde{x}(0)$ against: (1) total biomass (far left), (2) summed abundance of significantly enriched ASVs relative to CHL+ (second column), (3) summed abundance of significantly enriched ASVs relative to T0 (third column), and (4) summed abundances of ASVs with Nar and Nap genes inferred from PICRUSt2 (far right). (B) Scatter plots of $\tilde{C}(0)$ versus WSOC measurements in CHL+ or CHL- samples at T0 or T9 (endpoint), with colors representing endpoint perturbed pH. Note the log scale.

Here is the summary of our approach (see SI section: “*Prediction of nitrate dynamics from community structure*” for details). (Lines 1849-1928).

Step 1 & 2: (step 1) First, we evaluated which measurement—1) total biomass, 2) functional biomass (aggregated abundance of strains that are significantly enriched in endpoint absolute abundance (CHL-) relative to CHL+), or 3) functional gene abundances (as inferred via PICRUSt2)—provides the best estimation of $x(0)$. To do this we plotted quantities against $\log(\tilde{x}(0))$ since this must correlate with $x(0)$. These four quantities are plotted against the

inferred $\log(\tilde{x}(0))$ in panel A above). By comparing the correlations of each measurement to our inferred $\tilde{x}(0)$, we found that the 3) functional gene abundance was the best where we chose the abundance of Nar+Nap genes through PICRUSt2 analysis as the model parameter $x(0)$ (new **Fig. S37A**). (step 2) Similarly, we evaluated which measurements best correlate with our inferred $\tilde{C}(0)$ identifying WSOC measurements in CHL- samples at T9 as the model parameter $C(0)$ (new **Fig. S37B**, see above).

Step 3: We used the training dataset to learn the parameters r_A and r_A/r_C by using our estimates of $x(0)$ and $C(0)$ from **Steps 1 and 2** and our measurements of $\tilde{x}(0)$ and $\tilde{C}(0)$ from nitrate dynamics (and the fact that $r_A = \tilde{x}(0)/x(0)$ and $r_A/r_C = \tilde{C}(0)/C(0)$). We interrogated two models. (Model 1) which assumed constant values for r_A and r_A/r_C across all soils. (Model 2) To reflect the effects of the environment on structure-function mapping, we assumed the two parameters to be pH-dependent: $r_A(\text{pH})$ and $r_A/r_C(\text{pH})$ (r_A , r_C pH-dependent model in **Fig. S38B**). Using the Gaussian kernel regression (detailed in SI), to learn a smooth function $r_A(\text{pH})$. Similarly, for $r_A/r_C(\text{pH})$, we looked at the training sample distribution in pH vs $\tilde{C}(0)/C(0)$ space and learned a smooth function. To compare with the r_A , r_A/r_C pH-dependent model, we also tested a null model that bypasses r_A and r_A/r_C and just uses pH to predict $\tilde{x}(0)$ and $\tilde{C}(0)$ (**Fig. S38C**).

Reproduced from **Fig. S38** to facilitate reading. **Three alternative models for structure-function mapping.** **(A)** Fixed r_A and r_C model: The top panels show the log-distributions of r_A and r_A/r_C , computed as $\log(\tilde{x}(0) / x(0))$ and $\log(\tilde{C}(0) / C(0))$, with dashed lines and shaded areas indicating the mean and 0.5 standard deviation. The bottom panels display the learned $\tilde{x}(0)$ and $\tilde{C}(0)$ along with their uncertainty (purple curves and shades). **(B)** pH-dependent r_A and r_C model: The y-axes represent $\log(r_A)$ and $\log(r_A/r_C)$, calculated as $\log(\tilde{x}(0) / x(0))$ and $\log(\tilde{C}(0) / C(0))$. Orange curves and shades indicate the learned functions $\log(r_A) = f_1(\text{pH})$ and $\log(r_A/r_C) = f_2(\text{pH})$, along with learning uncertainty. **(C)** Null pH model: Yellow curves and shades show the learned $\log(\tilde{x}(0))$ and $\log(\tilde{C}(0))$ as pH-dependent functions with associated uncertainty.

Step 4: Compare to data and null models. For any sample in the test dataset, we used the measured Nar+Nap gene abundance as $x(0)$ in the model, the measured WSOC as $C(0)$ in the model, and the perturbed pH in the learned $r_A(\text{pH})$ and $r_A/r_C(\text{pH})$ function to get the consumption rates r_A and r_A/r_C in the model. Using these inferred parameters, we simulated

the consumer resource model and predicted the nitrate dynamics $A(t)$ for each test sample. We then compared our predicted $A(t)$ to the experimental measurements, our prediction performed well across all three Regimes, with only 0.28mM in root-mean-squared-error (RMSE) (third row of **Fig. S39B**, captured here). This was better than the other 2 null models (RMSE = 0.34, 0.32). In summary, our model provided accurate predictions of nitrate utilization dynamics through structure-function mapping.

Reproduced from **Fig. S39** to facilitate reading. **Comparing prediction results from the three models for structure-function mapping.** **(A)** Example fits using the effective 1-biomass consumer-resource model (first row) and prediction models (last three rows) across three regimes (columns). The pH-dependent model (orange, third row) shows the best prediction performance, closely matching the fit of the effective 1-biomass model (blue, first row). **(B)** Fitting and prediction errors for the training (left) and test (right) datasets. The pH-dependent model (orange) has the lowest errors among all models.

Action taken: We added a new section in the SI called “*Prediction of nitrate dynamics from community structure*” (Lines 1849-1928), following the reviewer’s suggestion to test the structure-function mapping. Given the length of the manuscript, and the main text revisions associated with establishing generality, we felt we could not include a new section in the Results portion of the manuscript. Therefore, we also added a brief Discussion of these predictions to the Discussion section “*Functional regimes as guides for understanding complex omics data*” to guide the reader to this section (Lines 429-436)

Rev3-Comment 2. *Secondly, I am skeptical of the evidence presented that the mechanism underlying Regime I is widespread death at low pH (Lines 283-296).*

Response: This is an important comment also raised by reviewer #1. We have broken your comments out into individual questions and responded to each in turn. We have undertaken a serious experimental effort to address these questions.

Rev3-Comment 2.1. *The death curves (Fig. S20A) look strikingly similar across diverse taxonomic groups, and the observation that members of the phylum Acidobacteriota are dying at low pH runs counter to many studies demonstrating that Acidobacteriota grow well in low pH soils (e.g. Jones ISMEJ 2009). While there tends to be lower bacterial biomass in low pH soils, the difference is not nearly as dramatic as in Fig. S20A (e.g. Rousk ISMEJ 2010).*

Response: The key difference between the study of Jones *et al.* (2009) and ours is the presence of short-term pH perturbations. Jones and colleagues measure the abundance of Acidobacteriota *without* perturbing soil pH. Therefore, the appropriate comparison between our paper and theirs is the relative abundance (since this is what Jones measures) of Acidobacteria at the native pH in our soils. **When we compute this quantity we observe, as expected, a rise in Acidobacteria abundances with declining pH (Fig. Rev3-2-1A below).** This result qualitatively agrees with Figure 1 of Jones *et al.* We conclude that reductions in Acidobacteria abundances in response to **short-term pH perturbations** represent a distinct process from the relative abundance of this phylum as a function of long-term soil pH.

To further explore this, we noted that Jones *et al.* observe some subgroups of the Acidobacteria that actually decline in relative abundance and others that rise with the native pH of the soil. To see if this was also the case in our soils we looked at two classes that belong to the Acidobacteriota phylum: Acidobacteriae and Vicinamibacteria, which account for most of the relative abundance in the Acidobacteriota phylum. In terms of long-term pH variation, they show distinct preferences for native soil pH (**Fig. Rev3-2-1B** below): Acidobacteriae having high abundance in low pH while Vicinamibacteria doing better at neutral pH. This phenomenon was also well described for many Acidobacteriota subgroups in Jones *et al.* (ISME J, 2009). However, when we impose **short-term pH perturbations**, despite their differing long-term pH preferences, **the two classes exhibit similar behavior where their abundances decrease under both acidic and basic perturbations (Fig. Rev3-2-1C, D below).** These results illustrate how microbes respond differently to long-term pH variations versus short-term pH perturbations, thereby biologically explaining the decreasing trend (towards low pH) we observed in **Fig. S20A**.

Figure Rev3-2-1. Scatter plots show the relative abundance of Acidobacteriota plotted against the soil's native pH or short-term perturbed pH. **(A)** The relationship between the soil's native pH (x-axis) and the relative abundance (y-axis) of the Acidobacteriota phylum at time point T0, prior to any short-term pH perturbations. Data points represent triplicates, and the dashed line represents the linear fit. **(B)** The same was plotted for two classes Acidobacteriae and Vicinamibacteria, within Acidobacteriota at time point T0 before any short-term pH perturbations. **(C)** Relative abundance (y-axis) of Class Acidobacteriae across different perturbed soil pH values (x-axis) at T9 (perturbed endpoint). **(D)** Relative

abundance (y-axis) of Vicinamibacteria across different perturbed soil pH values (x-axis) at T9 (perturbed endpoint). Each facet represents a specific soil. Vertical dotted lines in (C) and (D) indicate the soil's native pH for each corresponding soil.

Rev3-Comment 2.2. *Instead, I have a concern that the 2-week wet-up of the soils at a high gravimetric water content with Milli-Q water, performed prior to setting up the pH perturbations, resulted in widespread cell lysis, with a bias in survival of Firmicutes (Bacillota) (many of which are spore formers). This phenomenon has been observed previously (e.g. Blazewicz ISMEJ 2020).*

DNA is much less stable at acidic pHs than at neutral pH (e.g. An PLoS ONE 2014), so there is a chance that the “death” curves reflect the increased degradation of extracellular DNA at low pH of already-lysed organisms, rather than death due to low pH itself. It is of course still possible that there is widespread death due to low pH in these microcosms, but without further information, it is difficult to tell. Therefore, I recommend one of two options:

(i) Please demonstrate experimentally that there is widespread death in how you set up your microcosms at low pH (not just due to cell lysis upon wet-up followed by increased DNA degradation with decreasing pH). If the original soils are no longer available (which is very understandable), I think it would be okay to use other soils. A biomarker other than sequenced DNA would be convincing (though if you do use sequencing, I recommend extracting DNA from only the cellular fraction to avoid potential noise from extracellular DNA). Otherwise...

(ii) I recommend revising the language of your manuscript so that death is not the primary stated mechanism behind lower functional biomass activity at lower pH. There are numerous other mechanisms by which this could occur, and which precise mechanism it is does not strongly affect your general conclusions. However, without such an abrupt biological difference between the low and neutral pH microcosms to further differentiate Regimes I and II, I would caution against emphasizing the qualitative separation between these regimes and would acknowledge that it is a continuum.

References for literature cited above:

Jones, R. T. et al. A comprehensive survey of soil acidobacterial diversity using pyrosequencing and clone library analyses. The ISME Journal 3, 442–453 (2009).

Rousk, J. et al. Soil bacterial and fungal communities across a pH gradient in an arable soil. The ISME Journal 4, 1340–1351 (2010).

Blazewicz, S. J. et al. Taxon-specific microbial growth and mortality patterns reveal distinct temporal population responses to rewetting in a California grassland soil. The ISME Journal 14, 1520–1532 (2020).

An, R. et al. Non-Enzymatic Depurination of Nucleic Acids: Factors and Mechanisms. PLoS ONE 9, e115950 (2014).

Response: Thank you for sharing the insights on the rewetting phenomena and DNA depurination and for your justified skepticism of our claim of death in Regime I. We also appreciate you suggesting both an experiment or the opportunity to revise language. Reviewer 1 asked a similar question regarding DNA degradation, as a result, we felt this was an important point so we opted to do the experiment.

We added a new section in our manuscript to comprehensively address this comment (SI: “Justifying widespread death as the mechanism of Regime I” (Lines 1547-1767) with new supplementary **Figs. S21, S22, S23, and S24**). We summarized what we wrote in the SI in the response to **Reviewer 1 - Comment #7** (click to move to the comment). There, we provided a compact description of the experiments and analysis demonstrating that cell death is prevalent in Regime I, despite DNA degradation at low pH. We kindly direct you to read that first and proceed reading the rest of the response.

Here, we will address three points that are specific to your comment.

1. Firmicutes (Bacillota) may not be immune to lysis during rewetting according to Blazewicz *et al.* (ISMEJ, 2020):

To compare the mortality rate induced by rewetting between Firmicutes (Bacillota) and other phyla, we used the death rate data from Blazewicz *et al.* (ISME J, 2020), where they measured death rates by sequencing 16S rRNA genes coupled with heavy water ($H_2^{18}O$) DNA quantitative stable isotope probing. We plotted histograms of death rates (16S rRNA gene copies/day, d.boot.median column of the table) of TaxonID under each phylum in each time point (3, 24, 72, 168 hours) while overlaying Firmicutes’ (Bacillota) distribution (total of 7 data points) in pink color (**Fig. Rev3-2-2**). To test nonparametrically whether the death rate of Firmicutes (Bacillota) is significantly lower than that of other phyla, we used bootstrapping to resample with replacement 10,000 times from the distribution of each phylum at each time point, calculating the 0.05 quantile value (vertical dotted line).

Figure Rev3-2-2. Histograms of median death rates (16S rRNA gene copies/day) of taxa within each phylum over time points (3, 24, 72, 168 hours) for data from Blazewicz *et al.* (ISME J, 2020). Pink histograms are the death rate distribution of the Firmicutes (Bacillota) phylum (total 7 data points). To test nonparametrically whether the death rate of Firmicutes (Bacillota) is significantly lower than that of other phyla, we used bootstrapping to resample with replacement 10,000 times from the distribution of each phylum at each time point, calculating the 0.05 quantile value. Vertical dotted black lines are 0.05 quantile value of death rates for each phylum and time point.

We found that the death rate during rewetting is also high for Firmicutes (Bacillota), with no statistically significant difference compared to Proteobacteria (Pseudomonadota) or

Bacteroidota. According to the bootstrapping test, Firmicutes' (Bacillota) death rates during rewetting were not significantly lower than that of Proteobacteria (Pseudomonadota) or Bacteroidota, except for one case of Proteobacteria (Pseudomonadota) at 72 hr. There were mixed results when they were compared to other phyla. Compared to Planctomycetes, Chloroflexi, and Verrucomicrobia, Firmicutes (Bacillota) death rates were not significantly lower except for the 3 hr time point, while they were significantly lower at many time points compared to Actinobacteria and Acidobacteriota (**Fig. Rev3-2-2**). This result suggests that Firmicutes (Bacillota) like many other phyla is producing significant amounts of relic DNA during rewetting (at least 10^6 - 10^7 16S rRNA genes/day). Contrary to the Reviewer's expectation, we contend that Firmicutes (Bacillota) are not resistant to lysis during rewetting.

2. Other than Firmicutes (Bacillota), death is prevalent in other phyla in Regime I (Proteobacteria (Pseudomonadota) and Bacteroidota):

Your previous suggestion of experiments was to use soils to demonstrate that the observed decrease in the sequence-based abundance is not due to cell lysis upon wet-up, but true acid-induced death. Because it is hard to rule out the effect of rewetting when we use the soil substrate, we incubated individual strains in defined media after isolating the strains of interest to measure death rates across different pH levels. In addition, we extracted genomic DNA from each of these isolates to measure the DNA degradation rates across different pH levels (see response to **Reviewer 1 - Comment #7** and SI: "*Justifying widespread death as the mechanism of Regime I*" for a detailed description). To test whether DNA degradation alone (without death) could explain sequence abundance changes, we constructed a no-death model (death rate $d = 0$) using measured DNA degradation rates (δ) for each phylum and pH condition. We compared the no-death model predictions for endpoint abundance with data from chloramphenicol-treated (CHL+) samples, where growth was inhibited. At pH 5-7, no-death model's prediction of endpoint abundance did not significantly deviate from the observed abundance data, confirming that chloramphenicol does not induce cell death. However, at pH 3-4, significant deviations of no-death model predictions from abundance data of Proteobacteria (Pseudomonadota) and Bacteroidota were observed, particularly at pH 3.5 and 4 ($p < 0.05$). Even at pH 3, where DNA degradation rates were highest, sequence abundance changes required death to explain the data (when assuming 50% relic DNA). These results **provide strong evidence for widespread acid-induced death in Regime I** (new **Fig. S22C**, see response to **Reviewer 1 - Comment #7**).

3. Regime I and II are a continuum.

While the evidence for death at low pH (Regime I) is convincing, we agree that other mechanisms could also contribute to the observed changes in functional dynamics (specifically the decrease in our $\tilde{x}(0)$ parameter). Therefore, we agree that we should be careful emphasizing the qualitative separation between Regime I and II likely has some contributions from death but other mechanisms are likely also at work.

Action taken: In the main text, we added a sentence addressing the possibility of degradation of relic DNA in soil (Lines 315-316) and acknowledged the smooth transition between Regimes II and I can be due to other ecological mechanisms (Lines 323-325). In response to this comment, Rev. 1's Comment #2.2 and #7, we added a new section (SI: "*Justifying widespread death as the mechanism of Regime I*" with new supplementary **Figs. S21, S22, S23, and S24**) confirming death in Regime I despite DNA degradation (Lines 1547-1767).

Minor comments and suggestions:

Rev3-Minor 1. *I believe there are some instances in the discussion where it would be appropriate to temper the language and claims. For example:*

- Lines 357-359: "This demonstrates that understanding the community response to perturbation may not require grappling with every metabolic process or interaction in the community, but only with a handful of key features." I believe it would be worthwhile to acknowledge (perhaps in the "Limitations" section, which I appreciate) that this statement applies to the authors' system likely because they focused on one metabolic function that is catalyzed by a single enzymatic step, and so it is perhaps easier to find a model here than for a process that is dependent on more metabolic steps and/or interspecies interactions.

Action taken: We agree. We made the change stating that this model may be more complex depending on the complexity of the metabolic process. We added these sentences: "A key question going forward is whether this approach can reveal similar insights into soil microbiome responses to other perturbations." (Lines 417-419), "Further generalizing regimes beyond pH perturbations is an important avenue for future work." (Lines 422-423).

Rev3-Minor 2. *- Lines 392-393: "... the metabolism of a single strain can mirror the metabolism of the soil microbiome." It is not surprising that there is a strain that has the same coarse-grained nitrate reduction rates as some communities, and the focus of this study is not on how to build up an understanding of microbial community function from the physiology of its individual members.*

Action taken: Fair point. We removed that part from the sentence.

Rev3-Minor 3. *- Lines 394-398: "More broadly, the discovery of the three functional regimes, including the nutrient-limiting regime, is notable because it reflects a potential duality between the physiology of an ecosystem and the three phases of a cell: growth (Regime III), stationary (Regime II), and stress (Regime I). This duality suggests the possibility that cellular physiology might provide a conceptual framework for understanding the ecosystem." I understand that discussions are a space for large-scale thinking and speculation, but I believe this statement is a great oversimplification. Cells are experiencing all three of these phases across the functional regimes, and the relative contribution of each growth phase to the make-up of the cells in each regime is not quantified. (This statement is also in contradiction to the authors' own interpretation of the shift from Regime II to Regime III involving Regime II taxa experiencing stress; see line 377).*

Action taken: We believe we have not explained this duality clearly and it may be true that this is an oversimplification. We removed these sentences.

Rev3-Minor 4. *My apologies if you have included this information and I missed it: could you please provide values for the increase in the rate of nitrate reduction for all of the samples, indicating to which regime the sample belongs? Can the distribution of these values also be used to define the regimes (in addition to the thresholds for the inferred biomass activity and available limiting nutrient)? To what extent is there overlap in these values for Regimes I vs. II?*

Response: No need to apologize! We will interpret your 'increase' in nitrate reduction rate as the fold change in rate for CHL- (endpoint) relative to CHL+. Throughout our paper, the constant CHL+ slope (rate) is used as the reduction rate at T0, which is equal to the indigenous biomass activity parameter $\tilde{x}(0)$. If you meant the increase in nitrate reduction rate as the difference in rate (delta rate) for CHL- (endpoint) compared to T0, then it would be exactly equivalent to the available limiting nutrient parameter $\gamma^*\tilde{C}(0)$, which we are already using to define the regimes. Given these definitions, the fold increase of nitrate reduction rate (rate of CHL-/rate of CHL+) is $\tilde{x}(0) + \gamma^*\tilde{C}(0)$ (rate of CHL-(endpoint)) over $\tilde{x}(0)$ (rate of CHL+), resulting to $1 + \gamma^*\tilde{C}(0) / \tilde{x}(0)$. These values are what we used for the y-axis in **Fig. 4A** and **Fig. S11A**.

Figure Rev3-M4 (A) Rate fold increase values are visualized as color gradients on a grid of perturbed pH (x-axis) and native pH (y-axis). Darker colors indicate higher rate fold increases (log10 scale). (B) Histograms of rate fold increase values (CHL-/CHL+) are shown for each regime (Regime 1, 2, and 3), with a log10-transformed x-axis. Frequencies (y-axis) represent the distribution of rate fold increase values within each regime.

Because the rate fold increase, $1 + \gamma^*\tilde{C}(0) / \tilde{x}(0)$, is already a function of $\gamma^*\tilde{C}(0)$ and $\tilde{x}(0)$, it would be redundant if we used these values in addition to the thresholds for the indigenous biomass activity and available limiting nutrient values. When we plot these values (**Fig. Rev3-M4A**) and compare the result with **Fig. 3C** or **D**, we can see that the rate increase values are very high at extreme basic conditions similar to available limiting nutrient (**Fig. 3D**). It gets even higher than the available limiting nutrient parameter $\gamma^*\tilde{C}(0)$, because it is divided by the indigenous biomass activity parameter $\tilde{x}(0)$, which approaches 0 as we enter the Regime III. There are a few samples where rate increase values are also high in Regime I, but this is a result of $\tilde{x}(0)$ being close to 0 in the Acidic death regime. Other than these unusual spikes in Regime I, there is an overlap in rate increase values between Regime I and II (**Fig. Rev3-M4B**). Therefore, only by having $\gamma^*\tilde{C}(0)$ and $\tilde{x}(0)$ separately can we discern Regime I and II, the signature of Regime I being the decline of $\tilde{x}(0)$. Therefore, we can conclude that using the rate increase does not add additional value than our currently used 2 parameters.

Rev3-Minor 5. *Fig. 3A: The y-axis is strange since it's not really on a log scale - the represented distance between $10e-2$ and $10e-1$ is much smaller than $10e-1$ to $10e0$ to $10e1$. Why did you choose to represent it like this?*

Response: You're right, this is a very astute comment. Thanks for pointing out our lack of description. We used a $\log_{10}(x+0.01)$ transformation instead of a \log_{10} transformation. This choice was made to prevent very small parameter values (less than 0.01) from dominating the entire parameter space. By adding a small number, we can minimize the impact of these small values for visualization and better highlight the variations in larger values.

Action taken: We added the note ' $\log_{10}(x+0.01)$ transformation for visualization' to the Fig. 3 caption.

Rev3-Minor 6. *Fig. 5D caption: It would be helpful to note that the points in Fig. 5D are a subset of those in Fig. 3A because only half of the samples were sequenced (rather than saying it is the same plot).*

Action taken: We added this note at the end of the Fig. 5 caption for panels A, B, and D.

Rev3-Minor 7. *Fig. S7 (right): The model seems to be biased to have the highest error for the samples that are at the Regime II/III boundary. Could you please comment on the direction of the error and why this might be?*

Response: Amazing question, which we have discussed previously at some length but felt it was too detailed for the main text. The highest error at the Regime II/III boundary is the major failure mode of our 1-biomass model. We believe that there is some ecological signal to this failure. In essence, what is happening is that the increase in nitrate reduction rate cannot be captured by a single biomass due to a higher rate of reduction in CHL- at T0 relative CHL+ AND and late time increase in the CHL- nitrate reduction rate.

Action taken: Therefore, we added a new section in the SI to comprehensively address all potential failure modes of the 1-biomass model (see full analysis and discussion in the new SI section "*Justifying the effective 1-biomass model despite the diversity of denitrifying taxa*" (Lines 1768-1848) and **Fig. S34-36**). We mentioned this in the Discussion section for readers who may be interested in the failure modes of the effective 1-biomass model (Lines 409-410).

In summary, the 1-biomass model fails at the Regime II/III boundary because this transition involves both slow-growing taxa with high initial abundance (Regime II) and fast-growing taxa with low initial abundance (Regime III), creating dynamics that the model cannot simultaneously capture with the 1-biomass model (see **new Fig. S35** for examples of nitrate dynamics). From the sequencing data, we confirmed that the former are Proteobacteria (Pseudomonadota) and Bacteroidota (Regime II strains), and the latter are the Firmicutes (Bacillota) (Regime III strains) (**Fig. 3**). To systematically verify this failure mode, we conducted simulations using a 10-taxon model, varying anti-correlated initial abundances $x(0)$ and growth rates to identify conditions under which deviations might occur (**new Fig. S34C**). As expected, the 1-biomass model predictions had the highest error in these dynamics with both high initial slope and late-time exponential growth (**new Fig. S34D**). However, this effect is very subtle, and even in this regime, the 1-biomass model correctly captures the fact that the late-time growth must indicate a large metabolic activity of a taxon that starts at low initial

abundance---the signature of the Resurgent growth regime, independently confirmed with the sequencing data.

Rev3-Minor 8. *There are some instances where $x(0)$ is referred to as biomass (e.g. Lines 182 and 193), rather than biomass activity, which was confusing on a first read. It would help with the clarity of the writing if the latter were used consistently.*

Action taken: Previously, for $x(t)$, we initially used a mix of terms such as ‘functional biomass’, ‘active biomass’, and ‘biomass activity’. To reduce confusion, we changed all references to $x(t)$ to ‘functional biomass’ throughout the paper. Because $\tilde{x}(t) = x(t) * rA$, $\tilde{x}(t)$ (the one with tilde) has units of a rate and is the activity from the functional biomass ($x(t)$), we changed all instances referring to $\tilde{x}(0)$ from ‘indigenous functional biomass’ to ‘indigenous biomass activity’.

Referee #3 (Remarks on code availability):

There is no code provided.

Response: We sincerely apologize. The codes associated with the paper are in the Open Science Framework (<https://doi.org/10.17605/OSF.IO/CTF8K>).

Sepe Kuehn Ph.D.
Associate Professor

*Department of Ecology and Evolution
Center for the Physics of Evolving Systems
The University of Chicago*

Gordon Center for Integrative Science
929 E. 57th St.
Chicago, IL 60637

607.351.2041
sepekuehn@uchicago.edu
kuehnlab.org

April 1, 2025

Dear Dr. Editor

Thank you for the opportunity to revise our manuscript further. Below, we have responded to the additional comments from Reviewer #1. We have also revised the manuscript, with all changes marked in red.

Sincerely,

Sepe Kuehn (on behalf of all authors)

Referee #1 (Remarks to the Author):

The authors have presented an impressive, expanded analysis in response to the criticisms voiced by the previous reviews. We appreciate that the regimes hold – at least qualitatively – at different sites and that the simple model can capture these dynamics.

We thank the referee for their positive response to our first revision and their attention to the details of the data we presented. Both comments below address the question of variation in the taxonomic responses (growth) across different sites. We present a high-level response to these questions and a technical analysis to support our claims.

General response: The referee is correct to point out that the conservation of functional regimes across sites is qualitative. Indeed, the nitrate utilization dynamics vary quantitatively from site to site, some with higher overall rates than others. This quantitative variation is reflected in differences in the *parameters* of the model across sites. As the referee appreciates, irrespective of this quantitative variation, the model describes the data well. Similarly, for the taxonomic (growth) data, while the quantitative features of the abundances vary across sites, the qualitative features are conserved. Our presentation of the data in the first revision obscured this, so we understand the source of the referee's concern. Below, we correct this oversight to demonstrate the qualitatively conserved taxonomic responses across sites.

Rev1-Comment1.1. *However, the analysis of community dynamics at the additional sites shows very high variation. While in Fig. 5, Pseudomonadota and Bacteriodota are lumped and presented as growing in regime II, Fig. 6 actually shows very high variation and much less clear trends.*

Pseudomonadota are highly variable and appear to also respond at high pH, and Bacteriodota mostly respond at high pH, contradicting the results presented in Fig. 5.

Response: In our original submission, the presentation of the data confused the conserved qualitative trends in the sequencing data. There were three challenges in our presentation of the data in the first revision. (1) The non-negative matrix factorization of the growth folds in the supplement was shown in two separate plots (**Figures S15** and **S27**) that were hard to compare due to different color schemes. (2) The presentation of the sequencing results in **Figure 5** showed Pseudomonadota and Bacteriodota grouped, and in **Figure 6** these phyla were plotted separately. (3) In **Figure 5**, the y-axis of panel B growth folds was log-scale, and in **Figure 6**, it was not.

First, when we perform non-negative matrix factorization (NMF) on the Cook Agronomy Farm (CAF) and Revision 1 soils *separately*, we observe highly conserved patterns in the phyla that dominate the response of the soil to pH perturbations. Namely, Pseudomonadota and Bacteriodota comprise one axis of variation and Bacilliota the other – and **this is the case for both datasets** (see **Figure Rev1-1-1** for side-by-side comparison). These results were separated previously (**Figures S15**, and **S27**, making them hard to compare). In addition, for Axis #1 in *both datasets* Bacteriodota has a higher weighting than do Pseudomonadota, with the latter projection being closer to the background (points near the origin). This reflects the lower overall importance of Pseudomonadota and their higher variability, as noted by the referee. Still, it is striking that these two analyses show the same basic patterns.

Second, given the lumped Pseudomonadota and Bacteriodota in **Figure 5** and the separated presentation of these phyla in **Figure 6**, the referee correctly questions conserved patterns in the response of these phyla between the two datasets. In **Figure Rev1-1-2** we plot growth folds across soils and pH perturbations for both experiments side by side (left and right columns), both with log-scale y-axes. In this presentation of the data, we see the conserved qualitative features of the response:

1. Pseudomonadota exhibit higher responses at larger pH perturbations in both datasets.
2. Bacteriodota increase with increasing pH (coherent with nutrient release hypothesis in Regime II) and then drop at the highest pH perturbations (Regime III). Also true in both datasets.
3. The lumped Pseudomonadota and Bacteriodota show similar trends in both experiments. We note that the Pseudomonadota are at higher relative abundance than the Bacteriodota, so they dominate the pattern when the two phyla are combined.
4. The Bacilliota increase at the extreme basic pH in all soils (although the growth fold is much higher in some soils than others). In addition, there is an increase in Bacilliota in acidic conditions for some soils in *both* CAF and the Revision experiment. We note that this quantitative detail of the Bacilliota response depends on the soil.

CAF soils

Revision soils

Figure Rev1-1-1. Results of performing non-negative matrix factorization (NMF) separately on the Cook Agronomy Farm (CAF, left column) and Revision 1 soils (right column). The top panel shows the weight on the 2 axes from NMF for each major phyla, denoted by the different colors (see legends). The bottom 2 panels show the 2 axes from the NMF decomposing the growth folds across the perturbed pH.

Figure Rev1-1-2. Growth folds (absolute abundance ratio of CHL-/CHL+ conditions) of Pseudomonadota, Bacteroidota, and Pseudomonadota + Bacteroidota (lumped abundance), and Bacillota are plotted to compare CAF (Cook Agronomy Farm) and 4 revision soils (LaBagh, Pinhook, CLG13, ELG13, denoted by colors) with 3 biological replicates. The Y-axis is transformed with $\log(\text{Growth fold} + 0.01)$ as we did for Figure 5. Data points with CHL+ absolute abundance of 0 or less than 0.0001 were removed due to making the growth fold infinite or very large.

Rev1-Comment1.2. *Bacillota* only appear to respond in 2 samples and appear to have a peak at low and high pH in the same samples. This high variation questions how the data are presented in Fig. 5 and the generality of the taxonomic analysis.

Other than that, we are happy with the changes made to the paper.

Response: Again, this comment is astute, but reflects our insufficient presentation of this data in the first revision. In **Figure Rev1-2-1** we show the *Bacillota* growth folds for each of the four soils presented in the revision experiments. All show an increase in the extreme basic (Regime III) direction, which was obscured previously by a linear y-axis in Figure 6 and large differences in the magnitude of the response. We agree, as stated above, that the differences are quantitative, and the agreement is qualitative.

Second, the referee points out the response of *Bacillota* in the acid perturbed regime for some samples. We would like to note that this response was also observed in samples in the CAF dataset from our original submission (**Figure Rev1-1-2**, bottom left). We see this occurring in the Pinhook and LaBagh soils from the revision experiment as well (**Figure Rev1-2-1**).

Figure Rev1-2-1. Growth folds (Absolute abundance ratio of CHL-/CHL+ conditions) of *Bacillota* (Firmicutes) are plotted for each new soil separately with 3 biological replicates. The Y-axis is transformed with $\log(x+0.01)$.

Action taken: To address these comments in the manuscript, we remade **Figure 6D** comparable to **Figure 5 A, B**, by plotting the lumped Pseudomonadota and Bacteroidota and a log-scale y-axis. Furthermore, to clearly show all major phyla's growth fold-variation across the perturbed pH, we added panel **B** in **Figure S27** showing growth folds of 10 major phyla similar to our **Figure S15**. We fixed an error in the coloring of the points, which made Figure S15 and **Figure S27** harder to compare.

Second, we wanted to emphasize more clearly the qualitative nature of regimes across soils. In our first revision, we used the word "qualitative" to describe the functional dynamics across all soils tested (line 332), but our language regarding the taxonomic patterns was unclear. Therefore, we added language to more explicitly state that the patterns of growth differ from site to site in their quantitative details but that the qualitative patterns are conserved (lines 336-341 in revision 2).

Referee #2 (Remarks to the Author):

The authors have addressed all of my concerns!

Referee #2 (Remarks on code availability): The code seems well documented and well-structured. I have not attempted to run it.

We thank the referee for their careful consideration of our manuscript.

Referee #3 (Remarks to the Author):

The authors have satisfactorily addressed my concerns and comments, and I appreciate the extensive additional experiments and analyses that the authors performed to strengthen their manuscript.

Referee #3 (Remarks on code availability): All code is available and is reasonably readable/accessible to readers.

We thank the referee for their careful consideration of our manuscript.